# OPTIMIZING NEURAL NETWORK REPRESENTATIONS OF BOOLEAN NETWORKS

**Joshua Russell   Ignacio Gavier   Devdhar Patel   Edward Rietman   Hava T. Siegelmann**
University of Massachusetts Amherst
`{jgrussell,igavier,devdharpatel,erietman,hava}@umass.edu`

## ABSTRACT

Neural networks are known to be universal computers for Boolean functions. Recent advancements in hardware have significantly reduced matrix multiplication times, making neural network simulation both fast and efficient. Consequently, functions defined by complex Boolean networks are increasingly viable candidates for simulation through their neural network representation. Prior research has introduced a general method for deriving neural network representations of Boolean networks. However, the resulting neural networks are often suboptimal in terms of the number of neurons and connections, leading to slower simulation performance. Optimizing them while preserving functional equivalence –lossless optimization– is an NP-hard problem, and current methods only provide lossy solutions. In this paper, we present a deterministic algorithm to optimize such neural networks in terms of neurons and connections while preserving functional equivalence. Moreover, to accelerate the compression of the neural network, we introduce an objective-aware algorithm that exploits representations that are shared among subproblems of the overall optimization. We demonstrate experimentally that we are able to reduce connections and neurons by up to $70\%$ and $60\%$, respectively, in comparison to state-of-the-art. We also find that our objective-aware algorithm results in consistent speedups in optimization time, achieving up to $34.3\times$ and $5.9\times$ speedup relative to naive and caching solutions, respectively. Our methods are of practical relevance to applications such as high-throughput circuit simulation and placing neurosymbolic systems on the same hardware architecture.

## 1   INTRODUCTION

Universal approximation theorems (Hornik et al., 1989) and Turing-completeness (Siegelmann & Sontag, 1992) provide the theoretical foundation for implementing computable functions with neural networks (NNs). At present, the predominant methods for implementing functions with NNs are learning-based techniques, which rely on optimization algorithms, data, and compute to elicit a function of interest (LeCun et al., 2015). While these techniques provide a solution for obtaining functions that are difficult to programmatically specify, they come at the cost of unsustainable energy usage (Strubell et al., 2019), ever-longer training times, and safety concerns due to the current lack of understanding surrounding functions derived from learning-based techniques (Qi et al., 2023).

Another much less prevalent implementation method for NNs is to synthesize a network from a known functional specification. The specification may be in the form of an explicit construction, detailing the network structure, neurons, and connections (Hewitt et al., 2020; Karuvally et al., 2024), or it may be provided in a programming language, requiring a method to convert the specification into a resulting NN (Siegelmann, 1994; Gruau et al., 1995; Neto et al., 2003). These implementation methods generally require significantly less time and energy in comparison to learning-based techniques and provide guarantees on NN functionality which are critical for safety and security.

One such programmatic implementation method is that of NN-based technology mapping (Patel et al., 2022; Gavier et al., 2023), the problem of finding a NN that is *functionally equivalent* to a Boolean network (BN) (i.e., $\forall \boldsymbol{x} \in \{0,1\}^n, NN(\boldsymbol{x}) = BN(\boldsymbol{x})$). This method holds significant potential for

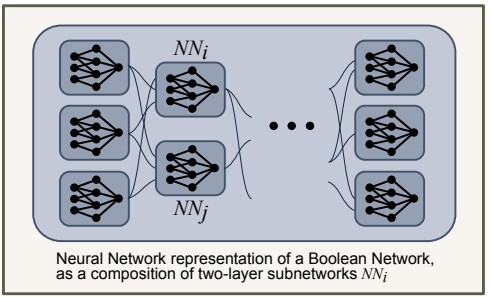 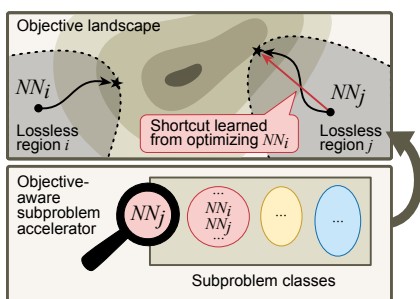

Figure 1: Summary of our optimization problem. A neural network (NN) representing a Boolean network (BN) can be constructed as a composition of two-layer sub-NNs $NN_i$ (left). Optimizing the size of the NN is performed by optimizing each sub-NN $NN_i$ (top right). To maintain functional equivalence of the original NN, it is necessary that each subproblem solution lies within its *lossless* region. Subproblems can be accelerated for sub-NNs $NN_j$ that have shared representation with (i.e., are in the same class as) already optimized sub-NNs $NN_i$ (bottom and top right).

applications in the integrated circuits industry and for neurosymbolic hardware architectures; however, the current state-of-the-art (SOTA) solution of Gavier et al. (2023) faces two significant drawbacks: (1) the resulting NNs are often suboptimal in terms of neurons and connections, and (2) slow execution time makes the technique less appealing for industrial settings.

In light of these shortcomings in the SOTA, our **first goal** is to develop a technique for optimizing NN size while maintaining functional equivalence. NN compression techniques, such as pruning and quantization, have been extensively researched under the learning-based paradigm (Liang et al., 2021; Li et al., 2023). However, in general, pruning techniques are *lossy* and do not guarantee preservation of functional equivalence (Vadera & Ameen, 2022). Moreover, since optimal bit widths for parameters and activations can be calculated analytically following NN-based technology mapping, further quantization cannot be applied as it would break functional equivalence. While *lossless* compression techniques that maintain functional equivalence have been presented for ReLU (Serra et al., 2020) and single-hidden-layer hyperbolic tangent (Sussmann, 1992; Farrugia-Roberts, 2024) NNs, they are not applicable to the Heaviside threshold NNs we investigate in this work.

Figure 1 (left) depicts the NN representation of a BN as proposed by Gavier et al. (2023); a composition of two-layer sub-NNs. Optimizing the size of the NN can be thought of as the aggregation of subproblems, one for each sub-NN, as shown in Figure 1 (top right). Our **second goal** is to develop an algorithm that exploits shared representations among these subproblems to accelerate the NN optimization process, which is key for dealing with the large-scale BNs faced in industrial applications.

> **Summary:** The SOTA solution for NN-based technology mapping does not effectively optimize the resulting NN representation. Since existing NN compression techniques are either lossy or inapplicable, the challenge of efficiently optimizing NN representations of BNs remains open.

**Contributions.** We introduce a practical methodology to overcome these challenges:

- We propose a lossless technique for optimizing a two-layer NN representation of a Boolean function (see § 3.1). We present two objectives that correlate with minimizing neurons or connections.
- We propose an objective-aware optimization algorithm based on Negation-Permutation-Negation (NPN) classification, exploiting shared representations among the two-layer sub-NNs (see § 3.2). We show that our algorithm consistently accelerates optimization time, achieving up to $34.3\times$ and $5.9\times$ speedup relative to naive and caching solutions, respectively (see Figure 6).
- We propose an architecture-aware lossless optimization algorithm that selects which two-layer sub-NNs to minimize (see § 3.3). We demonstrate experimentally that connections and neurons can be reduced by up to $70\%$ and $60\%$ in comparison to SOTA (see Figure 5).

## 2 BACKGROUND

In this section we introduce the necessary definitions and background for NN-based technology mapping and our optimization algorithms. The essential notions are covered here, while a comprehensive treatment is provided in the appendix.

### 2.1 DEFINITIONS

**Notation.** We denote the set of binary numbers as $\mathbb{B} = \{0, 1\}$, the set of natural numbers (including 0) as $\mathbb{N}_0 = \{0, 1, 2, \dots\}$, and the set of integers as $\mathbb{Z} = \{\dots, -2, -1, 0, 1, 2, \dots\}$.

**Definition 2.1** (Boolean function). A $k$-input, (single-output) Boolean function (BF) is a function $f : \mathbb{B}^k \to \mathbb{B}$, $\boldsymbol{x} \mapsto f(\boldsymbol{x})$, where $\boldsymbol{x} = (x_1, \dots, x_k)$ is a bit vector representing the $k$ input variables $x_i \in \mathbb{B}$, $1 \le i \le k$.

**Definition 2.2** (Truth table representation of a BF). The truth table representation is an expansion of BF $f : \mathbb{B}^k \to \mathbb{B}$ in a basis of indicator functions $f(\boldsymbol{x}) = \sum_{\boldsymbol{a} \in \mathbb{B}^k} f(\boldsymbol{a}) \mathbb{1}_{\boldsymbol{a}}(\boldsymbol{x})$, where $\mathbb{1}_{\boldsymbol{a}} : \mathbb{B}^k \to \mathbb{B}$ is defined as $\mathbb{1}_{\boldsymbol{a}}(\boldsymbol{x}) = 1$ if $\boldsymbol{x} = \boldsymbol{a}$, and $0$ otherwise. The truth table vector $\boldsymbol{f} \in \mathbb{B}^{2^k}$ of $f$ is the vector of coordinates $f(\boldsymbol{a})$ in such a basis. E.g., the truth table vector of $f(a, b) = a \vee b$ is $\boldsymbol{f} = [0, 1, 1, 1]^\mathsf{T}$.

**Definition 2.3** (Multilinear polynomial representation of a BF). The multilinear polynomial (MP) representation is an expansion of BF $f : \mathbb{B}^k \to \mathbb{B}$, extended to $p : \mathbb{Z}^k \to \mathbb{Z}$, in the basis (O'Donnell, 2014) of logical AND functions $p(\boldsymbol{x}) = \sum_{S \subseteq \{1, \dots, k\}} \hat{f}(S) \chi_S(\boldsymbol{x})$, where $\chi_S : \mathbb{Z}^k \to \mathbb{Z}$ is defined as $\chi_S(\boldsymbol{x}) = \prod_{i \in S} x_i$ and $S \subseteq \{1, \dots, k\}$. The MP vector $\boldsymbol{p} \in \mathbb{Z}^{2^k}$ of $f$ is the vector of coordinates $\hat{f}(S)$ in such a basis. E.g., if $f(a, b) = a \vee b$ then $p(a, b) = a + b - ab$ and $\boldsymbol{p} = [0, 1, 1, -1]^\mathsf{T}$.

**Definition 2.4** (Boolean network). A Boolean network (BN) $G_{\mathcal{F}} = (V, E)$ is a directed-acyclic graph (DAG), with vertex labeling function $\psi : V \to \mathcal{F}$, that represents an $n$-input, $m$-output BF $\mathcal{B} : \mathbb{B}^n \to \mathbb{B}^m$. Vertices with indegree zero are *primary inputs* (PI), and vertices with outdegree zero are *primary outputs* (PO). The labeling function $\psi$ assigns a single-output BF to each non-PI vertex in the graph. A directed edge $(u, v) \in E$ exists if the output of $u$'s BF is an input to $v$'s BF. The depth of vertex $v \in V$, denoted $depth(v) \in \mathbb{N}_0$, is the length of the longest path from any PI to $v$. The depth of a BN, denoted $depth(G_{\mathcal{F}})$, is the maximum vertex depth in the BN.

**Definition 2.5** (Neural network). Let $L \in \mathbb{N}_0$ and $(D_0, D_1, \dots, D_L) \in \mathbb{N}_0^{L+1}$. A neural network (NN) $\mathcal{N} : \mathbb{B}^{D_0} \to \mathbb{B}^{D_L}$ is a function with specification

$$\mathcal{N}(\boldsymbol{x}) = \sigma\left( W^L \left( \sigma\left( W^{L-1} \left( \dots \sigma\left( W^1 \boldsymbol{x} + \boldsymbol{b}^1 \right) \dots \right) + \boldsymbol{b}^{L-1} \right) \right) + \boldsymbol{b}^L \right), \tag{1}$$

where $\sigma$ is the Heaviside step function[1] and $\forall l \in \{1, \dots, L\}$, $W^l \in \mathbb{Z}^{D_l \times D_{l-1}}$ and $\boldsymbol{b}^l \in \mathbb{Z}^{D_l}$. The depth of a NN is denoted $depth(\mathcal{N}) = L$.

**Definition 2.6** (Functional equivalence). Two functions $f, g : \mathcal{X} \to \mathcal{Y}$ are functionally equivalent if $\forall x \in \mathcal{X}, f(x) = g(x)$.

### 2.2 OBTAINING A NEURAL NETWORK REPRESENTATION OF A BOOLEAN NETWORK

Here we review the SOTA for NN-based technology mapping (Gavier et al., 2023). The method takes a BN $G_{\mathcal{F}}^{init}$ as input and, through a sequence of mappings, converts it into a functionally equivalent NN.

**LUT-based technology mapping.** $K$-LUT-based technology mapping (Cong & Ding, 1994), $K \in \mathbb{N}_0$, is used to convert $G_{\mathcal{F}}^{init}$ into a functionally equivalent BN $G_{\mathcal{F}_{\boldsymbol{f}}}$ with function set $\mathcal{F}_{\boldsymbol{f}} = \cup_{k=0}^K \mathbb{B}^{2^k}$ consisting of $k$-input look-up tables (LUTs), or truth table vectors $\boldsymbol{f} \in \mathbb{B}^{2^k}$, $k \le K$.

**MP-based technology mapping.** Following $K$-LUT-based technology mapping, each truth table vector $\boldsymbol{f} \in \mathbb{B}^{2^k}$, $k \le K$, in BN $G_{\mathcal{F}_{\boldsymbol{f}}}$ is mapped to a functionally equivalent MP vector $\boldsymbol{p} \in \mathbb{Z}^{2^k}$. The resulting function set is $\mathcal{F}_{\boldsymbol{p}} = \cup_{k=0}^K \mathbb{Z}^{2^k}$. Each conversion from truth table vector to MP vector can be computed via a divide-and-conquer algorithm in time $O(2^k k)$ (Gavier et al., 2023). We denote this conversion algorithmically as `ttToMP`.

---

[1] $\sigma(x) = 1$ if $x > 0$, and $0$ otherwise, and it is extended to an element-wise function.

**NN-based technology mapping from MP BNs.** Following MP-based technology mapping, each MP vector $\boldsymbol{p} \in \mathbb{Z}^{2^k}$, $k \leq K$, in BN $G_{\mathcal{F}_p}$ is mapped to a functionally equivalent two-layer NN $\zeta : \mathbb{Z}^k \to \mathbb{Z}$, $\zeta(\boldsymbol{x}) = W_2\sigma(W_1\boldsymbol{x} + \boldsymbol{b}_1)$ with parameters $(W_1, \boldsymbol{b}_1, W_2) \in \mathbb{Z}^{(2^k \times k)+(2^k)+(1 \times 2^k)}$. The resulting function set is $\mathcal{F}_{\mathcal{N}} = \cup_{k=0}^{K} \mathbb{Z}^{(2^k \times k)+(2^k)+(1 \times 2^k)}$. See Figure 2 for an example of $k = 2$.

**Intra-depth parallelization & the Unmerged NN.** Let $G_{\mathcal{F}_{\mathcal{N}}} = (V, E)$ be a BN following NN-based technology mapping in which all vertices $v \in V$ have incoming edges only from vertices at depth $depth(v) - 1$, that is, there are no "skip connections" (this can be achieved by adding additional vertices that compute the identity function). Moreover, let $V_d = \{v \in V \mid depth(v) = d\}$ be the set of vertices at depth $d$, and $d \in \{1, \ldots, depth(G_{\mathcal{F}_{\mathcal{N}}})\}$. Then the output of all vertices in $V_d = \{v_1, v_2, \ldots, v_{|V_d|}\}$ can be computed *in parallel* as

$$\begin{bmatrix} \zeta(\boldsymbol{x}^{v_1}) \\ \vdots \\ \zeta(\boldsymbol{x}^{v_{|V_d|}}) \end{bmatrix} = \begin{bmatrix} W_2^{v_1} & \cdots & 0 \\ \vdots & \ddots & \vdots \\ 0 & \cdots & W_2^{v_{|V_d|}} \end{bmatrix} \sigma \left( \begin{bmatrix} W_1^{v_1} & \cdots & 0 \\ \vdots & \ddots & \vdots \\ 0 & \cdots & W_1^{v_{|V_d|}} \end{bmatrix} \begin{bmatrix} \boldsymbol{x}^{v_1} \\ \vdots \\ \boldsymbol{x}^{v_{|V_d|}} \end{bmatrix} + \begin{bmatrix} \boldsymbol{b}_1^{v_1} \\ \vdots \\ \boldsymbol{b}_1^{v_{|V_d|}} \end{bmatrix} \right) \quad (2)$$

where $(W_1^{v_i}, \boldsymbol{b}_1^{v_i}, W_2^{v_i}) = \psi(v_i)$, $i \in \{1, \ldots, |V_d|\}$. Note that various vertices at the same depth could share the same input signals and in turn hidden vertices, thus reducing the input and hidden dimensions of the above two-layer NN. We call the NN obtained via intra-depth parallelization of $G_{\mathcal{F}_{\mathcal{N}}}$ the *unmerged* NN of BN $G_{\mathcal{F}_{\mathcal{N}}}$.

**Inter-depth composition & the Layer-Merged NN.** Let $d \in \{1, \ldots, depth(G_{\mathcal{F}_{\mathcal{N}}}) - 1\}$. Moreover, let $\boldsymbol{\zeta}^{(d)}(\boldsymbol{x}^{(d-1)}) = W_2^{(d)}\sigma(W_1^{(d)}\boldsymbol{x}^{(d-1)} + \boldsymbol{b}_1^{(d)}) = \boldsymbol{x}^{(d)}$ and $\boldsymbol{\zeta}^{(d+1)}(\boldsymbol{x}^{(d)}) = W_2^{(d+1)}\sigma(W_1^{(d+1)}\boldsymbol{x}^{(d)} + \boldsymbol{b}_1^{(d+1)}) = \boldsymbol{x}^{(d+1)}$ be the intra-depth parallelized two-layer NNs (as in Equation 2) for vertices in $V_d$ and $V_{d+1}$, respectively. Then,

$$\boldsymbol{x}^{(d+1)} = (\boldsymbol{\zeta}^{(d+1)} \circ \boldsymbol{\zeta}^{(d)})(\boldsymbol{x}^{(d-1)}) \quad (3)$$

$$= W_2^{(d+1)}\sigma \left( W_1^{(d+1)} \left( W_2^{(d)}\sigma \left( W_1^{(d)}\boldsymbol{x}^{(d-1)} + \boldsymbol{b}_1^{(d)} \right) \right) + \boldsymbol{b}_1^{(d+1)} \right) \quad (4)$$

$$= W_2^{(d+1)}\sigma \left( W'\sigma \left( W_1^{(d)}\boldsymbol{x}^{(d-1)} + \boldsymbol{b}_1^{(d)} \right) + \boldsymbol{b}_1^{(d+1)} \right), \quad (5)$$

where $W' = W_1^{(d+1)}W_2^{(d)}$. That is, since the second layer at depth $d$ does not required a threshold, we can merge it with the first layer at depth $d + 1$. We call the NN obtained via intra-depth parallelization and inter-depth composition of $G_{\mathcal{F}_{\mathcal{N}}}$ the *layer-merged* NN of BN $G_{\mathcal{F}_{\mathcal{N}}}$.

*Remark* 2.1 (Implications for depth). Let $d = depth(G_{\mathcal{F}_{\mathcal{N}}})$. Then the depth of the *unmerged* NN is $2d$ (non-PI vertices in $V_1, \ldots, V_d$ have a layer of hidden neurons added before them), and the depth of the *layer-merged* NN is $d + 1$ (the second layer of non-PI and *non-PO* vertices in $V_1, \ldots, V_{d-1}$ is merged with the first layer of vertices at the next depth).

## 3 METHODS

In this section we set out to achieve the two goals stated in the introduction: (1) optimizing the NN size and (2) doing so efficiently. We begin by considering the problem of optimizing NN size while maintaining functional equivalence. The SOTA mapping procedure reviewed in § 2.2 constructs a NN representation of a BN via a composition of sub-NNs, each built from the MP representation of a BF. We propose to optimize the size of the NN by treating the sub-NNs as optimization subproblems. In § 3.1, we identify sub-NN optimization as a constrained minimization problem, and propose a convex technique to solve it.

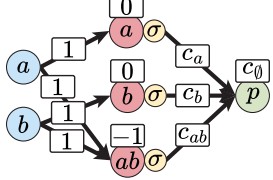

Figure 2: NN representation of MP $p(a, b) = c_\emptyset + c_a a + c_b b + c_{ab} ab$. Weights and biases are shown above connections and neurons.

Next, we turn our attention to accelerating the NN optimization process in aggregate. Since we pose NN optimization as minimizing various sub-NNs, we seek to exploit similarities among the subproblems that must be solved. We develop this idea in § 3.2, showing that by classifying the subproblems into classes, the solution for a representative of the class can be used to accelerate the acquisition of the solutions for all other members of the class.

Finally, in § 3.3, we piece these insights together to form architecture-aware optimization algorithms for the problem of NN-based technology mapping. In particular, we consider how to apply the proposed optimization techniques to the two kinds of NNs that result from the SOTA mapping procedure reviewed in § 2.2: *unmerged* and *layer-merged* NNs. The essential notions critical for describing the main results are presented here, while a comprehensive treatment is provided in the appendix.

### 3.1 Optimizing Neural Network Representations of Boolean Functions

We begin by establishing a relationship between MPs and the size of their two-layer NN representation. Figure 2 depicts this relationship for $k = 2$. We see that the number of hidden neurons in the NN corresponds to the number of non-constant monomials in the MP, and the number of (layer one) connections corresponds to the sum of non-constant monomial degrees. To express this relationship in terms of MP vectors, we introduce the following definition and lemma.

**Definition 3.1** (Uniform & degree criterions). Let $k \in \mathbb{N}_0$ and $c_k^{uni}, c_k^{deg} \in \mathbb{Z}^{2^k}$ have specification $(c_k^{uni})_i = 1$ if $i > 0$ and 0 otherwise, and $c_k^{deg} = \begin{bmatrix} c_{k-1}^{deg}, & c_{k-1}^{deg} + \mathbf{1}_{k-1} \end{bmatrix}^\mathsf{T}$ if $k > 0$ and $[0]$ otherwise, where $\mathbf{1}_{k-1} \in \mathbb{B}^{2^{k-1}}$ denotes the all ones vector. E.g., $c_2^{uni} = [0, 1, 1, 1]^\mathsf{T}$ and $c_2^{deg} = [0, 1, 1, 2]^\mathsf{T}$.

**Lemma 3.1.** *Let $p \in \mathbb{Z}^{2^k}$ be a MP vector, $c \in \mathbb{Z}^{2^k}$, and $C : \mathbb{Z}^{2^k} \to \mathbb{N}_0$ be a cost function with specification $C(p) = \|p\|_{0,c}$, where $\|\cdot\|_{0,c}$ denotes the zero-norm weighted by vector $c$. Then, (1) $C(p) = \|p\|_{0,c_k^{uni}}$ is the number of non-constant monomials in the MP $p$, and (2) $C(p) = \|p\|_{0,c_k^{deg}}$ is the sum of non-constant monomial degrees in the MP $p$.*

Following Lemma 3.1, we can now formally restate our observation of the relationship between MPs and the size of their two-layer NN representation: $\|p\|_{0,c_k^{uni}}$ and $\|p\|_{0,c_k^{deg}}$ are the number of neurons and (layer one) connections in the two-layer NN representation of MP $p$. Optimizing the two-layer NN can now be viewed as finding a polynomial with, for example, fewer monomials. However, at present, removing monomials from a MP will change the BF it represents, and we would lose functional equivalence. Since the BN to NN mapping procedure results in a $\sigma$ (Heaviside) threshold NN, the key insight is to consider MPs equivalent under $\sigma$ threshold.

**Definition 3.2** (The $\sim_\sigma$ equivalence relation). Let $p, q \in \mathbb{Z}^{2^k}$ be MPs. We define $\sim_\sigma : \mathbb{Z}^{2^k} \times \mathbb{Z}^{2^k} \to \mathbb{B}$ as the relation $p \sim_\sigma q \iff \forall x \in \mathbb{B}^k \big( (\sigma \circ p)(x) = (\sigma \circ q)(x) \big)$. Since $\sim_\sigma$ is an equivalence relation (see Appendix C), we denote the $\sigma$ equivalence class of MP $p$ as $[p]_\sigma = \{ q \in \mathbb{Z}^{2^k} \mid p \sim_\sigma q \}$.

**Definition 3.3** (Minimal multilinear polynomial representation of a BF). Let $f : \mathbb{B}^k \to \mathbb{B}$ be a BF with MP $p \in \mathbb{Z}^{2^k}$, and $C : \mathbb{Z}^{2^k} \to \mathbb{N}_0$ be a cost function. A minimal multilinear polynomial (MMP) representation of $f$ with respect to $C$ is a MP $p' \in \mathbb{Z}^{2^k}$ where

$$p' \in \arg\min_{q \in [p]_\sigma} C(q). \tag{6}$$

E.g., $p'(a, b) = a + b$ is an MMP of the MP $p(a, b) = a + b - ab$ (logical OR) with respect to the number of monomials, as they are equivalent under $\sigma$ threshold. Notice, that by removing $-ab$, the resulting two-layer NN has 1 fewer neuron and 3 fewer connections (see Figure 2). However, to maintain functional equivalence, the two-layer NN requires a $\sigma$ threshold on its output neuron.

Lemma 3.1 and Definition 3.3 suggest that finding the MMP of an arbitrary BF requires minimizing an objective with $\ell_0$-norm. These problems are non-convex and NP-hard (Feng et al., 2013), so it is not possible to solve them with conventional convex optimization methods. However, by relaxing the norm to $\ell_1$, the cost function becomes convex and of the form $\|q\|_{1,c} = \|c \odot q\|_1$.[2] Although the solution to the relaxed problem may differ from the actual minimum, both aim for sparsity (Feng et al., 2013), which results in compressed NNs. Next, we will show that $q \in [p]_\sigma$ can be defined as a linear constraint, making the optimization problem convex.

**Theorem 3.1** (Linear constraints determine a $\sigma$ equivalence class). *Let $p \in \mathbb{Z}^{2^k}$ be an MP, $b = \mathbf{1}_k - 2\sigma(S_k p) \in \mathbb{Z}^{2^k}$, and $A = 2(\text{diag}(b)S_k) \in \mathbb{Z}^{2^k \times 2^k}$, where $S_k \in \mathbb{B}^{2^k \times 2^k}$ is the $k$-th Sierpiński*

---

[2] $\odot$ denotes the Hadamard product.

*matrix,[3] $\mathbf{1}_k \in \mathbb{B}^{2^k}$ is the all ones vector, and $\sigma$ is the Heaviside step function. Then, for MP $\boldsymbol{q} \in \mathbb{Z}^{2^k}$,*

$$q \in [p]_\sigma \iff Aq \leq b. \tag{7}$$

Following Theorem 3.1, we have all the components necessary to define an optimization problem for obtaining an MMP representation of a BF and in turn its compressed two-layer NN.

**Corollary 3.1** (Obtaining an MMP representation of a BF). *If $\boldsymbol{p} \in \mathbb{Z}^{2^k}$ is the MP of a BF $f : \mathbb{B}^k \to \mathbb{B}$, and $C : \mathbb{Z}^{2^k} \to \mathbb{N}_0$ has specification $C(\boldsymbol{q}) = \|\boldsymbol{c} \odot \boldsymbol{q}\|_1$, where $\boldsymbol{c} \in \mathbb{Z}^{2^k}$, then the set of MMP representations of $f$ with respect to $C$ equals the solution set of the following integer linear program:*

$$\underset{\boldsymbol{q} \in \mathbb{Z}^{2^k}}{\text{minimize}} \quad C(\boldsymbol{q}), \qquad \text{subject to} \quad A\boldsymbol{q} \leq \boldsymbol{b}. \tag{8}$$

Corollary 3.1 shows that indeed we can find an MMP of an arbitrary BF. While relaxing the optimization objective to an $\ell_1$-norm provides us with convexity, since the variables are integers the problem is still NP-hard (Karp, 1972). We conclude by connecting the new representation with the technology mapping sequence discussed in § 2.2.

**MMP-based technology mapping.** Following MP-based technology mapping, each MP vector $\boldsymbol{p} \in \mathbb{Z}^{2^k}$, $k \leq K$, in BN $G_{\mathcal{F}_{\boldsymbol{p}}}$ is mapped to a functionally equivalent MMP $\boldsymbol{p}' \in [\boldsymbol{p}]_\sigma$ by solving an integer linear program of the form presented in Corollary 3.1. The resulting function set is $\mathcal{F}_{\boldsymbol{p}'} = \cup_{k \leq K} \mathbb{Z}^{2^k}$. We denote this conversion algorithmically as mpToMMP.

## 3.2 OBJECTIVE-AWARE OPTIMIZATION USING SHARED REPRESENTATIONS

In § 2.2, we saw that a NN that is functionally equivalent to a BN can be constructed by computing the MP of each BF (ttToMP) in the BN. In § 3.1, we saw that MMPs are the solution to lossless optimization of each MP, resulting in a compressed two-layer (sub-)NN (mpToMMP). At first glance, one may think we need to solve all these subproblems independently. However, we can improve such an algorithm by *caching* the solution to each subproblem. Alternatively, we can classify the subproblems in terms of shared representations, and find solutions to their class as a whole. One option for this classification problem is to use Negation-Permutation-Negation (NPN) classification of BFs (Sasao & Butler, 2022). We introduce NPN classification with the following definitions.

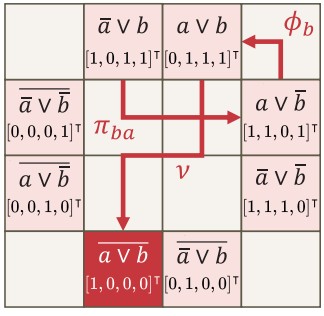

Figure 3: BFs $\boldsymbol{f} \in \mathbb{B}^{2^2}$ in the same NPN equivalence class. Arrows depict NPN transformations, and the NPN canonical form (dark red) is defined to be the member with minimum truth table.

**Definition 3.4** (NPN transformation). An NPN transformation $\tau = \tau_\nu \circ \tau_\pi \circ \tau_\phi$ of a BF or MP is the composition of an input phase assignment $\tau_\phi$ (whether variable $x_i$ is to be negated), an input permutation $\tau_\pi$, and an output polarity assignment $\tau_\nu$ (whether the output is to be negated).

**Definition 3.5** (NPN equivalence). BFs $\boldsymbol{f}, \boldsymbol{g} \in \mathbb{B}^{2^k}$ are NPN equivalent, denoted $\boldsymbol{f} \equiv_{\text{NPN}} \boldsymbol{g}$, if there exists an NPN transformation $\tau$ such that $\tau(\boldsymbol{f}) = \boldsymbol{g}$. The NPN equivalence class of $\boldsymbol{f}$ is denoted $[\boldsymbol{f}]_{\text{NPN}} = \{\boldsymbol{g} \in \mathbb{B}^{2^k} \mid \boldsymbol{f} \equiv_{\text{NPN}} \boldsymbol{g}\}$.

*Remark* 3.1. There are $2^{k+1}k!$ NPN transformations for $k$-input BFs, as there are $2^k$ possible input phase assignments, $k!$ possible input permutations, and 2 possible output polarity assignments. Hence, an NPN equivalence class contains at most $2^{k+1}k!$ BFs.

**Definition 3.6** (NPN canonical form). The NPN canonical form of BF $\boldsymbol{f} \in \mathbb{B}^{2^k}$, denoted $\kappa(\boldsymbol{f})$, is a mapping $\kappa : \mathbb{B}^{2^k} \to \mathbb{B}^{2^k}$ that satisfies the following conditions: (1) $\kappa(\boldsymbol{f}) \in [\boldsymbol{f}]_{\text{NPN}}$, and (2) $\forall \boldsymbol{g} \in [\boldsymbol{f}]_{\text{NPN}}, \kappa(\boldsymbol{g}) = \kappa(\boldsymbol{f})$ (i.e., the canonical form is a unique representative of the class). We denote $\kappa(\boldsymbol{f})$ as $\kappa_{\boldsymbol{f}} \in \mathbb{B}^{2^k}$, and the MP of $\kappa_{\boldsymbol{f}}$ as $\kappa_{\boldsymbol{p}} \in \mathbb{Z}^{2^k}$.

---

[3] $S_k = \begin{bmatrix} S_{k-1}, & O_{k-1} \\ S_{k-1}, & S_{k-1} \end{bmatrix}$ if $k > 0$, and $[1]$ otherwise, where $O_{k-1}$ denotes the all zeros matrix.

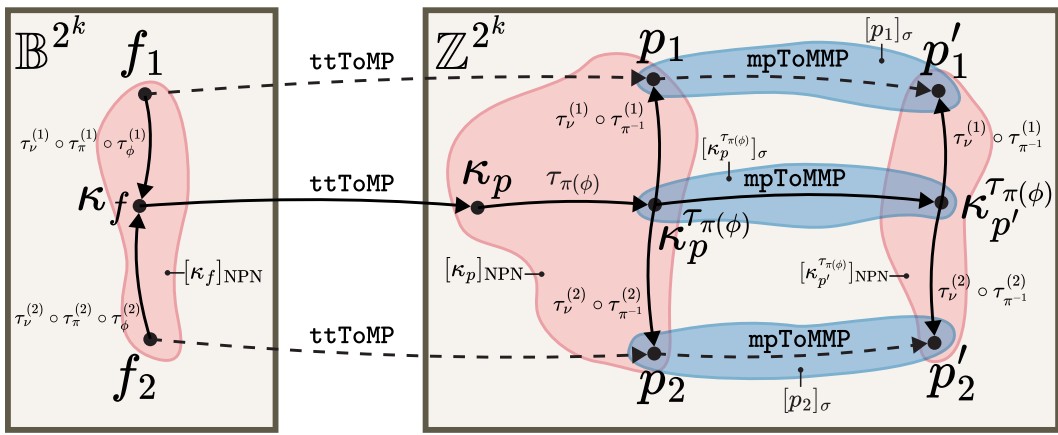

Figure 4: Accelerating the computation of multilinear polynomial (MP) and minimal MP (MMP) representations of Boolean functions (BFs) with Negation-Permutation-Negation (NPN) transformations. Given distinct BFs $\boldsymbol{f}_1, \boldsymbol{f}_2 \in \mathbb{B}^{2^k}$ (as truth table vectors), the dashed arrows indicate that MPs $\boldsymbol{p}_1, \boldsymbol{p}_2 \in \mathbb{Z}^{2^k}$ and MMPs $\boldsymbol{p}'_1, \boldsymbol{p}'_2 \in \mathbb{Z}^{2^k}$ can be obtained via two calls to ttToMP (truth table to MP) followed by two calls to mpToMMP (MP to MMP). If (i) the two functions share NPN canonical form $(\tau_\nu^{(1)} \circ \tau_\pi^{(1)} \circ \tau_\phi^{(1)})(\boldsymbol{f}_1) = (\tau_\nu^{(2)} \circ \tau_\pi^{(2)} \circ \tau_\phi^{(2)})(\boldsymbol{f}_2) = \kappa_{\boldsymbol{f}} \in \mathbb{B}^{2^k}$, (ii) equality $\pi^{(1)}(\phi^{(1)}) = \pi^{(2)}(\phi^{(2)})$ holds, and (iii) the MMP criterion is invariant under input permutation and output negation, then the solid arrows indicate that by utilizing NPN transformations MP and MMP representations of $\boldsymbol{f}_1$ and $\boldsymbol{f}_2$ can be obtained via a single call to ttToMP and mpToMMP. In fact, since the number of functions in each NPN equivalence class is at most $2^{k+1}k!$, if all MPs and MMPs are required to be computed for a class, this process can lead to $2^{k+1}k! - 1$ fewer calls to ttToMP and $2k! - 1$ fewer calls to mpToMMP (at the cost of computing the NPN canonical form and transformations). The red and blue shaded regions depict NPN and $\sigma$ equivalence classes, respectively (where two MPs are $\sigma$-equivalent if their outputs over $\boldsymbol{x} \in \mathbb{B}^k$ are equivalent under threshold).

Figure 3 depicts one of the NPN equivalence classes for $k = 2$. NPN classification provide us with a way to partition the set of BFs in $G_{\mathcal{F}_{\boldsymbol{f}}}$ into equivalence classes, where each function in a class is related by an NPN transformation. We can now investigate how to define solutions to the subproblems of obtaining MPs and MMPs in terms of NPN classes as a whole. The central idea is to transform functions in the same equivalence class into a designated canonical form, perform ttToMP and mpToMMP on the canonical form, and then use inverse transformations to convert the solutions for the canonical form into solutions for all members of the class. The first question we must address is whether the MMP solutions we obtain for members of the class are $\sim_\sigma$ equivalent to their MPs.

**Theorem 3.2** (NPN transformations between $\sigma$ equivalence classes). *Let $\boldsymbol{p}, \boldsymbol{q} \in \mathbb{Z}^{2^k}$ be MPs of BFs such that $\boldsymbol{p} = \tau(\boldsymbol{q})$, where $\tau$ is an NPN transformation and $\boldsymbol{q}' \in [\boldsymbol{q}]_\sigma$. Then $\tau(\boldsymbol{q}') \in [\boldsymbol{p}]_\sigma$.*

We introduce a running example. Let $\kappa_{\boldsymbol{p}} \in \mathbb{Z}^{2^k}$ be the NPN canonical form of MP $\boldsymbol{p} \in \mathbb{Z}^{2^k}$, meaning there exists an NPN transformation $\tau = (\tau_\nu \circ \tau_\pi \circ \tau_\phi)$ such that $\tau(\boldsymbol{p}) = \kappa_{\boldsymbol{p}}$. Now let $\kappa_{\boldsymbol{p}'}$ be an MMP of $\kappa_{\boldsymbol{p}}$ w.r.t. some criterion $\boldsymbol{c} \in \mathbb{Z}^{2^k}$. Since $\tau^{-1}(\kappa_{\boldsymbol{p}}) = \boldsymbol{p}$, Theorem 3.2 ensures us that $\tau^{-1}(\kappa_{\boldsymbol{p}'}) \in [\boldsymbol{p}]_\sigma$. Hence, we are one step closer to converting the MMP solution of the canonical form into an MMP solution for a member of the class. The final question is whether $\tau^{-1}(\kappa_{\boldsymbol{p}'})$ is a minimum w.r.t. the cost function defined by criterion $\boldsymbol{c}$. We introduce the following notions of invariance to answer this.

**Definition 3.7** (PN-invariant criterion). Let $\boldsymbol{p} \in \mathbb{Z}^{2^k}$ be an MP, $\tau_\pi$ be an input permutation, and $\tau_\nu$ be an output polarity assignment. A criterion vector $\boldsymbol{c} \in \mathbb{Z}^{2^k}$ is PN-invariant if

$$\|\boldsymbol{c} \odot \boldsymbol{p}\|_1 = \|\boldsymbol{c} \odot (\tau_\nu \circ \tau_\pi)(\boldsymbol{p})\|_1. \tag{9}$$

**Lemma 3.2** (Uniform & degree criterions are PN-invariant). *$\boldsymbol{c}_k^{uni}$ and $\boldsymbol{c}_k^{deg}$ are PN-invariant.*

If a criterion is PN-invariant, it means that the MMP cost function of MP $\boldsymbol{p}$ will not change when we apply input permutation and output negation transformations on $\boldsymbol{p}$. However, it says nothing about

input negation transformations $\tau_\phi$. Consequently, if $\kappa_{\boldsymbol{p}'}$ is an MMP w.r.t. either the degree or uniform criterion, we cannot say for certain that $\tau^{-1}(\kappa_{\boldsymbol{p}'}) = (\tau_\nu \circ \tau_\pi \circ \tau_\phi)^{-1}(\kappa_{\boldsymbol{p}'})$ is an MMP of $\boldsymbol{p}$, since $\tau_\phi^{-1}$ is being applied. This suggests we should apply the inverse phase assignment $\tau_\phi^{-1}$ *before* finding an MMP. This idea is captured in the following two theorems.

**Theorem 3.3** (Inverse NPN transformation). *Let $\tau = \tau_\nu \circ \tau_\pi \circ \tau_\phi$ be an NPN transformation. Then $\tau^{-1} = \tau_\nu \circ \tau_{\pi^{-1}} \circ \tau_{\pi(\phi)}$. We call $\tau_{\pi(\phi)}$ the permuted phase assignment.*

**Theorem 3.4** (Computing MMPs with NPN transformations under PN-invariant criterions). *Let $\boldsymbol{c} \in \mathbb{Z}^{2^k}$ be a PN-invariant criterion, $\boldsymbol{p}, \boldsymbol{q} \in \mathbb{Z}^{2^k}$ be MPs of BFs such that $\boldsymbol{p} = (\tau_\nu \circ \tau_\pi)(\boldsymbol{q})$, and $\boldsymbol{q}' \in [\boldsymbol{q}]_\sigma$ be an MMP with respect to criterion $\boldsymbol{c}$. Then $\boldsymbol{p}' = (\tau_\nu \circ \tau_\pi)(\boldsymbol{q}') \in [\boldsymbol{p}]_\sigma$ is an MMP with respect to criterion $\boldsymbol{c}$.*

Using Theorem 3.3, we can firstly apply the permuted phase assignment to the canonical form $\tau_{\pi(\phi)}(\kappa_{\boldsymbol{p}}) = \kappa_{\boldsymbol{p}}^{\tau_{\pi(\phi)}}$ and then obtain an MMP of the resulting MP w.r.t. either the degree or uniform criterion, which we denote $\kappa_{\boldsymbol{p}'}^{\tau_{\pi(\phi)}}$. Now, since all the conditions of Theorem 3.4 are satisfied, computing $(\tau_\nu \circ \tau_{\pi^{-1}})(\kappa_{\boldsymbol{p}'}^{\tau_{\pi(\phi)}})$ results in an MMP for $\boldsymbol{p}$ (see Figure 4). Following these ideas, we propose an optimization-objective-aware NPN classification algorithm for MP- and MMP-based technology mapping. The complete algorithm, a formal analysis of its advantages over unique function caching, and the time complexity of computing NPN canonical forms and transformations are provided in Appendix D.4.

## 3.3 ARCHITECTURE-AWARE LOSSLESS OPTIMIZATION

The previous section provides us with a technique for accelerating the acquisition of all MPs and MMPs of BFs in a BN. We now consider how to use these two representations to optimize the two kinds of NN architectures that result from the SOTA mapping procedure reviewed in § 2.2: *unmerged* and *layer-merged* NNs. We begin by connecting this goal to the technology mapping discussed in § 2.2.

**NN-based technology mapping from MP and MMP BNs.** Let $G_\mathcal{F} = (V, E)$ be a BN, $V_{\text{PI}} \subseteq V$ denote the set of PIs, and $\psi_\mathcal{F}^{\boldsymbol{p}}, \psi_\mathcal{F}^{\boldsymbol{p}'} : V \to \mathcal{F}$, $\mathcal{F} = \cup_{k=0}^K \mathbb{Z}^{2^k}$, be the labeling functions following MP- and MMP-based technology mapping, respectively. Moreover, let $\mathcal{M} \subseteq V \setminus V_{\text{PI}}$ be a subset of vertices selected to be represented as MMPs. Then, if $v \notin \mathcal{M}$, its resulting two-layer NN is of the form $\zeta(\boldsymbol{x}) = W_2\sigma(W_1\boldsymbol{x} + \boldsymbol{b}_1)$. If $v \in \mathcal{M}$, its resulting two-layer NN is of the form $\zeta'(\boldsymbol{x}) = \sigma(W_2\sigma(W_1\boldsymbol{x} + \boldsymbol{b}_1))$. We consider three NN optimization algorithms for selecting $\mathcal{M}$:

- `optNone`: no vertices in the BN are represented as MMPs ($\mathcal{M} = \emptyset$). This results in no compression and corresponds to the SOTA NN-based technology mapping solution of Gavier et al. (2023).
- `optAll`: all vertices in the BN are represented as MMPs ($\mathcal{M} = V \setminus V_{\text{PI}}$).
- `optMaintainDepth`: a subset of vertices are selected to be represented as MMPs such that the depth of the resulting *layer-merged* NN does not increase ($\mathcal{M} \subseteq V \setminus V_{\text{PI}}$).

Applications that seek maximal compression would choose `optAll`, whereas applications that require latency (depth) constraints would choose `optMaintainDepth`. In the remainder of this section, we develop an algorithm for `optMaintainDepth`. Suppose $v \in V \setminus V_{\text{PI}}$ is a non-PO vertex on a longest path in $G_\mathcal{F}$. Then following inter-depth composition, $v$ will be on a longest path in the resulting *layer-merged* NN. Hence, if $v$ were to be represented as an MMP, an additional layer would be required to compute its output due to the addition of a threshold on its output neuron; pushing all of $v$'s descendants forward, and thus increasing the depth of the *layer-merged* NN. (Note we require $v$ to be a non-PO vertex in this argument since POs do not have any successor to merge with, and hence can always be represented as MMPs without increasing depth). In contrast, if $v$ were not on a longest path in $G_\mathcal{F}$, then the additional layer to compute $v$ would not cause the depth of the NN to increase. The following definition and lemma capture this insight.

**Definition 3.8** (Leeway). The leeway of vertex $v \in V$ in BN $G_\mathcal{F} = (V, E)$ is defined as

$$leeway(v) = depth(G_\mathcal{F}) - depth(v) - lpl(v) \in \mathbb{N}_0, \tag{10}$$

where $lpl(v)$ denotes the length of the longest path from $v$ to any PO in $G_\mathcal{F}$.

**Lemma 3.3** (Zero leeway indicates membership in longest path). *Let $G_\mathcal{F} = (V, E)$ be a BN and $v \in V$ be a vertex. Then $leeway(v) = 0 \iff v$ is on a longest path in $G_\mathcal{F}$.*

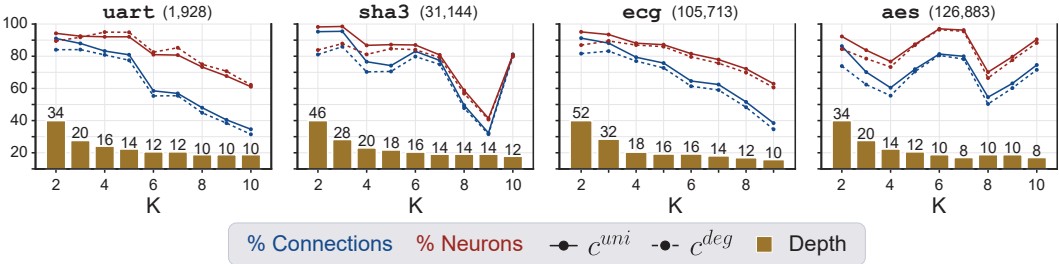

(a) Reduction in connections (blue) and neurons (red) relative to the *unmerged* NN by optimizing all sub-NNs with the proposed `optAll` algorithm. Both criterions achieve significant size reductions.

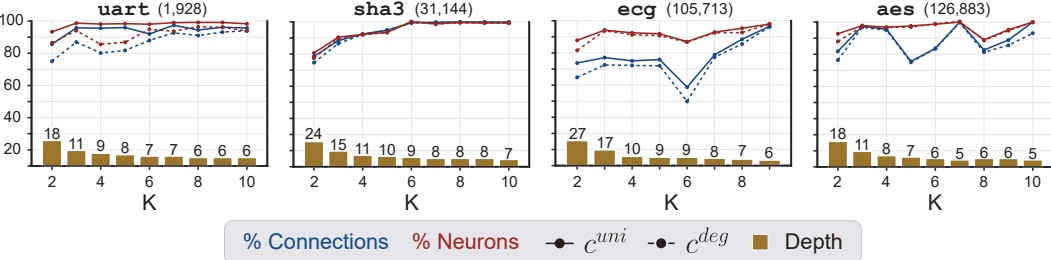

(b) Reduction in connections and neurons relative to the *layer-merged* NN by optimizing a subset of sub-NNs selected by the proposed `optMaintainDepth` algorithm. The size of the *layer-merged* NN is able to be reduced without increasing its depth. The degree criterion leads to larger reduction.

Figure 5: Neural network (NN) lossless optimization with respect to the uniform ($c^{uni}$) and degree ($c^{deg}$) criterions for minimal multilinear polynomials (MMPs) (see Definition 3.1). We find that the size (both in terms of connections and neurons) of *unmerged* (a) and *layer-merged* (b) NNs can be reduced by optimizing the sub-NN representations via MMPs. NN depths are shown in gold. Circuit sizes are shown in parentheses in terms of the number of logic gates.

Since $G_{\mathcal{F}} = (V, E)$ is a DAG, computing the leeway for all vertices can be done in time $O(V + E)$. This lets us propose a $O(V(V + E))$ time algorithm for `optMaintainDepth` that repeatedly selects vertices for $\mathcal{M}$ that have positive leeway (i.e., are not on a longest path). Each time a vertex is selected to be represented as an MMP, the algorithm augments the graph by adding a predecessor hidden vertex to simulate the increase in path length for all paths containing this vertex. This ensures the depth of the optimized NN is the same as the *layer-merged* NN. In Appendix E we present the complete algorithm and illustrate it with a figure.

## 4 RESULTS

We evaluate the proposed architecture- and objective-aware optimization algorithms on four BNs derived from digital circuits: `uart`, `sha3`, `ecg`, and `aes` (see Appendix F for details). In Appendix G.2, we report results on additional BNs to provide a comprehensive analysis. Experiments were conducted on an Intel(R) Xeon(R) Gold 6230R CPU @ 2.10GHz with 196 GB RAM. We use the algorithm proposed by Zhou et al. (2020) to compute NPN transformations and canonical forms. As the mapping procedure is deterministic, we only show statistical significance for timing results.

### 4.1 ARCHITECTURE-AWARE LOSSLESS OPTIMIZATION OF NEURAL NETWORKS

We consider two settings under which to apply the architecture-aware lossless optimization algorithms. The first setting considers optimizing the *unmerged* NN that results from NN-based technology mapping. In this case, we apply the `optAll` optimization algorithm to evaluate, without depth constraint, how many neurons and connections can be removed from the NN while maintaining functional equivalence (see Figure 5a). The second setting considers optimizing the *layer-merged* NN. We apply the `optMaintainDepth` optimization algorithm to see how many neurons and connections can be removed from the NN while maintaining both its depth and functional equivalence (see Figure 5b).

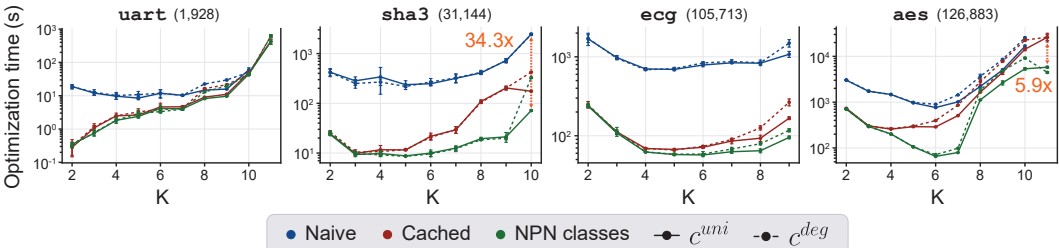

Figure 6: Accelerating neural network (NN) optimization with Negation-Permutation-Negation (NPN) classification. The optimized NN representation of a Boolean network (BN) with vertices associated with $k$-input truth tables, $1 \leq k \leq K$, requires computing the multilinear polynomial (MP) and obtaining a minimal MP (MMP) for each vertex. *Naive* (blue) solves each vertex independently, while *Cached* (red) does so per unique function. *NPN classes* (green) solves using the proposed objective-aware NPN classification algorithm. We report the optimization time for MP- and MMP-based technology mapping w.r.t. the uniform ($c^{uni}$) and degree ($c^{deg}$) criterions. *NPN classes* results in consistent speedups under both MMP criterions, achieving $34.3\times$ for sha3 ($K = 10$, $c^{uni}$) relative to *Naive* and $5.9\times$ for aes ($K = 11$, $c^{deg}$) relative to *Cached*. Lines and error bars denote sample mean and standard deviation over 3 trials. Circuit sizes are shown in parentheses in terms of number of logic gates. *Naive* (uart, $K = 11$) and (aes, $K = 11$) not reported due to compute constraints.

We note that in Figure 5, the SOTA solution of Gavier et al. (2023) corresponds to $100\%$ connections and $100\%$ neurons. In the first setting, optAll significantly optimizes the structure of the NN, reducing the connections and neurons by up to $70\%$ and $60\%$, respectively. In the second, depth-constrained setting, optMaintainDepth reduces the connections and neurons by up to $50\%$ and $20\%$, respectively. We observe that the degree criterion consistently results in fewer connections and neurons than when optimizing with the uniform criterion.

## 4.2 ACCELERATING OPTIMIZATION USING SHARED REPRESENTATIONS

Here we apply the objective-aware NPN classification algorithm from § 3.2 to accelerate NN optimization. Due to lack of prior work, we define two baselines for comparison. *Naive* computes the MP and obtains an MMP per vertex in BN $G_{\mathcal{F}_f}$, whereas *Cached* does so per unique BF. *NPN classes* implements the proposed NPN classification algorithm to compute MPs and obtain MMPs. See Figure 6. We found that when the BN or $K$ is small, *NPN classes* is competitive with *Cached*. However, for larger circuits and $K \geq 4$, *NPN classes* consistently accelerates the optimization time in comparison to both baselines, independent of the MMP criterion used. In particular, the greatest speedup of $34.3\times$ was achieved for sha3 ($K = 10$, $c^{uni}$) relative to *Naive*, and a speedup of $5.9\times$ was achieved for the largest circuit aes ($K = 11$, $c^{deg}$) relative to *Cached* (we analyze this result in Appendix G.3).

For larger BNs, we observed that the optimization time starts high for small $K$, lessens for intermediate $K$, and then increases again for large $K$. When the BN is large and $K$ small, there is a relatively large number of vertices in $G_{\mathcal{F}_f} = (V, E)$. Consequently, the mapping time is dominated by iterating over $V$. In contrast, for large $K$, although $G_{\mathcal{F}_f}$ has fewer vertices, the mapping time increases due to the runtime of ttToMP and mpToMMP. Intermediate $K$ avoids both extremes.

## 5 CONCLUSION

In this paper, we proposed techniques for optimizing the NN representation of a BN. Realizing the NN representation is composed of various sub-NNs, we began by establishing a relationship between MPs and sub-NN size, allowing us to compress sub-NNs by minimizing the MP representation. Next, to accelerate this minimization, we introduced an objective-aware algorithm to exploit the shared structure in optimization solutions among the members of an NPN equivalence class. Finally, we integrated these insights into architecture-aware lossless compression algorithms that allow for NN optimization in both depth constrained and unconstrained settings. Our experimental results demonstrate that our methods lead to significant optimization in NN size, as well as consistent speedups in the optimization process. Broader impacts of our work are further discussed in Appendix I.

ACKNOWLEDGMENTS

The authors would like to thank Arjun Karuvally from the University of Massachusetts Amherst for insightful feedback on an earlier draft of the manuscript. This research was supported in part by the Defense Advanced Research Projects Agency (DARPA) under contract number HR00112190049.

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

# Appendix

## Table of Contents

## A  APPENDIX OUTLINE

This appendix has the following order. In Appendix B, we define the truth table and multilinear polynomial (MP) representations of Boolean functions (BFs), and show how these representations are related via linear transformations. In Appendix C, we define the $\sim_\sigma$ relation between BFs to capture the notion of equivalence under threshold and prove that it is an equivalence relation. We proceed to define a minimal MP (MMP) representation of a BF in terms of $\sigma$ equivalence classes, and by proving that linear constraints determine a $\sigma$ equivalence class, show that such a representation can be obtained by solving an integer linear program. In Appendix D, we define Negation-Permutation-Negation (NPN) transformations and prove how these transformations act on MPs. In Appendix D.1, we prove that the inverse NPN transformation can be computed in a non-standard order, a result relevant for reusing solutions to MMPs. In Appendix D.2, we prove that the functions in the $\sigma$ equivalence classes of two NPN equivalent functions are related via NPN transformations. This result is applied in Appendix D.3, where we define the notion of PN-invariant criterions for MMPs, and prove that MMPs under PN-invariant criterions can be computed from other MMPs using NPN transformations. In Appendix D.4, we use these results to define an objective-aware algorithm based on NPN classification for MP- and MMP-based technology mapping that computes MPs and MMPs via NPN canonical forms in order to minimize the number of calls to computationally expensive procedures. In Appendix E, we define various neural network (NN) compression algorithms that incorporate MMPs into NN-based technology mapping. We present a $O(V(V + E))$ time algorithm that greedily selects vertices in a Boolean network (BN) to be represented as MMPs while ensuring the resulting *layer-merged* NN depth does not increase. Finally, Appendix F details the circuits and automata used in experiments, and Appendix G presents additional results and analyses. Appendix H presents the synthesized neural networks (NNs) for Figure 7, and Appendix I concludes with the broader impacts of our work.

Table 1: Outline of the key definitions, statements and algorithms covered in the appendix. Hyperlinks provide a correspondence between definitions and statements in the main paper and in the appendix.

| Definition/Statement | Main paper | Appendix |
|---|---|---|
| Boolean function (BF) | Definition 2.1 | Definition B.1 |
| Truth table representation of a BF | Definition 2.2 | Definition B.3 |
| Multilinear polynomial (MP) representation of a BF | Definition 2.3 | Definition B.4 |
| The $\sim_\sigma$ relation | Definition 3.2 | Definition C.2 |
| Minimal MP (MMP) representation of a BF | Definition 3.3 | Definition C.4 |
| Uniform criterion | Definition 3.1 | Definition C.5 |
| Degree criterion | Definition 3.1 | Definition C.6 |
| Negation-Permutation-Negation (NPN) transformation | Definition 3.4 | Definition D.1 |
| PN-invariant criterion | Definition 3.7 | Definition D.5 |
| Boolean network (BN) | Definition 2.4 | Definition E.1 |
| Leeway | Definition 3.8 | Definition E.3 |
| The $\sim_\sigma$ relation is an equivalence relation | | Lemma C.1 |
| Linear constraints determine a $\sigma$ equivalence class | Theorem 3.1 | Theorem C.1 |
| Obtaining an MMP representation of a BF | Corollary 3.1 | Corollary C.1 |
| Inverse NPN transformation | Theorem 3.3 | Theorem D.1 |
| NPN transformations between $\sigma$ equivalence classes | Theorem 3.2 | Theorem D.2 |
| Uniform criterion is PN-invariant | Lemma 3.2 | Lemma D.13 |
| Degree criterion is PN-invariant | Lemma 3.2 | Lemma D.14 |
| Computing MMPs with NPN transformations under PN-invariant criterions | Theorem 3.4 | Theorem D.3 |
| MP- and MMP-based technology mapping with NPN classes | | Algorithm 1 |
| Maintain depth neural network optimization algorithm | | Algorithm 2 |

## B  BACKGROUND

*Remark* B.1. We denote the set of binary numbers as $\mathbb{B} = \{0, 1\}$, the set of natural numbers (including 0) as $\mathbb{N}_0 = \{0, 1, 2, \dots\}$, and the set of integers as $\mathbb{Z} = \{\dots, -2, -1, 0, 1, 2, \dots\}$.

**Definition B.1** (Boolean function). A $k$-input, (single-output) Boolean function (BF) is a function $f : \mathbb{B}^k \to \mathbb{B}$, $\boldsymbol{x} \mapsto f(\boldsymbol{x})$, where $\boldsymbol{x} = (x_1, \dots, x_k)$ is a bit vector representing the $k$ input variables $x_i \in \mathbb{B}$, $1 \le i \le k$. If bit vector $\boldsymbol{x}$, when interpreted as binary number $(x_k x_{k-1} \dots x_1)_2$, encodes value $m \in \{0, 1 \dots, 2^k - 1\}$, we write $\boldsymbol{x}_{(m)}$.

*Example* B.1. If $k = 3$, $\boldsymbol{x}_{(4)}$ is the bit vector $(x_1, x_2, x_3) = (0, 0, 1)$ since its interpretation as a binary number $(x_3 x_2 x_1)_2 = (100)_2$ encodes the value 4.

**Definition B.2** (Multilinear polynomial). A degree-$k$ multilinear polynomial (MP) is a polynomial in variables $x_1, \dots, x_k$ where no variable occurs to a power of 2 or higher.

*Example* B.2. If $k = 2$, $p(x_1, x_2) = x_1 + x_2 - x_1 x_2$ is a MP whereas $q(x_1, x_2) = x_1^2 + x_2^2$ is not.

**Definition B.3** (Truth table representation of a BF). The truth table representation is an expression of BF $f : \mathbb{B}^k \to \mathbb{B}$ in the basis of the indicator functions $\mathbb{1}_{\boldsymbol{a}} : \mathbb{B}^k \to \mathbb{B}$, $\boldsymbol{a} \in \mathbb{B}^k$,

$$\mathbb{1}_{\boldsymbol{a}}(\boldsymbol{x}) = \begin{cases} 1, & \text{if } \boldsymbol{x} = \boldsymbol{a}, \\ 0, & \text{otherwise.} \end{cases} \tag{11}$$

The expression of $f$ in this basis is

$$f(\boldsymbol{x}) = \sum_{\boldsymbol{a} \in \mathbb{B}^k} f(\boldsymbol{a}) \mathbb{1}_{\boldsymbol{a}}(\boldsymbol{x}). \tag{12}$$

The truth table vector $\boldsymbol{f} \in \mathbb{B}^{2^k}$ of $f$ is the $2^k$-dimensional vector of coefficients $f(\boldsymbol{a})$ in the above linear combination of indicator functions. Hence,

$$\boldsymbol{f} = \sum_{\boldsymbol{a} \in \mathbb{B}^k} f(\boldsymbol{a}) \mathbf{1}_{\boldsymbol{a}}, \tag{13}$$

where $\mathbf{1}_a \in \mathbb{B}^{2^k}$ denotes the vector with a 1 in the $i$-th coordinate, $0 \le i \le 2^k - 1$, and 0's elsewhere, if $a = a_{(i)}$.

*Example* B.3. Let $f(x_1, x_2) = x_1 \Rightarrow x_2 = \bar{x}_1 \vee x_2$. Then the truth table representation of $f$ is the expression

$$
\begin{aligned}
f(x_1, x_2) &= \sum_{a \in \mathbb{B}^2} f(a) \mathbb{1}_a(x_1, x_2) \\
&= f(0,0) \mathbb{1}_{(0,0)}(x_1, x_2) + f(1,0) \mathbb{1}_{(1,0)}(x_1, x_2) + f(0,1) \mathbb{1}_{(0,1)}(x_1, x_2) + f(1,1) \mathbb{1}_{(1,1)}(x_1, x_2) \\
&= (1) \mathbb{1}_{(0,0)}(x_1, x_2) \quad + (0) \mathbb{1}_{(1,0)}(x_1, x_2) \quad + (1) \mathbb{1}_{(0,1)}(x_1, x_2) \quad + (1) \mathbb{1}_{(1,1)}(x_1, x_2),
\end{aligned}
$$

and the truth table vector $\boldsymbol{f} \in \mathbb{B}^{2^2}$ of $f$ is

$$
\boldsymbol{f} = \sum_{a \in \mathbb{B}^2} f(a) \mathbf{1}_a = f(0,0) \begin{bmatrix} 1 \\ 0 \\ 0 \\ 0 \end{bmatrix} + f(1,0) \begin{bmatrix} 0 \\ 1 \\ 0 \\ 0 \end{bmatrix} + f(0,1) \begin{bmatrix} 0 \\ 0 \\ 1 \\ 0 \end{bmatrix} + f(1,1) \begin{bmatrix} 0 \\ 0 \\ 0 \\ 1 \end{bmatrix} = \begin{bmatrix} f(0,0) \\ f(1,0) \\ f(0,1) \\ f(1,1) \end{bmatrix} = \begin{bmatrix} 1 \\ 0 \\ 1 \\ 1 \end{bmatrix}
$$

**Definition B.4** (Multilinear polynomial representation of a BF). The MP representation is an expression of BF $f : \mathbb{B}^k \to \mathbb{B}$ in the basis of the logical AND functions $\chi_S : \mathbb{Z}^k \to \mathbb{Z}$, $S \in \mathcal{P}(\{1, \ldots, k\})$, where $\mathcal{P}(\cdot)$ denotes the power set,

$$
\chi_S(\boldsymbol{x}) = \chi_S(x_1, \ldots, x_k) = \begin{cases} 1, & \text{if } S = \emptyset, \\ \prod_{i \in S} x_i, & \text{otherwise.} \end{cases} \tag{14}
$$

The expression of $f$ in this basis, which we denote $p : \mathbb{Z}^k \to \mathbb{Z}$, is

$$
p(\boldsymbol{x}) = \sum_{S \in \mathcal{P}(\{1,\ldots,k\})} \hat{f}(S) \chi_S(\boldsymbol{x}), \tag{15}
$$

where $\hat{f}(S) \in \mathbb{Z}$ and $\forall \boldsymbol{x} \in \mathbb{B}^k \big( f(\boldsymbol{x}) = p(\boldsymbol{x}) \big)$. The MP vector $\boldsymbol{p} \in \mathbb{Z}^{2^k}$ of $f$ is the $2^k$-dimensional vector of coefficients $\hat{f}(S)$ in the above linear combination of logical AND functions. Hence,

$$
\boldsymbol{p} = \sum_{S \in \mathcal{P}(\{1,\ldots,k\})} \hat{f}(S) \boldsymbol{\chi}_S, \tag{16}
$$

where $\boldsymbol{\chi}_S \in \mathbb{Z}^{2^k}$ denotes the vector with a 1 in coordinate $\mathcal{I}(S)$, $0 \le \mathcal{I}(S) \le 2^k - 1$, and 0's elsewhere, and $\mathcal{I} : \mathcal{P}(\{1, \ldots, k\}) \to \{0, \ldots 2^k - 1\}$ is a bijection with specification $(\emptyset, 0), (\{1\}, 1), (\{2\}, 2), (\{1,2\}, 3), (\{3\}, 4), \ldots, (\{1, \ldots, k\}, 2^k - 1)$.

*Remark* B.2 (Obtaining coefficients $\hat{f}(S)$ in the MP representation). We briefly mention one method for obtaining the coefficients in the MP representation of BF $f : \mathbb{B}^k \to \mathbb{B}$, which also serves as a construction proof for the MP representation (O'Donnell, 2014). Firstly, we define the indicator MPs $p_a^{\mathbb{1}} : \mathbb{Z}^k \to \mathbb{Z}$, $a \in \mathbb{B}^k$,

$$
p_a^{\mathbb{1}}(\boldsymbol{x}) = \prod_{i=1}^{k} (1 - a_i - x_i + 2a_i x_i).
$$

The indicator MPs have the property that for all $\boldsymbol{x} \in \mathbb{B}^k$, $p_a^{\mathbb{1}}(\boldsymbol{x})$ equals 1 if $\boldsymbol{x} = a$ and 0 otherwise. Let $p : \mathbb{Z}^k \to \mathbb{Z}$ be the MP

$$
p(\boldsymbol{x}) = \sum_{a \in \mathbb{B}^k} f(a) p_a^{\mathbb{1}}(\boldsymbol{x}), \tag{17}
$$

then $\forall \boldsymbol{x} \in \mathbb{B}^k \big( f(\boldsymbol{x}) = p(\boldsymbol{x}) \big)$ (follows from Equation 12). Since the indicator MPs $p_a^{\mathbb{1}}(\boldsymbol{x})$ are multilinear, simplifying $p(\boldsymbol{x})$ will produce a MP with the coefficients $\hat{f}(S) \in \mathbb{Z}$, $S \in \mathcal{P}(\{1, \ldots, k\})$, in the MP representation of $f$. Lemma B.1 generalizes this conversion procedure.

*Example* B.4. Let $f(x_1, x_2) = x_1 \Rightarrow x_2 = \bar{x}_1 \vee x_2$. In Example B.3, we found that the truth table representation of $f$ is the expression

$$
\begin{aligned}
f(x_1, x_2) &= \sum_{a \in \mathbb{B}^2} f(a) \mathbb{1}_a(x_1, x_2) \\
&= (1) \mathbb{1}_{(0,0)}(x_1, x_2) + (0) \mathbb{1}_{(1,0)}(x_1, x_2) + (1) \mathbb{1}_{(0,1)}(x_1, x_2) + (1) \mathbb{1}_{(1,1)}(x_1, x_2).
\end{aligned}
$$

Let $p : \mathbb{Z}^k \to \mathbb{Z}$ be the MP $p(x_1, x_2) = \sum_{\boldsymbol{a} \in \mathbb{B}^2} f(\boldsymbol{a}) p_{\boldsymbol{a}}^{\mathbb{1}}(x_1, x_2)$, then

$$
\begin{aligned}
p(x_1, x_2) &= \sum_{\boldsymbol{a} \in \mathbb{B}^2} f(\boldsymbol{a}) p_{\boldsymbol{a}}^{\mathbb{1}}(x_1, x_2) \\
&= (1) p_{(0,0)}^{\mathbb{1}}(x_1, x_2) && + (0) p_{(1,0)}^{\mathbb{1}}(x_1, x_2) + (1) p_{(0,1)}^{\mathbb{1}}(x_1, x_2) + (1) p_{(1,1)}^{\mathbb{1}}(x_1, x_2) \\
&= (1)(1 - x_1)(1 - x_2) && + (0)(x_1)(1 - x_2) + (1)(1 - x_1)(x_2) + (1)(x_1)(x_2) \\
&= (1 - x_1 - x_2 + x_1 x_2) && + (0) && + (x_2 - x_1 x_2) && + (x_1 x_2) \\
&= (1)(1) && + (-1)(x_1) && + (0)(x_2) && + (1)(x_1 x_2) \\
&= (1) \chi_\emptyset && + (-1) \chi_{\{1\}} && + (0) \chi_{\{2\}} && + (1) \chi_{\{1,2\}} \\
&= \sum_{S \in \mathcal{P}(\{1,2\})} \hat{f}(S) \chi_S(x_1, x_2),
\end{aligned}
$$

and $\sum_{S \in \mathcal{P}(\{1,2\})} \hat{f}(S) \chi_S(x_1, x_2)$ is the MP representation of $f$. The MP vector $\boldsymbol{p} \in \mathbb{Z}^{2^2}$ of $f$ is

$$
\begin{aligned}
\boldsymbol{p} = \sum_{S \in \mathcal{P}(\{1,2\})} \hat{f}(S) \boldsymbol{\chi}_S &= \hat{f}(\emptyset) \begin{bmatrix} 1 \\ 0 \\ 0 \\ 0 \end{bmatrix} + \hat{f}(\{1\}) \begin{bmatrix} 0 \\ 1 \\ 0 \\ 0 \end{bmatrix} + \hat{f}(\{2\}) \begin{bmatrix} 0 \\ 0 \\ 1 \\ 0 \end{bmatrix} + \hat{f}(\{1,2\}) \begin{bmatrix} 0 \\ 0 \\ 0 \\ 1 \end{bmatrix} \\
&= \begin{bmatrix} \hat{f}(\emptyset) \\ \hat{f}(\{1\}) \\ \hat{f}(\{2\}) \\ \hat{f}(\{1,2\}) \end{bmatrix} = \begin{bmatrix} 1 \\ -1 \\ 0 \\ 1 \end{bmatrix}
\end{aligned}
$$

**Definition B.5** (Cofactors of a BF and its MP representation). Let $i \in \{1, \dots, k\}$, and $f : \mathbb{B}^k \to \mathbb{B}$ be a BF with MP representation $p : \mathbb{Z}^k \to \mathbb{Z}$. The positive and negative cofactors of $f$ with respect to $x_i$ are the BFs $f_{x_i}, f_{\bar{x}_i} : \mathbb{B}^{k-1} \to \mathbb{B}$, where

$$
f_{x_i} = f(\dots, x_i = 1, \dots), \qquad\qquad f_{\bar{x}_i} = f(\dots, x_i = 0, \dots). \tag{18}
$$

Similarly, the positive and negative cofactors of $p$ with respect to $x_i$ are the MPs $p_{x_i}, p_{\bar{x}_i} : \mathbb{Z}^{k-1} \to \mathbb{Z}$, where

$$
p_{x_i} = p(\dots, x_i = 1, \dots), \qquad\qquad p_{\bar{x}_i} = p(\dots, x_i = 0, \dots). \tag{19}
$$

Note that $p_{x_i}$ and $p_{\bar{x}_i}$ are the MP representations of $f_{x_i}$ and $f_{\bar{x}_i}$, respectively.

**Definition B.6** (Fourier transformation matrix). Let $k \in \mathbb{N}_0$ and $T_k \in \mathbb{Z}^{2^k \times 2^k}$ be a matrix with specification

$$
T_k = \begin{cases} \begin{bmatrix} 1 \end{bmatrix}, & k = 0, \\ \begin{bmatrix} T_{k-1}, & O_{k-1} \\ -T_{k-1}, & T_{k-1} \end{bmatrix}, & k > 0, \end{cases} \tag{20}
$$

where $O_{k-1} \in \mathbb{B}^{2^{k-1} \times 2^{k-1}}$ denotes the all zeros matrix. We call $T_k$ the $k$-th Fourier transformation matrix.

**Lemma B.1** (Computing the MP vector from the truth table vector). *Let $k \in \mathbb{N}_0$, $f : \mathbb{B}^k \to \mathbb{B}$ be a BF with truth table vector $\boldsymbol{f} \in \mathbb{B}^{2^k}$ and MP vector $\boldsymbol{p} \in \mathbb{Z}^{2^k}$, and $T_k \in \mathbb{Z}^{2^k \times 2^k}$ be the $k$-th Fourier transformation matrix. Then $T_k \boldsymbol{f} = \boldsymbol{p}$.*

*Proof.* We prove this statement by induction on $k \in \mathbb{N}_0$.

Basis ($k = 0$). Let $f : \mathbb{B}^0 \to \mathbb{B}$ be the constant BF $f() = c$, where $c \in \mathbb{B}$. Then $\boldsymbol{f} = [c]$ and $\boldsymbol{p} = [c]$. Thus, as $T_0 = [1]$, $T_0 \boldsymbol{f} = \boldsymbol{p}$.

Inductive step (assume $k-1$). Let $f : \mathbb{B}^k \to \mathbb{B}$ be a BF. Following Boole's expansion theorem,

$$f(\boldsymbol{x}) = \sum_{\boldsymbol{a} \in \mathbb{B}^k} f(\boldsymbol{a}) \mathbb{1}_{\boldsymbol{a}}(\boldsymbol{x}) \tag{21}$$

$$= \sum_{\boldsymbol{a}=(x_1,\dots,x_{k-1},0) \in \mathbb{B}^k} f(\boldsymbol{a}) \mathbb{1}_{\boldsymbol{a}}(\boldsymbol{x}) \tag{22}$$

$$+ \sum_{\boldsymbol{a}=(x_1,\dots,x_{k-1},1) \in \mathbb{B}^k} f(\boldsymbol{a}) \mathbb{1}_{\boldsymbol{a}}(\boldsymbol{x}) \tag{23}$$

$$= \mathbb{1}_{(0)}(x_k) \Big( \sum_{\boldsymbol{a}=(x_1,\dots,x_{k-1}) \in \mathbb{B}^{k-1}} f(\boldsymbol{a},0) \mathbb{1}_{\boldsymbol{a}}(x_1,\dots,x_{k-1}) \Big) \tag{24}$$

$$+ \mathbb{1}_{(1)}(x_k) \Big( \sum_{\boldsymbol{a}=(x_1,\dots,x_{k-1}) \in \mathbb{B}^{k-1}} f(\boldsymbol{a},1) \mathbb{1}_{\boldsymbol{a}}(x_1,\dots,x_{k-1}) \Big) \tag{25}$$

$$= \mathbb{1}_{(0)}(x_k) f_{\bar{x}_k}(x_1,\dots,x_{k-1}) + \mathbb{1}_{(1)}(x_k) f_{x_k}(x_1,\dots,x_{k-1}) \tag{26}$$

Hence, following the discussion in Example B.4, MP $p : \mathbb{Z}^k \to \mathbb{Z}$ with specification

$$p(\boldsymbol{x}) = p_{(0)}^{\mathbb{1}}(x_k) \Big( \sum_{\boldsymbol{a}=(x_1,\dots,x_{k-1}) \in \mathbb{B}^{k-1}} f(\boldsymbol{a},0) p_{\boldsymbol{a}}^{\mathbb{1}}(x_1,\dots,x_{k-1}) \Big) \tag{27}$$

$$+ p_{(1)}^{\mathbb{1}}(x_k) \Big( \sum_{\boldsymbol{a}=(x_1,\dots,x_{k-1}) \in \mathbb{B}^{k-1}} f(\boldsymbol{a},1) p_{\boldsymbol{a}}^{\mathbb{1}}(x_1,\dots,x_{k-1}) \Big) \tag{28}$$

$$= p_{(0)}^{\mathbb{1}}(x_k) p_{x_k}(x_1,\dots,x_{k-1}) + p_{(1)}^{\mathbb{1}}(x_k) p_{\bar{x}_k}(x_1,\dots,x_{k-1}) \tag{29}$$

$$= (1 - x_k) p_{\bar{x}_k}(x_1,\dots,x_{k-1}) + (x_k) p_{x_k}(x_1,\dots,x_{k-1}) \tag{30}$$

$$= p_{\bar{x}_k}(x_1,\dots,x_{k-1}) + \big( p_{x_k}(x_1,\dots,x_{k-1}) - p_{\bar{x}_k}(x_1,\dots,x_{k-1}) \big) x_k \tag{31}$$

has the property that $\forall \boldsymbol{x} \in \mathbb{B}^k \big( f(\boldsymbol{x}) = p(\boldsymbol{x}) \big)$. We next consider the truth table vector $\boldsymbol{f} \in \mathbb{B}^{2^k}$ and MP vector $\boldsymbol{p} \in \mathbb{Z}^{2^k}$ of $f$. From the above analysis, they must have specifications

$$\boldsymbol{f} = \begin{bmatrix} \boldsymbol{f}_{\bar{x}_k} \\ \boldsymbol{f}_{x_k} \end{bmatrix}, \qquad\qquad \boldsymbol{p} = \begin{bmatrix} \boldsymbol{p}_{\bar{x}_k} \\ \boldsymbol{p}_{x_k} - \boldsymbol{p}_{\bar{x}_k} \end{bmatrix}, \tag{32}$$

where $\boldsymbol{f}_{x_k}, \boldsymbol{f}_{\bar{x}_k} \in \mathbb{B}^{2^{k-1}}$ and $\boldsymbol{p}_{x_k}, \boldsymbol{p}_{\bar{x}_k} \in \mathbb{Z}^{2^{k-1}}$. Thus, by the inductive hypothesis for $k-1$, the following sequence of equalities completes the lemma,

$$T_k \boldsymbol{f} = \begin{bmatrix} T_{k-1}, & O_{k-1} \\ -T_{k-1}, & T_{k-1} \end{bmatrix} \begin{bmatrix} \boldsymbol{f}_{\bar{x}_k} \\ \boldsymbol{f}_{x_k} \end{bmatrix} = \begin{bmatrix} \boldsymbol{p}_{\bar{x}_k} \\ \boldsymbol{p}_{x_k} - \boldsymbol{p}_{\bar{x}_k} \end{bmatrix} = \boldsymbol{p}. \tag{33}$$

$\square$

*Example* B.5. We can now find the MP vector from Example B.4 using matrix multiplication. Let $f(x_1, x_2) = x_1 \Rightarrow x_2 = \bar{x}_1 \lor x_2$ be a BF with truth table vector $\boldsymbol{f} \in \mathbb{B}^{2^2}$ and MP vector $\boldsymbol{p} \in \mathbb{Z}^{2^2}$. Following Example B.3,

$$\boldsymbol{f} = \sum_{\boldsymbol{a} \in \mathbb{B}^2} f(\boldsymbol{a}) \mathbb{1}_{\boldsymbol{a}} = \begin{bmatrix} 1 \\ 0 \\ 1 \\ 1 \end{bmatrix}.$$

Following Lemma B.1, we have

$$\boldsymbol{p} = \sum_{S \in \mathcal{P}(\{1,\dots,k\})} \hat{f}(S) \boldsymbol{\chi}_S, = T_2 \boldsymbol{f} = \begin{bmatrix} 1 & 0 & 0 & 0 \\ -1 & 1 & 0 & 0 \\ -1 & 0 & 1 & 0 \\ 1 & -1 & -1 & 1 \end{bmatrix} \begin{bmatrix} 1 \\ 0 \\ 1 \\ 1 \end{bmatrix} = \begin{bmatrix} 1 \\ -1 \\ 0 \\ 1 \end{bmatrix},$$

which was our result for the MP vector of $f$ in Example B.4.

**Definition B.7** (Sierpiński matrix). Let $k \in \mathbb{N}_0$ and $S_k \in \mathbb{B}^{2^k \times 2^k}$ be a matrix with specification

$$S_k = \begin{cases} \begin{bmatrix} 1 \end{bmatrix}, & k = 0, \\ \begin{bmatrix} S_{k-1}, & O_{k-1} \\ S_{k-1}, & S_{k-1} \end{bmatrix}, & k > 0, \end{cases} \tag{34}$$

where $O_{k-1} \in \mathbb{B}^{2^{k-1} \times 2^{k-1}}$ denotes the all zeros matrix. We call $S_k$ the $k$-th Sierpiński matrix.

**Lemma B.2** (The Sierpiński matrix is the inverse of the Fourier transformation matrix). *Let $k \in \mathbb{N}_0$, $T_k \in \mathbb{Z}^{2^k \times 2^k}$ be the $k$-th Fourier transformation matrix, and $S_k \in \mathbb{B}^{2^k \times 2^k}$ be the $k$-th Sierpiński matrix. Then $S_k T_k = I_k$, where $I_k \in \mathbb{B}^{2^k \times 2^k}$ is the identity matrix.*

*Proof.* We prove this statement by induction on $k \in \mathbb{N}_0$.

Basis ($k = 0$). $S_0 T_0 = [1][1] = [1] = I_0$.

Inductive step (assume $S_k T_k = I_k$).

$$S_{k+1} T_{k+1} = \begin{bmatrix} S_k, & O_k \\ S_k, & S_k \end{bmatrix} \begin{bmatrix} T_k, & O_k \\ -T_k, & T_k \end{bmatrix} = \begin{bmatrix} S_k T_k, & O_k \\ O_k, & S_k T_k \end{bmatrix} = \begin{bmatrix} I_k, & O_k \\ O_k, & I_k \end{bmatrix} = I_{k+1}. \tag{35}$$

$\square$

**Corollary B.1** (Computing the truth table vector from the MP vector). *Let $k \in \mathbb{N}_0$, $f : \mathbb{B}^k \to \mathbb{B}$ be a BF with truth table vector $\boldsymbol{f} \in \mathbb{B}^{2^k}$ and MP vector $\boldsymbol{p} \in \mathbb{Z}^{2^k}$, and $S_k \in \mathbb{B}^{2^k \times 2^k}$ be the $k$-th Sierpiński matrix. Then $S_k \boldsymbol{p} = \boldsymbol{f}$.*

*Proof.*

$$S_k \boldsymbol{p} = S_k T_k \boldsymbol{f} = \boldsymbol{f}. \tag{36}$$

where $T_k \in \mathbb{Z}^{2^k \times 2^k}$ is the $k$-th Fourier transformation matrix, and the first equality follows from Lemma B.1. $\square$

*Example* B.6. We can now find the truth table vector from Example B.3 using matrix multiplication. Let $f(x_1, x_2) = x_1 \Rightarrow x_2 = \bar{x}_1 \vee x_2$ be a BF with truth table vector $\boldsymbol{f} \in \mathbb{B}^{2^2}$ and MP vector $\boldsymbol{p} \in \mathbb{Z}^{2^2}$. Following Example B.5,

$$\boldsymbol{p} = \sum_{S \in \mathcal{P}(\{1,\dots,k\})} \hat{f}(S)\boldsymbol{\chi}_S = \begin{bmatrix} 1 \\ -1 \\ 0 \\ 1 \end{bmatrix}.$$

Following Corollary B.1, we have

$$\boldsymbol{f} = \sum_{\boldsymbol{a} \in \mathbb{B}^2} f(\boldsymbol{a})\boldsymbol{1}_{\boldsymbol{a}} = S_2 \boldsymbol{p} = \begin{bmatrix} 1 & 0 & 0 & 0 \\ 1 & 1 & 0 & 0 \\ 1 & 0 & 1 & 0 \\ 1 & 1 & 1 & 1 \end{bmatrix} \begin{bmatrix} 1 \\ -1 \\ 0 \\ 1 \end{bmatrix} = \begin{bmatrix} 1 \\ 0 \\ 1 \\ 1 \end{bmatrix},$$

which was our result for the truth table vector of $f$ in Example B.3.

*Remark* B.3. In the following sections, we refer to the truth table (resp. MP) representation and the truth table (resp. MP) vector of a BF interchangeably. Boldface notation is used for vectors in order to distinguish between the objects.

## C  Minimal Multilinear Polynomial Representation

**Definition C.1** (Heaviside step function). Let $\sigma : \mathbb{Z} \to \mathbb{B}$ be a function with specification

$$\sigma(x) = \begin{cases} 1, & x > 0, \\ 0, & x \le 0. \end{cases} \tag{37}$$

We extend the definition to $\sigma : \mathbb{Z}^{2^k} \to \mathbb{B}^{2^k}$, where $(\sigma(\boldsymbol{x}))_i = \sigma(\boldsymbol{x}_i)$ (i.e., it is applied element-wise).

**Definition C.2** (The $\sim_\sigma$ relation). Let $\boldsymbol{p}, \boldsymbol{q} \in \mathbb{Z}^{2^k}$ be MPs. We define $\sim_\sigma : \mathbb{Z}^{2^k} \times \mathbb{Z}^{2^k} \to \mathbb{B}$ as the relation

$$\boldsymbol{p} \sim_\sigma \boldsymbol{q} \iff \forall \boldsymbol{x} \in \mathbb{B}^k \big( (\sigma \circ p)(\boldsymbol{x}) = (\sigma \circ q)(\boldsymbol{x}) \big). \tag{38}$$

*Example* C.1. Let $f(x_1, x_2) = x_1 \Rightarrow x_2 = \bar{x}_1 \vee x_2$ be a BF with MP representation $p(x_1, x_2) = 1 - x_1 + x_1 x_2$ and MP vector $\boldsymbol{p} = [1, -1, 0, 1]$. Moreover, let $q(x_1, x_2) = 1 - x_1 + x_2$ be a MP with MP vector $\boldsymbol{q} = [1, -1, 1, 0]$. Then $\boldsymbol{p} \sim_\sigma \boldsymbol{q}$, as

$$
\begin{aligned}
p(0,0) &= 1, & q(0,0) &= 1, \\
p(1,0) &= 0, & q(1,0) &= 0, \\
p(0,1) &= 1, & q(0,1) &= 2, \\
p(1,1) &= 1, & q(1,1) &= 1.
\end{aligned}
$$

**Lemma C.1.** $\sim_\sigma$ *is an equivalence relation.*

*Proof.* We show the $\sim_\sigma$ relation is reflexive, symmetric and transitive. Let $\boldsymbol{p}, \boldsymbol{q}, \boldsymbol{r} \in \mathbb{Z}^{2^k}$ be MPs.

1. $\sim_\sigma$ is reflexive.

$$\boldsymbol{p} \sim_\sigma \boldsymbol{p} \equiv \forall \boldsymbol{x} \in \mathbb{B}^k \big( (\sigma \circ p)(\boldsymbol{x}) = (\sigma \circ p)(\boldsymbol{x}) \big) \tag{39}$$

$$\equiv \forall \boldsymbol{x} \in \mathbb{B}^k (\top) \tag{40}$$

$$\equiv \top \tag{41}$$

2. $\sim_\sigma$ is symmetric.

$$\boldsymbol{p} \sim_\sigma \boldsymbol{q} \equiv \forall \boldsymbol{x} \in \mathbb{B}^k \big( (\sigma \circ p)(\boldsymbol{x}) = (\sigma \circ q)(\boldsymbol{x}) \big) \tag{42}$$

$$\equiv \forall \boldsymbol{x} \in \mathbb{B}^k \big( (\sigma \circ q)(\boldsymbol{x}) = (\sigma \circ p)(\boldsymbol{x}) \big) \tag{43}$$

$$\equiv \boldsymbol{q} \sim_\sigma \boldsymbol{p} \tag{44}$$

3. $\sim_\sigma$ is transitive.

$$\boldsymbol{p} \sim_\sigma \boldsymbol{q} \wedge \boldsymbol{q} \sim_\sigma \boldsymbol{r}$$

$$\equiv \forall \boldsymbol{x} \in \mathbb{B}^k \big( (\sigma \circ p)(\boldsymbol{x}) = (\sigma \circ q)(\boldsymbol{x}) \big) \wedge \forall \boldsymbol{x} \in \mathbb{B}^k \big( (\sigma \circ q)(\boldsymbol{x}) = (\sigma \circ r)(\boldsymbol{x}) \big) \tag{45}$$

$$\equiv \forall \boldsymbol{x} \in \mathbb{B}^k \big( \big( (\sigma \circ p)(\boldsymbol{x}) = (\sigma \circ q)(\boldsymbol{x}) \big) \wedge \big( (\sigma \circ q)(\boldsymbol{x}) = (\sigma \circ r)(\boldsymbol{x}) \big) \big) \tag{46}$$

$$\equiv \forall \boldsymbol{x} \in \mathbb{B}^k \big( (\sigma \circ p)(\boldsymbol{x}) = (\sigma \circ r)(\boldsymbol{x}) \big) \tag{47}$$

$$\equiv \boldsymbol{p} \sim_\sigma \boldsymbol{r} \tag{48}$$

$\square$

**Definition C.3** ($\sigma$ equivalence class). The $\sigma$ equivalence class of MP $\boldsymbol{p} \in \mathbb{Z}^{2^k}$, denoted $[\boldsymbol{p}]_\sigma$, is the set $\{\boldsymbol{q} \in \mathbb{Z}^{2^k} \mid \boldsymbol{p} \sim_\sigma \boldsymbol{q}\}$.

**Definition C.4** (Minimal multilinear polynomial representation of a BF). Let $f : \mathbb{B}^k \to \mathbb{B}$ be a BF with MP $\boldsymbol{p} \in \mathbb{Z}^{2^k}$, and $C : \mathbb{Z}^{2^k} \to \mathbb{N}_0$ be a cost function. A minimal multilinear polynomial (MMP) representation of $f$ with respect to $C$ is a MP $\boldsymbol{p}' \in \mathbb{Z}^{2^k}$ where

$$\boldsymbol{p}' \in \operatorname*{arg\,min}_{\boldsymbol{q} \in [\boldsymbol{p}]_\sigma} C(\boldsymbol{q}). \tag{49}$$

*Example* C.2. Let $f(x_1, x_2) = x_1 \lor x_2$ be a BF with MP representation $p(x_1, x_2) = x_1 + x_2 - x_1 x_2$ and MP vector $\boldsymbol{p} = [0, 1, 1, -1]$. Moreover, let $p'(x_1, x_2) = x_1 + x_2$ be a MP with MP vector $\boldsymbol{p}' = [0, 1, 1, 0]$. Then, under the assumption that $C(\boldsymbol{q})$ represents the number of terms in MP $q$,

$$\boldsymbol{p}' \in \underset{\boldsymbol{q} \in [\boldsymbol{p}]_\sigma}{\arg \min} \, C(\boldsymbol{q}), \tag{50}$$

and $p'$ is an MMP representation of $f$ with respect to $C$.

**Definition C.5** (Uniform criterion). Let $k \in \mathbb{N}_0$ and $\boldsymbol{c}_k^{uni} \in \mathbb{B}^{2^k}$ be a vector with specification

$$(\boldsymbol{c}_k^{uni})_i = \begin{cases} 0, & i = 0, \\ 1, & i > 0. \end{cases} \tag{51}$$

*Example* C.3. $\boldsymbol{c}_3^{uni} \in \mathbb{B}^{2^3} = [0, 1, 1, 1, 1, 1, 1, 1]^\mathsf{T}$.

**Definition C.6** (Degree criterion). Let $k \in \mathbb{N}_0$ and $\boldsymbol{c}_k^{deg} \in \mathbb{Z}^{2^k}$ be a vector with specification

$$\boldsymbol{c}_k^{deg} = \begin{cases} [0], & k = 0, \\ \left[ \boldsymbol{c}_{k-1}^{deg}, \quad \boldsymbol{c}_{k-1}^{deg} + \mathbf{1}_{k-1} \right]^\mathsf{T}, & k > 0, \end{cases} \tag{52}$$

where $\mathbf{1}_{k-1} \in \mathbb{B}^{2^{k-1}}$ denotes the all ones vector.

*Example* C.4. $\boldsymbol{c}_3^{deg} \in \mathbb{Z}^{2^3} = [0, 1, 1, 2, 1, 2, 2, 3]^\mathsf{T}$.

**Lemma C.2** (Cost functions for MMPs). *Let $\boldsymbol{q} \in \mathbb{Z}^{2^k}$ be a MP vector, $\boldsymbol{c} \in \mathbb{Z}^{2^k}$, and $C : \mathbb{Z}^{2^k} \to \mathbb{N}_0$ be a cost function with specification $C(\boldsymbol{q}) = \|\boldsymbol{q}\|_{0,\boldsymbol{c}}$, where $\|\cdot\|_{0,\boldsymbol{c}}$ denotes the zero-norm weighted by vector $\boldsymbol{c}$. Then,*

1. *$C(\boldsymbol{q}) = \|\boldsymbol{q}\|_{0,\boldsymbol{c}_k^{uni}}$ is the number of non-constant monomials in the MP $\boldsymbol{q}$.*

2. *$C(\boldsymbol{q}) = \|\boldsymbol{q}\|_{0,\boldsymbol{c}_k^{deg}}$ is the sum of non-constant monomial degrees in the MP $\boldsymbol{q}$.*

*Proof.* Let $\mathcal{I}(S)$ be the bijection specified in Definition B.4 to index the components of the MP vector, and $\mathbb{1}_0 : \mathbb{Z} \to \mathbb{B}$ be an indicator function where $\mathbb{1}_0(x) = 1 \iff x = 0$. Then *1.* follows from the definition of the weighted zero-norm:

$$C(\boldsymbol{q}) = \|\boldsymbol{q}\|_{0,\boldsymbol{c}_k^{uni}} \tag{53}$$

$$= \sum_{S \in \mathcal{P}(\{1,\dots,k\})} (\boldsymbol{c}_k^{uni})_{\mathcal{I}(S)} \left( 1 - \mathbb{1}_0(\hat{f}(S)) \right) \quad \text{(weighted zero-norm)} \tag{54}$$

$$= (\boldsymbol{c}_k^{uni})_{\mathcal{I}(\emptyset)} \left( 1 - \mathbb{1}_0(\hat{f}(\emptyset)) \right) + \sum_{S \in \mathcal{P}(\{1,\dots,k\}) \setminus \{\emptyset\}} (\boldsymbol{c}_k^{uni})_{\mathcal{I}(S)} \left( 1 - \mathbb{1}_0(\hat{f}(S)) \right) \tag{55}$$

$$= \sum_{S \in \mathcal{P}(\{1,\dots,k\}) \setminus \{\emptyset\}} \left( 1 - \mathbb{1}_0(\hat{f}(S)) \right). \tag{56}$$

Similarly, *2.* follows from the same definitions. We prove this statement by induction on $k \in \mathbb{N}_0$.

Basis ($k = 0$). $C(\boldsymbol{q}) = \|\boldsymbol{q}\|_{0,\boldsymbol{c}_0^{deg}} = (\boldsymbol{c}_0^{deg})_{\mathcal{I}(\emptyset)} \left( 1 - \mathbb{1}_0(\hat{f}(\emptyset)) \right) = (\boldsymbol{c}_0^{deg})_0 \left( 1 - \mathbb{1}_0(\hat{f}(\emptyset)) \right) = 0$.

Inductive step (assume $k - 1$). Let $\mathbf{1}_{k-1} \in \mathbb{B}^{2^{k-1}}$ denote the all ones vector. Recall from the proof of Lemma B.1 that $\boldsymbol{q} = [\boldsymbol{q}_{\bar{x}_k}, \boldsymbol{q}_{x_k} - \boldsymbol{q}_{\bar{x}_k}]^\mathsf{T}$, where $\boldsymbol{q}_{x_k} \in \mathbb{Z}^{2^{k-1}}$ and $\boldsymbol{q}_{\bar{x}_k} \in \mathbb{Z}^{2^{k-1}}$ are the MP vectors

of the positive and negative cofactor of $\boldsymbol{q}$ with respect to $x_k$, respectively. For $\boldsymbol{q} \in \mathbb{Z}^{2^k}$ we have,

$$C(\boldsymbol{q}) = \|\boldsymbol{q}\|_{0, \boldsymbol{c}_k^{deg}} \tag{57}$$

$$= \sum_{S \in \mathcal{P}(\{1, \ldots, k\})} (\boldsymbol{c}_k^{deg})_{\mathcal{I}(S)} \left(1 - \mathbb{1}_0(\hat{f}(S))\right) \tag{58}$$

$$= \sum_{S \in \mathcal{P}(\{1, \ldots, k-1\})} (\boldsymbol{c}_{k-1}^{deg})_{\mathcal{I}(S)} \left(1 - \mathbb{1}_0(\hat{f}(S))\right)$$

$$+ \sum_{S \in \mathcal{P}(\{1, \ldots, k-1\})} \left((\boldsymbol{c}_{k-1}^{deg})_{\mathcal{I}(S)} + 1\right) \left(1 - \mathbb{1}_0(\hat{f}(S \cup \{k\}))\right) \tag{59}$$

$$= \|\boldsymbol{q}_{\bar{x}_k}\|_{0, \boldsymbol{c}_{k-1}^{deg}} + \|\boldsymbol{q}_{x_k} - \boldsymbol{q}_{\bar{x}_k}\|_{0, \boldsymbol{c}_{k-1}^{deg}} + \|\boldsymbol{q}_{x_k} - \boldsymbol{q}_{\bar{x}_k}\|_{0, \boldsymbol{1}_{k-1}}. \tag{60}$$

Where, by the inductive hypothesis, the first term corresponds to summing the degree of each monomial in $\boldsymbol{q}$ not containing $x_k$, and the second term corresponds to summing the degree *minus one* of each monomial in $\boldsymbol{q}$ containing $x_k$. The third term corresponds to the sum of monomials in $\boldsymbol{q}$ containing $x_k$. Hence, the second and third term correspond to summing the degree of each monomial in $\boldsymbol{q}$ containing $x_k$. □

**Theorem C.1** (Linear constraints determine a $\sigma$ equivalence class). *Let $\boldsymbol{p} \in \mathbb{Z}^{2^k}$ be an MP, $\boldsymbol{f} = \sigma(S_k \boldsymbol{p}) \in \mathbb{B}^{2^k}$, $\boldsymbol{b} = \boldsymbol{1}_k - 2\boldsymbol{f} \in \mathbb{Z}^{2^k}$, and $A = 2(\mathrm{diag}(\boldsymbol{b}) S_k) \in \mathbb{Z}^{2^k \times 2^k}$, where $S_k \in \mathbb{B}^{2^k \times 2^k}$ is the $k$-th Sierpiński matrix, $\boldsymbol{1}_k \in \mathbb{B}^{2^k}$ denotes the all ones vector, and $\sigma$ is the Heaviside step function. Then, for MP $\boldsymbol{q} \in \mathbb{Z}^{2^k}$,*

$$\boldsymbol{q} \in [\boldsymbol{p}]_\sigma \iff A\boldsymbol{q} \leq \boldsymbol{b}. \tag{61}$$

*Proof.* Let $\boldsymbol{g} = S_k \boldsymbol{q} \in \mathbb{Z}^{2^k}$. We begin by expanding the expression $A\boldsymbol{q} \leq \boldsymbol{b}$:

$$A\boldsymbol{q} \leq \boldsymbol{b}$$

$$\equiv 2(\mathrm{diag}(\boldsymbol{b}) S_k)\boldsymbol{q} \leq \boldsymbol{1}_k - 2\boldsymbol{f} \tag{62}$$

$$\equiv 2 \cdot \mathrm{diag}(\boldsymbol{b}) S_k \boldsymbol{q} \leq \boldsymbol{1}_k - 2\boldsymbol{f} \tag{63}$$

$$\equiv 2 \cdot \mathrm{diag}(\boldsymbol{1}_k - 2\boldsymbol{f})\boldsymbol{g} \leq \boldsymbol{1}_k - 2\boldsymbol{f} \tag{64}$$

$$\equiv 2 \cdot \begin{bmatrix} 1 - 2f(\boldsymbol{a}_{(0)}) & 0 & \cdots & 0 \\ 0 & 1 - 2f(\boldsymbol{a}_{(1)}) & \cdots & 0 \\ \vdots & \vdots & \ddots & \vdots \\ 0 & 0 & \cdots & 1 - 2f(\boldsymbol{a}_{(2^k-1)}) \end{bmatrix} \begin{bmatrix} g(\boldsymbol{a}_{(0)}) \\ g(\boldsymbol{a}_{(1)}) \\ \vdots \\ g(\boldsymbol{a}_{(2^k-1)}) \end{bmatrix} \leq \begin{bmatrix} 1 - 2f(\boldsymbol{a}_{(0)}) \\ 1 - 2f(\boldsymbol{a}_{(1)}) \\ \vdots \\ 1 - 2f(\boldsymbol{a}_{(2^k-1)}) \end{bmatrix} \tag{65}$$

$$\equiv 2 \cdot \begin{bmatrix} (1 - 2f(\boldsymbol{a}_{(0)}))g(\boldsymbol{a}_{(0)}) \\ (1 - 2f(\boldsymbol{a}_{(1)}))g(\boldsymbol{a}_{(1)}) \\ \vdots \\ (1 - 2f(\boldsymbol{a}_{(2^k-1)}))g(\boldsymbol{a}_{(2^k-1)}) \end{bmatrix} \leq \begin{bmatrix} 1 - 2f(\boldsymbol{a}_{(0)}) \\ 1 - 2f(\boldsymbol{a}_{(1)}) \\ \vdots \\ 1 - 2f(\boldsymbol{a}_{(2^k-1)}) \end{bmatrix} \tag{66}$$

$$\equiv \forall i \in \{0, 1, \ldots, 2^k - 1\} \left(2(1 - 2f(\boldsymbol{a}_{(i)}))g(\boldsymbol{a}_{(i)}) \leq 1 - 2f(\boldsymbol{a}_{(i)})\right). \tag{67}$$

We next derive an equivalence for $\boldsymbol{q} \in [\boldsymbol{p}]_\sigma$ in terms of the $k$-th Sierpiński matrix:

$$\boldsymbol{q} \in [\boldsymbol{p}]_\sigma \equiv \boldsymbol{p} \sim_\sigma \boldsymbol{q} \tag{68}$$

$$\equiv \forall \boldsymbol{x} \in \mathbb{B}^k \left((\sigma \circ p)(\boldsymbol{x}) = (\sigma \circ q)(\boldsymbol{x})\right) \tag{69}$$

$$\equiv \forall i \in \{0, 1, \ldots, 2^k - 1\} \left(\sigma((S_k \boldsymbol{p})_i) = \sigma((S_k \boldsymbol{q})_i)\right) \tag{70}$$

$$\equiv \forall i \in \{0, 1, \ldots, 2^k - 1\} \left((\sigma(S_k \boldsymbol{p}))_i = \sigma((S_k \boldsymbol{q})_i)\right) \tag{71}$$

$$\equiv \forall i \in \{0, 1, \ldots, 2^k - 1\} \left(f(\boldsymbol{a}_{(i)}) = \sigma(g(\boldsymbol{a}_{(i)}))\right). \tag{72}$$

We now show both directions.

**1.** ($\boldsymbol{q} \in [\boldsymbol{p}]_\sigma \implies A\boldsymbol{q} \leq \boldsymbol{b}$): If $\boldsymbol{q} \in [\boldsymbol{p}]_\sigma$, then by Equation 72,

$$\forall i \in \{0, 1, \dots, 2^k - 1\}\big(f(\boldsymbol{a}_{(i)}) = \sigma(g(\boldsymbol{a}_{(i)}))\big). \tag{73}$$

Since $f(\boldsymbol{a}_{(i)}) \in \mathbb{B}$, either $f(\boldsymbol{a}_{(i)}) = 0$ or $f(\boldsymbol{a}_{(i)}) = 1$. Taking $f(\boldsymbol{a}_{(i)}) = 0$, we have $\sigma(g(\boldsymbol{a}_{(i)})) = 0$ which implies $g(\boldsymbol{a}_{(i)}) \leq 0$ and that $2g(\boldsymbol{a}_{(i)}) \leq 1$ as $g(\boldsymbol{a}_{(i)}) \in \mathbb{Z}$. Taking $f(\boldsymbol{a}_{(i)}) = 1$, we have $\sigma(g(\boldsymbol{a}_{(i)})) = 1$ which implies $g(\boldsymbol{a}_{(i)}) > 0$ and that $2g(\boldsymbol{a}_{(i)}) \geq 1$ as $g(\boldsymbol{a}_{(i)}) \in \mathbb{Z}$. Consequently,

$$\forall i \in \{0, 1, \dots, 2^k - 1\}\big(f(\boldsymbol{a}_{(i)}) = \sigma(g(\boldsymbol{a}_{(i)}))\big)$$
$$\implies \forall i \in \{0, 1, \dots, 2^k - 1\}\big(2(1 - 2f(\boldsymbol{a}_{(i)}))g(\boldsymbol{a}_{(i)}) \leq 1 - 2f(\boldsymbol{a}_{(i)})\big). \tag{74}$$

By Equation 67 we conclude $A\boldsymbol{q} \leq \boldsymbol{b}$.

**2.** ($A\boldsymbol{q} \leq \boldsymbol{b} \implies \boldsymbol{q} \in [\boldsymbol{p}]_\sigma$): If $A\boldsymbol{q} \leq \boldsymbol{b}$, then by Equation 67,

$$\forall i \in \{0, 1, \dots, 2^k - 1\}\big(2(1 - 2f(\boldsymbol{a}_{(i)}))g(\boldsymbol{a}_{(i)}) \leq 1 - 2f(\boldsymbol{a}_{(i)})\big). \tag{75}$$

Since $f(\boldsymbol{a}_{(i)}) \in \mathbb{B}$, either $f(\boldsymbol{a}_{(i)}) = 0$ or $f(\boldsymbol{a}_{(i)}) = 1$. Taking $f(\boldsymbol{a}_{(i)}) = 0$, we have $2g(\boldsymbol{a}_{(i)}) \leq 1$ which implies $g(\boldsymbol{a}_{(i)}) \leq 0$ as $g(\boldsymbol{a}_{(i)}) \in \mathbb{Z}$, meaning $\sigma(g(\boldsymbol{a}_{(i)})) = 0 = f(\boldsymbol{a}_{(i)})$. Taking $f(\boldsymbol{a}_{(i)}) = 1$, we have $2g(\boldsymbol{a}_{(i)}) \geq 1$ which implies $g(\boldsymbol{a}_{(i)}) \geq 1$ as $g(\boldsymbol{a}_{(i)}) \in \mathbb{Z}$, meaning $\sigma(g(\boldsymbol{a}_{(i)})) = 1 = f(\boldsymbol{a}_{(i)})$. Consequently,

$$\forall i \in \{0, 1, \dots, 2^k - 1\}\big(2(1 - 2f(\boldsymbol{a}_{(i)}))g(\boldsymbol{a}_{(i)}) \leq 1 - 2f(\boldsymbol{a}_{(i)})\big)$$
$$\implies \forall i \in \{0, 1, \dots, 2^k - 1\}\big(f(\boldsymbol{a}_{(i)}) = \sigma(g(\boldsymbol{a}_{(i)}))\big). \tag{76}$$

By Equation 72 we conclude $\boldsymbol{q} \in [\boldsymbol{p}]_\sigma$. $\qquad\square$

*Remark* C.1. Lemma C.2 and Theorem C.1 allow us to formulate the MMP problem as an $\ell_0$-norm optimization problem. These problems are known to be non-convex and NP-hard (Feng et al., 2013), so it is not possible to use conventional convex optimization methods to find optimal solutions. However, as it is common in optimization problems, by relaxing the norm from $\ell_0$ to $\ell_1$, the cost functions specified in Lemma C.2 become $\|\boldsymbol{q}\|_{1,\boldsymbol{c}} = \|\boldsymbol{c} \odot \boldsymbol{q}\|_1$, resulting in a convex optimization problem.

*Example* C.5. Let $\boldsymbol{p} = [0, 1, 1, -1]^\intercal$ be a MP vector, and $C_\ell : \mathbb{Z}^{2^k} \to \mathbb{N}_0$ be a cost function with specification $C_\ell(\boldsymbol{q}) = \|\boldsymbol{q}\|_{\ell, \boldsymbol{c}_k^{uni}}$. Following Theorem C.1, we can compute $\boldsymbol{b} = (\boldsymbol{1}_k - 2\sigma(S_k\boldsymbol{p}))$ and $A = 2(\mathrm{diag}(\boldsymbol{b})S_k)$:

$$\boldsymbol{b} = \begin{bmatrix} 1 \\ -1 \\ -1 \\ -1 \end{bmatrix}, \qquad\qquad A = \begin{bmatrix} 2 & 0 & 0 & 0 \\ -2 & -2 & 0 & 0 \\ -2 & 0 & -2 & 0 \\ -2 & -2 & -2 & -2 \end{bmatrix}.$$

The MP vector that minimizes $C_0(\boldsymbol{q})$ such that $A\boldsymbol{q} \leq \boldsymbol{b}$ is $\boldsymbol{q} = [0, 1, 1, 0]^\intercal$. It also minimizes $C_1(\boldsymbol{q})$.

*Example* C.6. Let $\boldsymbol{p} = [1, -1, 0, 1]^\intercal$ be a MP vector, and $C_\ell : \mathbb{Z}^{2^k} \to \mathbb{N}_0$ be a cost function with specification $C_\ell(\boldsymbol{q}) = \|\boldsymbol{q}\|_{\ell, \boldsymbol{c}_k^{deg}}$. Following Theorem C.1, we can compute $\boldsymbol{b} = (\boldsymbol{1}_k - 2\sigma(S_k\boldsymbol{p}))$ and $A = 2(\mathrm{diag}(\boldsymbol{b})S_k)$:

$$\boldsymbol{b} = \begin{bmatrix} -1 \\ 1 \\ -1 \\ -1 \end{bmatrix}, \qquad\qquad A = \begin{bmatrix} -2 & 0 & 0 & 0 \\ 2 & 2 & 0 & 0 \\ -2 & 0 & -2 & 0 \\ -2 & -2 & -2 & -2 \end{bmatrix}.$$

The MP vector that minimizes $C_0(\boldsymbol{q})$ such that $A\boldsymbol{q} \leq \boldsymbol{b}$ is $\boldsymbol{q} = [1, -1, -1, 0]^\intercal$. It also minimizes $C_1(\boldsymbol{q})$.

**Corollary C.1** (Obtaining an MMP representation of a BF). *Let $\boldsymbol{p} \in \mathbb{Z}^{2^k}$ be the MP of BF $f : \mathbb{B}^k \to \mathbb{B}$, $\boldsymbol{f} = S_k\boldsymbol{p} \in \mathbb{B}^{2^k}$, $\boldsymbol{b} = \boldsymbol{1}_k - 2\boldsymbol{f} \in \mathbb{Z}^{2^k}$, and $A = 2(\mathrm{diag}(\boldsymbol{b})S_k) \in \mathbb{Z}^{2^k \times 2^k}$, where $S_k \in \mathbb{B}^{2^k \times 2^k}$ is the $k$-th Sierpiński matrix, $\boldsymbol{1}_k \in \mathbb{B}^{2^k}$ denotes the all ones vector, and $\sigma$ is the Heaviside step function. Moreover, let $C : \mathbb{Z}^{2^k} \to \mathbb{N}_0$ have specification $C(\boldsymbol{q}) = \|\boldsymbol{c} \odot \boldsymbol{q}\|_1$, where $\boldsymbol{c} \in \mathbb{Z}^{2^k}$. Then the set of*

MMP representations of $f$ with respect to $C$ is equal to the solution set of the following integer linear program:

$$\underset{\boldsymbol{q}\in\mathbb{Z}^{2^k}}{minimize}\quad C(\boldsymbol{q}), \tag{77}$$

$$subject\ to\quad A\boldsymbol{q}\leq\boldsymbol{b}. \tag{78}$$

*Proof.* This statement follows from Theorem C.1. □

*Remark* C.2 (Runtime analysis for obtaining an MMP representation of a BF). 0-1 integer linear programming, the decision problem in which variables are restricted to take on values in $\mathbb{B}$ and only the constraints must be satisfied, is NP-complete (Karp, 1972). This implies integer linear programming, the problem we solve to obtain an MMP representation of a BF, is NP-hard. We can also observe that the number of variables, constraints, and possible coefficient values in the program all grow according to $O(2^k)$.

Due to the problem being NP-hard, we bound the time complexity by considering a brute-force search for an MMP under a model of computation in which addition, multiplication, inequality checking, and computing the absolute value and indicator function all cost unit time. Let $\boldsymbol{q}\in\mathbb{Z}^{2^k}$, where $q_i\in\mathcal{Q}=\{-2^{k-1},\ldots,2^{k-1}\}\subseteq\mathbb{Z}$, meaning $q_i$ can take on $|\mathcal{Q}|=2^k+1$ possible values. Evaluating $A\boldsymbol{q}\leq\boldsymbol{b}$ from Corollary C.1, to check if $\boldsymbol{q}$ is a feasible point, requires $2^{2k+1}$ units of time. Moreover, computing cost function $\|\cdot\|_{0,\boldsymbol{c}}$ or $\|\cdot\|_{1,\boldsymbol{c}}$ requires $2^{k+1}+2^k-1$ units of time. Consequently, a brute-force search over all $|\mathcal{Q}|^{2^k}$ MMP candidates, checking feasibility with $A\boldsymbol{q}\leq\boldsymbol{b}$ and solution cost with either $\|\cdot\|_{0,\boldsymbol{c}}$ or $\|\cdot\|_{1,\boldsymbol{c}}$, requires time

$$T_{\mathtt{mpToMMP}}(k)=|\mathcal{Q}|^{2^k}\cdot(2^{2k+1}+2^{k+1}+2^k-1) \tag{79}$$

$$=(2^k+1)^{2^k}\cdot(2^{2k+1}+2^{k+1}+2^k-1) \tag{80}$$

$$\leq(2^k+2^k)^{2^k}\cdot(2^{2k+1}\cdot2^{2k+1}) \tag{81}$$

$$=(2^{k+1})^{2^k}\cdot2^{2k+2} \tag{82}$$

$$=2^{k2^k+2^k}\cdot2^{2k+2} \tag{83}$$

$$=2^{k2^k+2^k+2k+2} \tag{84}$$

$$=2^{O(k2^k)} \tag{85}$$

We note that while relaxing $\|\cdot\|_{0,\boldsymbol{c}}$ to $\|\cdot\|_{1,\boldsymbol{c}}$ does not reduce the asymptotic time complexity, in practice it allows for the use of branch and bound algorithms that make use of linear program subroutines over the reals. These algorithms tend to find solutions significantly faster than brute-force search. Such a relaxation can however lead to suboptimal solutions. We discuss this point further in Appendix C.1. We add that another relaxation would be to remove integer constraints, solve the resulting linear program, and then round the solution coefficients to the nearest integer. This would provide an algorithm with time complexity polynomial in $2^k$. However, since this kind of relaxation can lead to infeasible solutions that would break functional equivalence, we did not consider it.

## C.1 SUBOPTIMALITY OF $\ell_1$-RELAXATION IN MMP PROBLEMS

The weighted zero-norms $\|\boldsymbol{p}\|_{0,\boldsymbol{c}_k^{uni}}$ and $\|\boldsymbol{p}\|_{0,\boldsymbol{c}_k^{deg}}$ correspond to the number of neurons and (layer one) connections in the two-layer NN representation of MP $\boldsymbol{p}\in\mathbb{Z}^{2^k}$. Consequently, given that our objective is to find NN representations of BNs with as few neurons and connections as possible, we would like to find MMP representations of BFs that minimize these cost functions.

However, in Remark C.2, we saw that, in the worst case, solving such an optimization problem requires a brute-force search, which is computationally intractable even for small $k\in\mathbb{N}_0$. This motivated the relaxation to the weighted one-norm $\|\cdot\|_{1,\boldsymbol{c}}$ in our formulation of MMPs.

While a theoretical analysis of the consequences of this relaxation lies beyond the scope of this work, we discuss some preliminary investigations on the topic of solution suboptimality.

Table 2: The number (and percentage) of BFs for which the proposition in Equation 86 is true.

| | | $\boldsymbol{c}^{uni}$ | | $\boldsymbol{c}^{deg}$ | |
|---|---|---|---|---|---|
| $k$ | # BFs | Count | (%) | Count | (%) |
| 2 | 16 | 0 | 0.0 | 0 | 0.0 |
| 3 | 256 | 0 | 0.0 | 0 | 0.0 |
| 4 | 65536 | 180 | 0.3 | 557 | 0.8 |

Table 3: $\boldsymbol{c}_4^{uni}$- and $\boldsymbol{c}_4^{deg}$-weighted zero- and one-norms for MP vector $\boldsymbol{p}_{\texttt{0x635A}} \in \mathbb{Z}^{2^4}$ and its corresponding MMP vector $\boldsymbol{p}_{\texttt{0x635A}} \in \mathbb{Z}^{2^4}$.

| | $\boldsymbol{c}^{uni}$ | | $\boldsymbol{c}^{deg}$ | |
|---|---|---|---|---|
| | $\|\cdot\|_{0,\boldsymbol{c}_4^{uni}}$ | $\|\cdot\|_{1,\boldsymbol{c}_4^{uni}}$ | $\|\cdot\|_{0,\boldsymbol{c}_4^{deg}}$ | $\|\cdot\|_{1,\boldsymbol{c}_4^{deg}}$ |
| $\boldsymbol{p}_{\texttt{0x635A}}$ | 10 | 16 | 21 | 38 |
| $\boldsymbol{p}'_{\texttt{0x635A}}$ | 11 | 14 | 23 | 30 |
| Difference | -1 | 2 | -2 | 8 |

Particularly, we consider the following question. Let $\boldsymbol{p}' \in [\boldsymbol{p}]_\sigma$ be an MMP of MP $\boldsymbol{p} \in \mathbb{Z}^{2^k}$ w.r.t. the $\boldsymbol{c}$-weighted one-norm $\|\cdot\|_{1,\boldsymbol{c}}$, where $\boldsymbol{c} \in \{\boldsymbol{c}^{uni}, \boldsymbol{c}^{deg}\}$. Is the following proposition ever true?

$$\|\boldsymbol{p}\|_{1,\boldsymbol{c}} > \|\boldsymbol{p}'\|_{1,\boldsymbol{c}} \wedge \|\boldsymbol{p}\|_{0,\boldsymbol{c}} < \|\boldsymbol{p}'\|_{0,\boldsymbol{c}}. \tag{86}$$

That is, by optimizing the MP representation of the BF w.r.t. the weighted one-norm, we increase the value of the weighted zero-norm. In such a case, for the purposes of NN compression, we should choose the MP over the MMP. (Note that the results reported do not do this, selecting MMPs based on the value of the weighted one-norm). By iterating over all BFs for $k \in \{2,3,4\}$, we are able to answer this question in the affirmative. See Table 2. We find that this proposition is not true (for the MMPs found) for all BFs with $k = 2$ and $k = 3$ inputs. However, there is a small fraction of BFs with $k = 4$ inputs for which this proposition is true. We report one such function below.

Let $f_{\texttt{0x635A}} : \mathbb{B}^4 \to \mathbb{B}$ be the BF with truth table $\texttt{0x635A}$. The MP representation of $f_{\texttt{0x635A}}$ is

$$\begin{aligned} p_{\texttt{0x635A}}(x_1, x_2, x_3, x_4) = {}& x_4 + x_3 + x_1 - 2x_3x_4 - x_2x_4 - x_1x_4 - 2x_1x_3 \\ & + 2x_2x_3x_4 + 3x_1x_3x_4 - 2x_1x_2x_3x_4. \end{aligned} \tag{87}$$

The sets of MMP representations of $f_{\texttt{0x635A}}$ with respect to $\|\cdot\|_{1,\boldsymbol{c}_4^{uni}}$ and $\|\cdot\|_{1,\boldsymbol{c}_4^{deg}}$ have the following polynomial in their intersection

$$\begin{aligned} p'_{\texttt{0x635A}}(x_1, x_2, x_3, x_4) = {}& +x_4 + x_3 + x_1 - 2x_3x_4 - x_2x_4 + x_2x_3 - 2x_1x_3 \\ & + x_2x_3x_4 + 2x_1x_3x_4 - x_1x_2x_4 - x_1x_2x_3. \end{aligned} \tag{88}$$

We report the weighted zero- and one-norms of the corresponding MP and MMP vectors in Table 3. We see that under the $\boldsymbol{c}^{uni}$ criterion, the MMP reduces the weighted one-norm by 2, but increases the weighted zero-norm by 1, and that under the $\boldsymbol{c}^{deg}$ criterion, the MMP reduces the weighted one-norm by 8, but increases the weighted zero-norm by 2. Hence, the weighted one-norm MMP is suboptimal in comparison to the original MP in terms of the weighted zero-norm. In regard to the two-layer NN representation of MMP $p'_{\texttt{0x635A}}$, it would have one additional neuron and two additional connections in its first layer in comparison to the two-layer NN representation of MP $p_{\texttt{0x635A}}$.

We conclude this brief investigation by noting that there are many interesting questions in this direction that remain unanswered, such as the effect of MMP selection based on the weighted zero-norm (instead of the weighted one-norm) when considering the NN optimization algorithms `optAll` and `optMaintainDepth`, and, as pointed out by a reviewer, the consideration of alternative approximation methods for the weighted zero-norm MMP optimization problem.

## D   OBJECTIVE-AWARE OPTIMIZATION USING NPN CLASSES

**Definition D.1** (NPN transformation). A Negation-Permutation-Negation (NPN) transformation $\tau$ of a Boolean function (BF) ($\mathbb{F} = \mathbb{B}$) or multilinear polynomial (MP) ($\mathbb{F} = \mathbb{Z}$) is the composition of an input phase assignment $\tau_\phi$, an input permutation $\tau_\pi$, and an output polarity assignment $\tau_\nu$, where $\phi \subseteq \{1, \ldots, k\}$, $i \in \phi$ if and only if $x_i$ has negative phase (i.e., $x_i$ is to be negated), $\pi : \{1, \ldots, k\} \to \{1, \ldots, k\}$ is a bijection, $\nu \in \mathbb{B}$, and $\nu = 1$ if and only if the output polarity is negative (i.e., the function output is to be negated). Applying NPN transformation $\tau$ to vector $\boldsymbol{p} \in \mathbb{F}^{2^k}$ is denoted $\tau(\boldsymbol{p}) = (\tau_\nu \circ \tau_\pi \circ \tau_\phi)(\boldsymbol{p})$, where $\tau_\nu, \tau_\pi, \tau_\phi : \mathbb{F}^{2^k} \to \mathbb{F}^{2^k}$. Whereas applying transformation $\tau$ to function $p : \mathbb{F}^k \to \mathbb{F}$ is denoted $p \circ \tau = \tau_\nu \circ p \circ \tau_\pi \circ \tau_\phi$, where $\tau_\pi, \tau_\phi : \mathbb{F}^k \to \mathbb{F}^k$ and $\tau_\nu : \mathbb{F} \to \mathbb{F}$.

*Remark* D.1. Let $f : \mathbb{B}^3 \to \mathbb{B}$ be a BF in variables $x_1, x_2, x_3$. An input phase assignment corresponds to selecting a subset of variables in $f$ to negate. For example, the selection $\phi = \{2, 3\}$ corresponds to $g = f(x_1, \bar{x}_2, \bar{x}_3)$. We say that the phase of $x_2$ and $x_3$ in $g$ are negative w.r.t. $f$, whereas the phase of $x_1$ is positive w.r.t. $f$. Similarly, an output polarity assignment corresponds to selecting whether to negate the output of $f$. For example, the selection $\nu = 1$ corresponds to $h = \bar{f}(x_1, x_2, x_3)$. We say that the polarity of $h$ is negative w.r.t. $f$, whereas the polarity of $g$ is positive w.r.t. $f$.

**Definition D.2** (NPN equivalent). BFs $\boldsymbol{f}, \boldsymbol{g} \in \mathbb{B}^{2^k}$ are NPN equivalent, denoted $\boldsymbol{f} \equiv_{\text{NPN}} \boldsymbol{g}$, if there exists an NPN transformation $\tau$ such that $\tau(\boldsymbol{f}) = \boldsymbol{g}$.

**Definition D.3** (NPN equivalence class). The NPN equivalence class of BF $\boldsymbol{f} \in \mathbb{B}^{2^k}$, denoted $[\boldsymbol{f}]_{\text{NPN}}$, is the set $\{\boldsymbol{g} \in \mathbb{B}^{2^k} \mid \boldsymbol{f} \equiv_{\text{NPN}} \boldsymbol{g}\}$.

**Definition D.4** (NPN canonical form). The NPN canonical form of BF $\boldsymbol{f} \in \mathbb{B}^{2^k}$, denoted $\kappa(\boldsymbol{f})$, is a mapping $\kappa : \mathbb{B}^{2^k} \to \mathbb{B}^{2^k}$ that satisfies the following conditions:

1. $\kappa(\boldsymbol{f}) \in [\boldsymbol{f}]_{\text{NPN}}$,

2. $\forall \boldsymbol{g} \in [\boldsymbol{f}]_{\text{NPN}} \big(\kappa(\boldsymbol{g}) = \kappa(\boldsymbol{f})\big)$.

Note that the second condition implies that the NPN canonical form of a BF is a unique representative of the NPN equivalence class. We denote $\kappa(\boldsymbol{f})$ as $\kappa_{\boldsymbol{f}} \in \mathbb{B}^{2^k}$, and the MP of $\kappa_{\boldsymbol{f}}$ as $\kappa_{\boldsymbol{p}} \in \mathbb{Z}^{2^k}$.

*Remark* D.2. There are $2^{k+1} k!$ NPN transformations for $k$-input BFs, as there are $2^k$ possible input phase assignments, $k!$ possible input permutations, and 2 possible output polarity assignments. Hence, an NPN equivalence class contains at most $2^{k+1} k!$ BFs.

*Remark* D.3. The MP representations of BFs $\bar{x}_1$, $x_1 \wedge x_2$, and $x_1 \vee x_2$ are $1 - x_1$, $x_1 x_2$, and $x_1 + x_2 - x_1 x_2$, respectively.

**Lemma D.1** (Cofactors of an MP vector). *Let $i \in \{1, \ldots, k\}$, $p : \mathbb{Z}^k \to \mathbb{Z}$ be an MP, and $p_{x_i}, p_{\bar{x}_i} : \mathbb{Z}^{k-1} \to \mathbb{Z}$ be the positive and negative cofactors of $p$ with respect to $x_i$, respectively. Then the MP vectors $\boldsymbol{p}_{x_i}, \boldsymbol{p}_{\bar{x}_i} \in \mathbb{Z}^{2^{k-1}}$ of $p_{x_i}$ and $p_{\bar{x}_i}$ are given by the following expressions*

$$\boldsymbol{p}_{x_i} = \sum_{S \in \mathcal{P}(\{1,\ldots,k\} \setminus \{i\})} \big(\hat{f}(S) + \hat{f}(S \cup \{i\})\big) \boldsymbol{\chi}_S, \tag{89}$$

$$\boldsymbol{p}_{\bar{x}_i} = \sum_{S \in \mathcal{P}(\{1,\ldots,k\} \setminus \{i\})} \hat{f}(S) \boldsymbol{\chi}_S. \tag{90}$$

*Proof.* Following the definition of an MP as a linear combination of logical AND functions, we have

$$p(\boldsymbol{x}) = \sum_{S \in \mathcal{P}(\{1,\ldots,k\})} \hat{f}(S) \chi_S(\boldsymbol{x}) \tag{91}$$

$$= \sum_{S \in \mathcal{P}(\{1,\ldots,k\} \setminus \{i\})} \hat{f}(S) \chi_S(\boldsymbol{x}) + \sum_{S \in \overline{\mathcal{P}(\{1,\ldots,k\} \setminus \{i\})}} \hat{f}(S) \chi_S(\boldsymbol{x}), \tag{92}$$

where we use $\overline{\mathcal{P}(\{1,\ldots,k\}\setminus\{i\})}$ to denote $\mathcal{P}(\{1,\ldots,k\})\setminus\mathcal{P}(\{1,\ldots,k\}\setminus\{i\})$. Thus,

$$p_{x_i} = p(\ldots, x_i = 1, \ldots) \tag{93}$$

$$= \sum_{S\in\mathcal{P}(\{1,\ldots,k\}\setminus\{i\})} \hat{f}(S)\chi_S(\ldots, x_i = 1, \ldots) + \sum_{S\in\overline{\mathcal{P}(\{1,\ldots,k\}\setminus\{i\})}} \hat{f}(S)\chi_S(\ldots, x_i = 1, \ldots) \tag{94}$$

$$= \sum_{S\in\mathcal{P}(\{1,\ldots,k\}\setminus\{i\})} \hat{f}(S)\chi_S(\ldots, x_i = 1, \ldots) + \sum_{S\in\overline{\mathcal{P}(\{1,\ldots,k\}\setminus\{i\})}} \hat{f}(S)\chi_{S\setminus\{i\}}(\ldots, x_i = 1, \ldots) \tag{95}$$

$$= \sum_{S\in\mathcal{P}(\{1,\ldots,k\}\setminus\{i\})} (\hat{f}(S) + \hat{f}(S\cup\{i\}))\chi_S(\ldots, x_i = 1, \ldots), \tag{96}$$

and

$$p_{x_i} = p(\ldots, x_i = 0, \ldots) \tag{97}$$

$$= \sum_{S\in\mathcal{P}(\{1,\ldots,k\}\setminus\{i\})} \hat{f}(S)\chi_S(\ldots, x_i = 0, \ldots) + \sum_{S\in\overline{\mathcal{P}(\{1,\ldots,k\}\setminus\{i\})}} \hat{f}(S)\chi_S(\ldots, x_i = 0, \ldots) \tag{98}$$

$$= \sum_{S\in\mathcal{P}(\{1,\ldots,k\}\setminus\{i\})} \hat{f}(S)\chi_S(\ldots, x_i = 0, \ldots). \tag{99}$$

$\square$

**Lemma D.2** ($\tau_{\phi_i}$ acting on an MP vector). *Let $p \in \mathbb{Z}^k \to \mathbb{Z}$ be an MP with MP vector $\boldsymbol{p} \in \mathbb{Z}^{2^k}$, and $\phi_i = \{i\} \subseteq \{1,\ldots,k\}$. Then the MP vector $\tau_{\phi_i}(\boldsymbol{p}) \in \mathbb{Z}^{2^k}$ is given by the expression*

$$\tau_{\phi_i}(\boldsymbol{p}) = \sum_{S\in\mathcal{P}(\{1,\ldots,k\}\setminus\{i\})} (\hat{f}(S) + \hat{f}(S\cup\{i\}))\boldsymbol{\chi}_S + \sum_{S\in\overline{\mathcal{P}(\{1,\ldots,k\}\setminus\{i\})}} (-\hat{f}(S))\boldsymbol{\chi}_S. \tag{100}$$

*Proof.* Following Boole's expansion theorem,

$$(p \circ \tau_{\phi_i})(\boldsymbol{x}) = p(\tau_{\phi_i}(\boldsymbol{x})) \tag{101}$$

$$= (1 - x_i)p_{x_i}(\boldsymbol{x}) + x_i p_{\bar{x}_i}(\boldsymbol{x}) \tag{102}$$

$$= p_{x_i}(\boldsymbol{x}) + (p_{\bar{x}_i}(\boldsymbol{x}) - p_{x_i}(\boldsymbol{x}))x_i. \tag{103}$$

Substituting for the cofactors according to [Lemma D.1](#), we have

$$(p \circ \tau_{\phi_i})(\boldsymbol{x})$$

$$= \sum_{S\in\mathcal{P}(\{1,\ldots,k\}\setminus\{i\})} (\hat{f}(S) + \hat{f}(S\cup\{i\}))\chi_S(\boldsymbol{x})$$

$$+ \Big( \sum_{S\in\mathcal{P}(\{1,\ldots,k\}\setminus\{i\})} \hat{f}(S)\chi_S(\boldsymbol{x}) - \sum_{S\in\mathcal{P}(\{1,\ldots,k\}\setminus\{i\})} (\hat{f}(S) + \hat{f}(S\cup\{i\}))\chi_S(\boldsymbol{x}) \Big) x_i \tag{104}$$

$$= \sum_{S\in\mathcal{P}(\{1,\ldots,k\}\setminus\{i\})} (\hat{f}(S) + \hat{f}(S\cup\{i\}))\chi_S(\boldsymbol{x})$$

$$+ \Big( \sum_{S\in\mathcal{P}(\{1,\ldots,k\}\setminus\{i\})} (-\hat{f}(S\cup\{i\}))\chi_S(\boldsymbol{x}) \Big) x_i \tag{105}$$

$$= \sum_{S\in\mathcal{P}(\{1,\ldots,k\}\setminus\{i\})} (\hat{f}(S) + \hat{f}(S\cup\{i\}))\chi_S(\boldsymbol{x}) + \sum_{S\in\overline{\mathcal{P}(\{1,\ldots,k\}\setminus\{i\})}} (-\hat{f}(S))\chi_S(\boldsymbol{x}). \tag{106}$$

$\square$

**Lemma D.3** ($\tau_\pi$ acting on an MP vector). *Let $p \in \mathbb{Z}^k \to \mathbb{Z}$ be an MP with MP vector $\boldsymbol{p} \in \mathbb{Z}^{2^k}$, and $\pi : \{1,\ldots,k\} \to \{1,\ldots,k\}$ be a bijection. Then the MP vector $\tau_\pi(\boldsymbol{p}) \in \mathbb{Z}^{2^k}$ is given by the expression*

$$\tau_\pi(\boldsymbol{p}) = \sum_{S\in\mathcal{P}(\{1,\ldots,k\})} \hat{f}(\pi^{-1}(S))\boldsymbol{\chi}_S. \tag{107}$$

*Proof.*

$$(p \circ \tau_\pi)(\boldsymbol{x}) = p(\tau_\pi(\boldsymbol{x})) \tag{108}$$

$$= \sum_{S \in \mathcal{P}(\{1,\dots,k\})} \hat{f}(S) \chi_S(\tau_\pi(\boldsymbol{x})) \tag{109}$$

$$= \sum_{S \in \mathcal{P}(\{1,\dots,k\})} \hat{f}(S) \chi_{\pi(S)}(\boldsymbol{x}) \tag{110}$$

$$= \sum_{S \in \mathcal{P}(\{1,\dots,k\})} \hat{f}(\pi^{-1}(S)) \chi_S(\boldsymbol{x}). \tag{111}$$

$\square$

**Lemma D.4** ($\tau_\nu$ acting on an MP vector). *Let $p \in \mathbb{Z}^{2^k}$ be an MP with MP vector $\boldsymbol{p} \in \mathbb{Z}^{2^k}$, and $\nu = 1$. Then the MP vector $\tau_\nu(\boldsymbol{p}) \in \mathbb{Z}^{2^k}$ is given by the expression*

$$\tau_\nu(\boldsymbol{p}) = \big(1 - \hat{f}(\emptyset)\big)\boldsymbol{\chi}_\emptyset + \sum_{S \in \mathcal{P}(\{1,\dots,k\})\setminus\emptyset} \big(-\hat{f}(S)\big)\boldsymbol{\chi}_S. \tag{112}$$

*Proof.*

$$(\tau_\nu \circ p)(\boldsymbol{x}) = 1 - p(\boldsymbol{x}) \tag{113}$$

$$= 1 - \Big( \sum_{S \in \mathcal{P}(\{1,\dots,k\})} \hat{f}(S)\chi_S(\boldsymbol{x}) \Big) \tag{114}$$

$$= 1 - \Big( \hat{f}(\emptyset)\chi_\emptyset(\boldsymbol{x}) + \sum_{S \in \mathcal{P}(\{1,\dots,k\})\setminus\emptyset} \hat{f}(S)\chi_S(\boldsymbol{x}) \Big) \tag{115}$$

$$= \big(1 - \hat{f}(\emptyset)\big)\chi_\emptyset(\boldsymbol{x}) + \sum_{S \in \mathcal{P}(\{1,\dots,k\})\setminus\emptyset} \big(-\hat{f}(S)\big)\chi_S(\boldsymbol{x}). \tag{116}$$

Where the final equality holds since $\chi_\emptyset(\boldsymbol{x}) = 1$. $\square$

*Example* D.1. Let $f(x_1, x_2, x_3) = (x_1 \wedge x_2) \vee x_3$. The MP representation of $f$ is $p(x_1, x_2, x_3) = x_3 + x_1 x_2 - x_1 x_2 x_3$, and the MP vector of $f$ is

$$\boldsymbol{p} = \sum_{S \in \mathcal{P}(\{1,2,3\})} \hat{f}(S)\boldsymbol{\chi}_S$$

$$= \big[\hat{f}(\emptyset), \hat{f}(\{1\}), \hat{f}(\{2\}), \hat{f}(\{1,2\}), \hat{f}(\{3\}), \hat{f}(\{1,3\}), \hat{f}(\{2,3\}), \hat{f}(\{1,2,3\})\big]^\intercal$$

$$= [0, 0, 0, 1, 1, 0, 0, -1]^\intercal.$$

Following Lemma D.2, $\tau_{\phi_3}(\boldsymbol{p}) \in \mathbb{Z}^{2^3}$ is equal to

$$\tau_{\phi_3}(\boldsymbol{p}) = \sum_{S \in \mathcal{P}(\{1,2,3\}\setminus\{3\})} \big(\hat{f}(S) + \hat{f}(S \cup \{i\})\big)\boldsymbol{\chi}_S + \sum_{S \in \overline{\mathcal{P}(\{1,2,3\}\setminus\{3\})}} \big(-\hat{f}(S)\big)\boldsymbol{\chi}_S \tag{117}$$

$$= \begin{bmatrix} \hat{f}(\emptyset) + \hat{f}(\emptyset \cup \{3\}) \\ \hat{f}(\{1\}) + \hat{f}(\{1\} \cup \{3\}) \\ \hat{f}(\{2\}) + \hat{f}(\{2\} \cup \{3\}) \\ \hat{f}(\{1,2\}) + \hat{f}(\{1,2\} \cup \{3\}) \\ -\hat{f}(\{3\}) \\ -\hat{f}(\{1,3\}) \\ -\hat{f}(\{2,3\}) \\ -\hat{f}(\{1,2,3\}) \end{bmatrix} = \begin{bmatrix} (0) + (1) \\ (0) + (0) \\ (0) + (0) \\ (1) + (-1) \\ -(1) \\ -(0) \\ -(0) \\ -(-1) \end{bmatrix} = \begin{bmatrix} 1 \\ 0 \\ 0 \\ 0 \\ -1 \\ 0 \\ 0 \\ 1 \end{bmatrix}, \tag{118}$$

which corresponds to the MP $(p \circ \tau_{\phi_3})(x_1, x_2, x_3) = (1-x_3) + x_1 x_2 - x_1 x_2(1-x_3) = 1 - x_3 + x_1 x_2 x_3$.

Let $\pi_{231} = \left( \begin{smallmatrix} 1 & 2 & 3 \\ 2 & 3 & 1 \end{smallmatrix} \right)$ be a permutation with inverse $\pi_{231}^{-1} = \pi_{312} = \left( \begin{smallmatrix} 1 & 2 & 3 \\ 3 & 1 & 2 \end{smallmatrix} \right)$. Following Lemma D.3, $\tau_{\pi_{231}}(\boldsymbol{p}) \in \mathbb{Z}^{2^3}$ is equal to

$$\tau_{\pi_{231}}(\boldsymbol{p}) = \sum_{S \in \mathcal{P}(\{1,2,3\})} \hat{f}(\pi_{231}^{-1}(S)) \boldsymbol{\chi}_S \tag{119}$$

$$= \begin{bmatrix} \pi_{312}(\hat{f}(\emptyset)) \\ \pi_{312}(\hat{f}(\{1\})) \\ \pi_{312}(\hat{f}(\{2\})) \\ \pi_{312}(\hat{f}(\{1,2\})) \\ \pi_{312}(\hat{f}(\{3\})) \\ \pi_{312}(\hat{f}(\{1,3\})) \\ \pi_{312}(\hat{f}(\{2,3\})) \\ \pi_{312}(\hat{f}(\{1,2,3\})) \end{bmatrix} = \begin{bmatrix} \hat{f}(\emptyset) \\ \hat{f}(\{3\}) \\ \hat{f}(\{1\}) \\ \hat{f}(\{1,3\}) \\ \hat{f}(\{2\}) \\ \hat{f}(\{2,3\}) \\ \hat{f}(\{1,2\}) \\ \hat{f}(\{1,2,3\}) \end{bmatrix} = \begin{bmatrix} 0 \\ 1 \\ 0 \\ 0 \\ 0 \\ 0 \\ 1 \\ -1 \end{bmatrix}, \tag{120}$$

which corresponds to the MP $(p \circ \tau_{\pi_{231}})(x_1, x_2, x_3) = x_1 + x_2 x_3 - x_1 x_2 x_3$.

Lastly, let $\nu = 1$. Following Lemma D.4, $\tau_\nu(\boldsymbol{p}) \in \mathbb{Z}^{2^3}$ is equal to

$$\tau_\nu(\boldsymbol{p}) = \left(1 - \hat{f}(\emptyset)\right)\boldsymbol{\chi}_\emptyset + \sum_{S \in \mathcal{P}(\{1,2,3\})\setminus\emptyset} \left(-\hat{f}(S)\right)\boldsymbol{\chi}_S \tag{121}$$

$$= \begin{bmatrix} 1 - \hat{f}(\emptyset) \\ -\hat{f}(\{1\}) \\ -\hat{f}(\{2\}) \\ -\hat{f}(\{1,2\}) \\ -\hat{f}(\{3\}) \\ -\hat{f}(\{1,3\}) \\ -\hat{f}(\{2,3\}) \\ -\hat{f}(\{1,2,3\}) \end{bmatrix} = \begin{bmatrix} 1-(0) \\ -(0) \\ -(0) \\ -(1) \\ -(1) \\ -(0) \\ -(0) \\ -(-1) \end{bmatrix} = \begin{bmatrix} 1 \\ 0 \\ 0 \\ -1 \\ -1 \\ 0 \\ 0 \\ 1 \end{bmatrix}, \tag{122}$$

which corresponds to the MP $(\tau_\nu \circ p)(x_1, x_2, x_3) = 1 - (x_3 + x_1 x_2 - x_1 x_2 x_3) = 1 - x_3 - x_1 x_2 + x_1 x_2 x_3$.

*Remark* D.4 (Linear algebra perspective). Let $\boldsymbol{p} \in \mathbb{Z}^{2^k}$ be an MP vector, $\phi_i = \{i\} \subseteq \{1, \dots, k\}$, $\pi, \pi_{1 \leftrightarrow i} : \{1, \dots, k\} \to \{1, \dots, k\}$ be bijections, where $\pi_{1 \leftrightarrow i}$ represents the transposition $(1, i)$, and $\nu = 1$. Lemmas D.2, D.3, and D.4 respectively show

$$\tau_{\phi_i}(\boldsymbol{p}) = \left( P_{\pi_{1 \leftrightarrow i}} \begin{bmatrix} I_{k-1}, & I_{k-1} \\ O_{k-1}, & -I_{k-1} \end{bmatrix} P_{\pi_{1 \leftrightarrow i}} \right) \boldsymbol{p}, \tag{123}$$

$$\tau_\pi(\boldsymbol{p}) = P_\pi \boldsymbol{p}, \tag{124}$$

$$\tau_\nu(\boldsymbol{p}) = \begin{bmatrix} 1 \\ \boldsymbol{0}_{2^k-1} \end{bmatrix} - \boldsymbol{p}, \tag{125}$$

where $\boldsymbol{0}_{2^k-1} \in \mathbb{B}^{2^k-1}$, $O_{k-1} \in \mathbb{B}^{2^{k-1} \times 2^{k-1}}$, and $I_{k-1} \in \mathbb{B}^{2^{k-1} \times 2^{k-1}}$ denote the all zeros vector, all zeros matrix, and identity matrix, respectively, and $P_\pi, P_{\pi_{1 \leftrightarrow i}} \in \mathbb{B}^{2^k \times 2^k}$ denote permutation matrices. We see $\tau_{\phi_i}$ (in turn $\tau_\phi$) and $\tau_\pi$ are linear transformations, whereas $\tau_\nu$ is an affine transformation.

*Remark* D.5 (Time complexity of computing NPN transformations). If $\boldsymbol{p} \in \mathbb{Z}^{2^k}$ is stored in memory as a tensor of shape $\underbrace{(2, \dots, 2)}_{k}$, then computing $\tau_\pi(\boldsymbol{p})$ requires time $O(k)$ from permuting tensor indices. Computing $\tau_{\phi_i}(\boldsymbol{p})$ requires time $O(k) + O(2^k) + O(k) = O(2^k)$ from (i) tensor index transposition, (ii) adding the bottom half of $\boldsymbol{p}$ to the top half ($O(2^{k-1})$) and then sign flipping the bottom half ($O(2^{k-1})$), and (iii) tensor index transposition. Lastly, computing $\tau_\nu(\boldsymbol{p})$ requires time $O(2^k)$ from vector addition.

### D.1 Computing Inverse NPN Transformations

**Lemma D.5** (Commutability of $\tau_\nu$ and $\tau_\pi$). *Let $\boldsymbol{p} \in \mathbb{Z}^{2^k}$ be an MP vector, $\nu \in \mathbb{B}$ and $\pi : \{1, \ldots, k\} \to \{1, \ldots, k\}$ be a bijection. Then $(\tau_\nu \circ \tau_\pi)(\boldsymbol{p}) = (\tau_\pi \circ \tau_\nu)(\boldsymbol{p})$.*

*Proof.* If $\nu = 0$, then $\tau_\nu = Id.$ and the statement is clearly true. We now consider the case where $\nu = 1$. Under the functional perspective of $\tau$, the statement is equivalent to showing $\tau_\nu \circ (p \circ \tau_\pi) = (\tau_\nu \circ p) \circ \tau_\pi$, where $p : \mathbb{Z}^k \to \mathbb{Z}$ is an MP, which holds by associativity of function composition. The statement also follows from Lemmas D.3 and D.4,

$$(\tau_\nu \circ \tau_\pi)(\boldsymbol{p}) = \tau_\nu\Big( \sum_{S \in \mathcal{P}(\{1,\ldots,k\})} \hat{f}(\pi^{-1}(S))\boldsymbol{\chi}_S \Big) \tag{126}$$

$$= \tau_\nu\Big( \hat{f}(\pi^{-1}(\emptyset))\boldsymbol{\chi}_\emptyset + \sum_{S \in \mathcal{P}(\{1,\ldots,k\})\setminus\{\emptyset\}} \hat{f}(\pi^{-1}(S))\boldsymbol{\chi}_S \Big) \tag{127}$$

$$= \tau_\nu\Big( \hat{f}(\emptyset)\boldsymbol{\chi}_\emptyset + \sum_{S \in \mathcal{P}(\{1,\ldots,k\})\setminus\{\emptyset\}} \hat{f}(\pi^{-1}(S))\boldsymbol{\chi}_S \Big) \tag{128}$$

$$= (1 - \hat{f}(\emptyset))\boldsymbol{\chi}_\emptyset + \sum_{S \in \mathcal{P}(\{1,\ldots,k\})\setminus\{\emptyset\}} (-\hat{f}(\pi^{-1}(S)))\boldsymbol{\chi}_S \tag{129}$$

$$(\tau_\pi \circ \tau_\nu)(\boldsymbol{p}) = \tau_\pi\Big( (1 - \hat{f}(\emptyset))\boldsymbol{\chi}_\emptyset + \sum_{S \in \mathcal{P}(\{1,\ldots,k\})\setminus\{\emptyset\}} (-\hat{f}(S))\boldsymbol{\chi}_S \Big) \tag{130}$$

$$= \tau_\pi\Big( (\hat{f}'(\emptyset))\boldsymbol{\chi}_\emptyset + \sum_{S \in \mathcal{P}(\{1,\ldots,k\})\setminus\{\emptyset\}} \hat{f}'(S)\boldsymbol{\chi}_S \Big) \tag{131}$$

$$= \hat{f}'(\pi^{-1}(\emptyset))\boldsymbol{\chi}_\emptyset + \sum_{S \in \mathcal{P}(\{1,\ldots,k\})\setminus\{\emptyset\}} \hat{f}'(\pi^{-1}(S))\boldsymbol{\chi}_S \tag{132}$$

$$= \hat{f}'(\emptyset)\boldsymbol{\chi}_\emptyset + \sum_{S \in \mathcal{P}(\{1,\ldots,k\})\setminus\{\emptyset\}} \hat{f}'(\pi^{-1}(S))\boldsymbol{\chi}_S \tag{133}$$

$$= (1 - \hat{f}(\emptyset))\boldsymbol{\chi}_\emptyset + \sum_{S \in \mathcal{P}(\{1,\ldots,k\})\setminus\{\emptyset\}} (-\hat{f}(\pi^{-1}(S)))\boldsymbol{\chi}_S \tag{134}$$

$\square$

**Lemma D.6** (Commutability of $\tau_\nu$ and $\tau_\phi$). *Let $\boldsymbol{p} \in \mathbb{Z}^{2^k}$ be an MP vector, $\nu \in \mathbb{B}$, and $\phi \subseteq \{1, \ldots, k\}$. Then $(\tau_\nu \circ \tau_\phi)(\boldsymbol{p}) = (\tau_\phi \circ \tau_\nu)(\boldsymbol{p})$.*

*Proof.* We begin by noting that $\tau_\phi = \tau_{\phi_1} \circ \cdots \circ \tau_{\phi_k}$, where $\tau_{\phi_i} = \tau_{\{i\}}$ if $i \in \phi$, and $\tau_{\phi_i} = Id.$ if $i \notin \phi$. Now, to prove the statement of the lemma, it suffices to show $(\tau_\nu \circ \tau_{\phi_i})(\boldsymbol{p}) = (\tau_{\phi_i} \circ \tau_\nu)(\boldsymbol{p})$. If $\nu = 0$ or $i \notin \phi$, then either $\tau_\nu = Id.$ or $\tau_{\phi_i} = Id.$, and the statement is clearly true. We now consider the case where $\nu = 1$ and $i \in \phi$. Under the functional perspective of $\tau$, the statement is equivalent to showing $\tau_\nu \circ (p \circ \tau_{\phi_i}) = (\tau_\nu \circ p) \circ \tau_{\phi_i}$, where $p : \mathbb{Z}^k \to \mathbb{Z}$, which holds by associativity of function composition. The statement also follows from Lemmas D.2 and D.4,

$$(\tau_\nu \circ \tau_{\phi_i})(\boldsymbol{p})$$

$$= \tau_\nu \Big( \sum_{S \in \mathcal{P}(\{1,\dots,k\}\setminus\{i\})} (\hat{f}(S) + \hat{f}(S \cup \{i\})) \boldsymbol{\chi}_S + \sum_{S \in \overline{\mathcal{P}(\{1,\dots,k\}\setminus\{i\})}} (-\hat{f}(S)) \boldsymbol{\chi}_S \Big) \tag{135}$$

$$= \tau_\nu \Big( (\hat{f}(\emptyset) + \hat{f}(\emptyset \cup \{i\})) \boldsymbol{\chi}_\emptyset$$
$$+ \sum_{S \in \mathcal{P}(\{1,\dots,k\}\setminus\{i\})\setminus\{\emptyset\}} (\hat{f}(S) + \hat{f}(S \cup \{i\})) \boldsymbol{\chi}_S + \sum_{S \in \overline{\mathcal{P}(\{1,\dots,k\}\setminus\{i\})}} (-\hat{f}(S)) \boldsymbol{\chi}_S \Big) \tag{136}$$

$$= \tau_\nu \Big( \hat{f}'(\emptyset) \boldsymbol{\chi}_\emptyset$$
$$+ \sum_{S \in \mathcal{P}(\{1,\dots,k\}\setminus\{i\})\setminus\{\emptyset\}} \hat{f}'(S) \boldsymbol{\chi}_S + \sum_{S \in \overline{\mathcal{P}(\{1,\dots,k\}\setminus\{i\})}} \hat{f}'(S) \boldsymbol{\chi}_S \Big) \tag{137}$$

$$= (1 - \hat{f}'(\emptyset)) \boldsymbol{\chi}_\emptyset$$
$$+ \sum_{S \in \mathcal{P}(\{1,\dots,k\}\setminus\{i\})\setminus\{\emptyset\}} (-\hat{f}'(S)) \boldsymbol{\chi}_S + \sum_{S \in \overline{\mathcal{P}(\{1,\dots,k\}\setminus\{i\})}} (-\hat{f}'(S)) \boldsymbol{\chi}_S \tag{138}$$

$$= (1 - (\hat{f}(\emptyset) + \hat{f}(\emptyset \cup \{i\}))) \boldsymbol{\chi}_\emptyset$$
$$+ \sum_{S \in \mathcal{P}(\{1,\dots,k\}\setminus\{i\})\setminus\{\emptyset\}} (-(\hat{f}(S) + \hat{f}(S \cup \{i\}))) \boldsymbol{\chi}_S + \sum_{S \in \overline{\mathcal{P}(\{1,\dots,k\}\setminus\{i\})}} (-(-\hat{f}(S))) \boldsymbol{\chi}_S \tag{139}$$

$$= (1 - \hat{f}(\emptyset) - \hat{f}(\emptyset \cup \{i\}))) \boldsymbol{\chi}_\emptyset$$
$$+ \sum_{S \in \mathcal{P}(\{1,\dots,k\}\setminus\{i\})\setminus\{\emptyset\}} (-\hat{f}(S) - \hat{f}(S \cup \{i\})) \boldsymbol{\chi}_S + \sum_{S \in \overline{\mathcal{P}(\{1,\dots,k\}\setminus\{i\})}} \hat{f}(S) \boldsymbol{\chi}_S \tag{140}$$

$$(\tau_{\phi_i} \circ \tau_\nu)(\boldsymbol{p})$$

$$= \tau_{\phi_i} \Big( (1 - \hat{f}(\emptyset)) \boldsymbol{\chi}_\emptyset + \sum_{S \in \mathcal{P}(\{1,\dots,k\})\setminus\{\emptyset\}} (-\hat{f}(S)) \boldsymbol{\chi}_S \Big) \tag{141}$$

$$= \tau_{\phi_i} \Big( \hat{f}'(\emptyset) \boldsymbol{\chi}_\emptyset + \sum_{S \in \mathcal{P}(\{1,\dots,k\})\setminus\{\emptyset\}} \hat{f}'(S) \boldsymbol{\chi}_S \Big) \tag{142}$$

$$= \tau_{\phi_i} \Big( \hat{f}'(\emptyset) \boldsymbol{\chi}_\emptyset + \sum_{S \in \mathcal{P}(\{1,\dots,k\}\setminus\{i\})\setminus\{\emptyset\}} \hat{f}'(S) \boldsymbol{\chi}_S + \sum_{S \in \overline{\mathcal{P}(\{1,\dots,k\}\setminus\{i\})}} \hat{f}'(S) \boldsymbol{\chi}_S \Big) \tag{143}$$

$$= (\hat{f}'(\emptyset) + \hat{f}'(\emptyset \cup \{i\})) \boldsymbol{\chi}_\emptyset$$
$$+ \sum_{S \in \mathcal{P}(\{1,\dots,k\}\setminus\{i\})\setminus\{\emptyset\}} (\hat{f}'(S) + \hat{f}'(S \cup \{i\})) \boldsymbol{\chi}_S + \sum_{S \in \overline{\mathcal{P}(\{1,\dots,k\}\setminus\{i\})}} (-\hat{f}'(S)) \boldsymbol{\chi}_S \tag{144}$$

$$= ((1 - \hat{f}(\emptyset)) + (-1 \cdot \hat{f}(\emptyset \cup \{i\}))) \boldsymbol{\chi}_\emptyset$$
$$+ \sum_{S \in \mathcal{P}(\{1,\dots,k\}\setminus\{i\})\setminus\{\emptyset\}} ((-\hat{f}(S)) + (-\hat{f}(S \cup \{i\}))) \boldsymbol{\chi}_S + \sum_{S \in \overline{\mathcal{P}(\{1,\dots,k\}\setminus\{i\})}} (-(-\hat{f}(S))) \boldsymbol{\chi}_S$$
$$\tag{145}$$

$$= (1 - \hat{f}(\emptyset) - \hat{f}(\emptyset \cup \{i\})) \boldsymbol{\chi}_\emptyset$$
$$+ \sum_{S \in \mathcal{P}(\{1,\dots,k\}\setminus\{i\})\setminus\{\emptyset\}} (-\hat{f}(S) - \hat{f}(S \cup \{i\})) \boldsymbol{\chi}_S + \sum_{S \in \overline{\mathcal{P}(\{1,\dots,k\}\setminus\{i\})}} \hat{f}(S) \boldsymbol{\chi}_S \tag{146}$$

$$\square$$

**Lemma D.7** (Rearrangement of $\tau_\pi$ and $\tau_\phi$). *Let $\boldsymbol{p} \in \mathbb{Z}^{2^k}$ be an MP vector, $\phi \subseteq \{1, \dots, k\}$, and $\pi : \{1, \dots, k\} \to \{1, \dots, k\}$ be a bijection. Then $(\tau_\phi \circ \tau_{\pi^{-1}})(\boldsymbol{p}) = (\tau_{\pi^{-1}} \circ \tau_{\pi(\phi)})(\boldsymbol{p})$.*

*Proof.* We begin by noting that $\tau_\phi = \tau_{\phi_1} \circ \cdots \circ \tau_{\phi_k}$, where $\tau_{\phi_i} = \tau_{\{i\}}$ if $i \in \phi$, and $\tau_{\phi_i} = Id.$ if $i \notin \phi$. Now, to prove the statement of the lemma, it suffices to show $(\tau_{\phi_i} \circ \tau_{\pi^{-1}})(\boldsymbol{p}) = (\tau_{\pi^{-1}} \circ \tau_{\pi(\phi)_{\pi(i)}})(\boldsymbol{p})$. If $i \notin \phi$, then $\pi(i) \notin \pi(\phi)$, $\tau_{\phi_i} = Id.$, $\tau_{\pi(\phi)_{\pi(i)}} = Id.$, and the statement is clearly true. We now consider the case where $i \in \phi$, which implies $\pi(i) \in \pi(\phi)$.

$$(\tau_{\phi_i} \circ \tau_{\pi^{-1}})(\boldsymbol{p})$$

$$= \tau_{\phi_i}\Big( \sum_{S \in \mathcal{P}(\{1,\ldots,k\})} \hat{f}(\pi(S))\boldsymbol{\chi}_S \Big) \tag{147}$$

$$= \tau_{\phi_i}\Big( \sum_{S \in \mathcal{P}(\{1,\ldots,k\})} \hat{f}'(S)\boldsymbol{\chi}_S \Big) \tag{148}$$

$$= \sum_{S \in \mathcal{P}(\{1,\ldots,k\}\backslash\{i\})} (\hat{f}'(S) + \hat{f}'(S \cup \{i\}))\boldsymbol{\chi}_S + \sum_{S \in \overline{\mathcal{P}(\{1,\ldots,k\}\backslash\{i\})}} (-\hat{f}'(S))\boldsymbol{\chi}_S \tag{149}$$

$$= \sum_{S \in \mathcal{P}(\{1,\ldots,k\}\backslash\{i\})} (\hat{f}(\pi(S)) + \hat{f}(\pi(S \cup \{i\})))\boldsymbol{\chi}_S + \sum_{S \in \overline{\mathcal{P}(\{1,\ldots,k\}\backslash\{i\})}} (-\hat{f}(\pi(S)))\boldsymbol{\chi}_S \tag{150}$$

$$= \sum_{S \in \mathcal{P}(\{1,\ldots,k\}\backslash\{i\})} (\hat{f}(\pi(S)) + \hat{f}(\pi(S) \cup \{\pi(i)\}))\boldsymbol{\chi}_S + \sum_{S \in \overline{\mathcal{P}(\{1,\ldots,k\}\backslash\{i\})}} (-\hat{f}(\pi(S)))\boldsymbol{\chi}_S \tag{151}$$

$$(\tau_{\pi^{-1}} \circ \tau_{\pi(\phi)_{\pi(i)}})(\boldsymbol{p})$$

$$= \tau_{\pi^{-1}}\Big( \sum_{S \in \mathcal{P}(\{1,\ldots,k\}\backslash\{\pi(i)\})} (\hat{f}(S) + \hat{f}(S \cup \{\pi(i)\}))\boldsymbol{\chi}_S + \sum_{S \in \overline{\mathcal{P}(\{1,\ldots,k\}\backslash\{\pi(i)\})}} (-\hat{f}(S))\boldsymbol{\chi}_S \Big) \tag{152}$$

$$= \tau_{\pi^{-1}}\Big( \sum_{S \in \mathcal{P}(\{1,\ldots,k\}\backslash\{\pi(i)\})} \hat{f}'(S)\boldsymbol{\chi}_S + \sum_{S \in \overline{\mathcal{P}(\{1,\ldots,k\}\backslash\{\pi(i)\})}} \hat{f}'(S)\boldsymbol{\chi}_S \Big) \tag{153}$$

$$= \sum_{S \in \mathcal{P}(\{1,\ldots,k\}\backslash\{\pi(i)\})} \hat{f}'(\pi(S))\boldsymbol{\chi}_S + \sum_{S \in \overline{\mathcal{P}(\{1,\ldots,k\}\backslash\{\pi(i)\})}} \hat{f}'(\pi(S))\boldsymbol{\chi}_S \tag{154}$$

$$= \sum_{S \in \mathcal{P}(\{1,\ldots,k\}\backslash\{i\})} \hat{f}'(\pi(S))\boldsymbol{\chi}_S + \sum_{S \in \overline{\mathcal{P}(\{1,\ldots,k\}\backslash\{i\})}} \hat{f}'(\pi(S))\boldsymbol{\chi}_S \tag{155}$$

$$= \sum_{S \in \mathcal{P}(\{1,\ldots,k\}\backslash\{i\})} (\hat{f}(\pi(S)) + \hat{f}(\pi(S) \cup \{\pi(i)\}))\boldsymbol{\chi}_S + \sum_{S \in \overline{\mathcal{P}(\{1,\ldots,k\}\backslash\{i\})}} (-\hat{f}(\pi(S)))\boldsymbol{\chi}_S \tag{156}$$

Equation 156 holds since $i \notin S \implies \pi(i) \notin \pi(S)$ and $i \in S \implies \pi(i) \in \pi(S)$. $\qquad \square$

**Theorem D.1** (Inverse NPN transformation). *Let* $\tau = \tau_\nu \circ \tau_\pi \circ \tau_\phi$ *be an NPN transformation. Then* $\tau^{-1} = \tau_\nu \circ \tau_{\pi^{-1}} \circ \tau_{\pi(\phi)}$.

*Proof.*

$$\tau^{-1} = \tau_\phi^{-1} \circ \tau_\pi^{-1} \circ \tau_\nu^{-1} \tag{157}$$

$$= \tau_\phi \circ \tau_{\pi^{-1}} \circ \tau_\nu \qquad \text{(Boolean algebra)} \tag{158}$$

$$= \tau_\phi \circ \tau_\nu \circ \tau_{\pi^{-1}} \qquad \text{(Lemma D.5)} \tag{159}$$

$$= \tau_\nu \circ \tau_\phi \circ \tau_{\pi^{-1}} \qquad \text{(Lemma D.6)} \tag{160}$$

$$= \tau_\nu \circ \tau_{\pi^{-1}} \circ \tau_{\pi(\phi)} \qquad \text{(Lemma D.7)} \tag{161}$$

$\qquad \square$

## D.2 RELATING NPN AND $\sigma$ EQUIVALENCE CLASSES

**Lemma D.8.** *Let $p, q \in \mathbb{Z}^{2^k}$ be MPs of BFs, and $p' \in [p]_\sigma$. Then*

$$\forall x \in \mathbb{B}^k \Big( q(x) \cdot \sigma\big(p'(x)\big) = \sigma\big(q(x) \cdot p'(x)\big) \Big). \tag{162}$$

*Proof.* If $q(x) = 0$, then $0 = 0$. If $q(x) = 1$, then $\sigma\big(p'(x)\big) = \sigma\big(p'(x)\big)$. $\qquad\square$

**Lemma D.9.** *Let $p, q, r \in \mathbb{Z}^{2^k}$ be MPs of BFs, $p' \in [p]_\sigma$ and $q' \in [q]_\sigma$. Then*

$$\forall x \in \mathbb{B}^k \Big( \sigma\big(r(x) \cdot p'(x)\big) + \sigma\big(\bar{r}(x) \cdot q'(x)\big) = \sigma\big(r(x) \cdot p'(x) + \bar{r}(x) \cdot q'(x)\big) \Big), \tag{163}$$

*where $\bar{r}(x) = 1 - r(x)$.*

*Proof.* The expression is akin to a 2-1 multiplexer, where $r$ acts as a select line. If $r(x) = 0$, then $\sigma\big(q'(x)\big) = \sigma\big(q'(x)\big)$. If $r(x) = 1$, then $\sigma\big(p'(x)\big) = \sigma\big(p'(x)\big)$. $\qquad\square$

**Lemma D.10** (Relating $\sim_\sigma$ and $\tau_{\phi_i}$). *Let $p, q \in \mathbb{Z}^{2^k}$ be MPs of BFs such that $p = \tau_{\phi_i}(q)$, where $\phi_i = \{i\} \subseteq \{1, \ldots, k\}$, and $q' \in [q]_\sigma$. Then $\tau_{\phi_i}(q') \in [p]_\sigma$.*

*Proof.*

$$p = \tau_{\phi_i}(q) \Rightarrow \forall x \in \mathbb{B}^k \Big( p(x) = (1 - x_i)q_{x_i}(x) + x_i q_{\bar{x}_i}(x) \Big) \tag{164}$$

$$\Rightarrow \forall x \in \mathbb{B}^k \Big( p(x) = (1 - x_i)\sigma\big(q'_{x_i}(x)\big) + x_i \sigma\big(q'_{\bar{x}_i}(x)\big) \Big) \qquad (q' \in [q]_\sigma) \tag{165}$$

$$\Rightarrow \forall x \in \mathbb{B}^k \Big( p(x) = \sigma\big((1 - x_i)q'_{x_i}(x)\big) + \sigma\big(x_i q'_{\bar{x}_i}(x)\big) \Big) \qquad (\text{Lemma D.8}) \tag{166}$$

$$\Rightarrow \forall x \in \mathbb{B}^k \Big( p(x) = \sigma\big((1 - x_i)q'_{x_i}(x) + x_i q'_{\bar{x}_i}(x)\big) \Big) \qquad (\text{Lemma D.9}) \tag{167}$$

$$\Rightarrow \forall x \in \mathbb{B}^k \Big( \sigma\big(p(x)\big) = \sigma\big((1 - x_i)q'_{x_i}(x) + x_i q'_{\bar{x}_i}(x)\big) \Big) \tag{168}$$

$$\Rightarrow p \sim_\sigma \tau_{\phi_i}(q') \tag{169}$$

$\qquad\square$

**Lemma D.11** (Relating $\sim_\sigma$ and $\tau_\pi$). *Let $p, q \in \mathbb{Z}^{2^k}$ be MPs of BFs such that $p = \tau_\pi(q)$, where $\pi : \{1, \ldots, k\} \to \{1, \ldots, k\}$ is a bijection, and $q' \in [q]_\sigma$. Then $\tau_\pi(q') \in [p]_\sigma$.*

*Proof.*

$$p = \tau_\pi(q) \Rightarrow \forall x \in \mathbb{B}^k \Big( p(x) = q(\tau_\pi(x)) \Big) \tag{170}$$

$$\Rightarrow \forall x \in \mathbb{B}^k \Big( p(x) = \sigma\big(q'(\tau_\pi(x))\big) \Big) \qquad (q' \in [q]_\sigma) \tag{171}$$

$$\Rightarrow \forall x \in \mathbb{B}^k \Big( \sigma\big(p(x)\big) = \sigma\big(q'(\tau_\pi(x))\big) \Big) \tag{172}$$

$$\Rightarrow p \sim_\sigma \tau_\pi(q') \tag{173}$$

$\qquad\square$

**Lemma D.12** (Relating $\sim_\sigma$ and $\tau_\nu$). *Let $\boldsymbol{p}, \boldsymbol{q} \in \mathbb{Z}^{2^k}$ be MPs of BFs such that $\boldsymbol{p} = \tau_\nu(\boldsymbol{q})$, where $\nu = 1$, and $\boldsymbol{q}' \in [\boldsymbol{q}]_\sigma$. Then $\tau_\nu(\boldsymbol{q}') \in [\boldsymbol{p}]_\sigma$.*

*Proof.*

$$\boldsymbol{p} = \tau_\nu(\boldsymbol{q}) \Rightarrow \forall \boldsymbol{x} \in \mathbb{B}^k \Big( p(\boldsymbol{x}) = 1 - q(\boldsymbol{x}) \Big) \tag{174}$$

$$\Rightarrow \forall \boldsymbol{x} \in \mathbb{B}^k \Big( p(\boldsymbol{x}) = 1 - \sigma\big(q'(\boldsymbol{x})\big) \Big) \qquad (\boldsymbol{q}' \in [\boldsymbol{q}]_\sigma) \tag{175}$$

$$\Rightarrow \forall \boldsymbol{x} \in \mathbb{B}^k \Big( p(\boldsymbol{x}) = \sigma\big(1 - q'(\boldsymbol{x})\big) \Big) \tag{176}$$

$$\Rightarrow \forall \boldsymbol{x} \in \mathbb{B}^k \Big( \sigma\big(p(\boldsymbol{x})\big) = \sigma\big(1 - q'(\boldsymbol{x})\big) \Big) \tag{177}$$

$$\Rightarrow \boldsymbol{p} \sim_\sigma \tau_\nu(\boldsymbol{q}') \tag{178}$$

To prove Equation 176 holds, it suffices to consider the two cases where $q'(\boldsymbol{x}) > 0$ and $q'(\boldsymbol{x}) \leq 0$. If $q'(\boldsymbol{x}) > 0$, then $q(\boldsymbol{x}) = 1$ and $p(\boldsymbol{x}) = 0$. Since $q'(\boldsymbol{x}) \in \mathbb{Z}$, $1 - q'(\boldsymbol{x}) \leq 0$ and hence $\sigma\big(1 - q'(\boldsymbol{x})\big) = 0 = p(\boldsymbol{x})$. If $q'(\boldsymbol{x}) \leq 0$, then $q(\boldsymbol{x}) = 0$ and $p(\boldsymbol{x}) = 1$. Since $q'(\boldsymbol{x}) \in \mathbb{Z}$, $1 - q'(\boldsymbol{x}) > 0$ and hence $\sigma\big(1 - q'(\boldsymbol{x})\big) = 1 = p(\boldsymbol{x})$. $\qquad\square$

**Theorem D.2** (NPN transformations between $\sigma$ equivalence classes). *Let $\boldsymbol{p}, \boldsymbol{q} \in \mathbb{Z}^{2^k}$ be MPs of BFs such that $\boldsymbol{p} = \tau(\boldsymbol{q})$, where $\tau = \tau_\nu \circ \tau_\pi \circ \tau_\phi$ is an NPN transformation, and $\boldsymbol{q}' \in [\boldsymbol{q}]_\sigma$. Then $\tau(\boldsymbol{q}') \in [\boldsymbol{p}]_\sigma$.*

*Proof.* We begin by noting that $\tau_\phi = \tau_{\phi_1} \circ \cdots \circ \tau_{\phi_k}$, where $\tau_{\phi_i} = \tau_{\{i\}}$ if $i \in \phi$, and $\tau_{\phi_i} = Id.$ if $i \notin \phi$. Let $\boldsymbol{q}_1 = \tau_\phi(\boldsymbol{q})$, $\boldsymbol{q}_2 = \tau_\pi(\boldsymbol{q}_1)$, and $\boldsymbol{q}_3 = \tau_\nu(\boldsymbol{q}_2)$. Then

$$\boldsymbol{q}_1 \sim_\sigma \tau_\phi(\boldsymbol{q}') = \boldsymbol{q}_1', \qquad\qquad \text{(Lemma D.10)} \tag{179}$$

$$\boldsymbol{q}_2 \sim_\sigma \tau_\pi(\boldsymbol{q}_1') = \tau_\pi(\tau_\phi(\boldsymbol{q}')) = \boldsymbol{q}_2', \qquad \text{(Lemma D.11)} \tag{180}$$

$$\boldsymbol{q}_3 \sim_\sigma \tau_\nu(\boldsymbol{q}_2') = \tau_\nu(\tau_\pi(\tau_\phi(\boldsymbol{q}'))) = \boldsymbol{q}_3'. \qquad \text{(Lemma D.12)} \tag{181}$$

Since, $\boldsymbol{p} = \boldsymbol{q}_3$, $\boldsymbol{p} \sim_\sigma (\tau_\nu \circ \tau_\pi \circ \tau_\phi)(\boldsymbol{q}') = \tau(\boldsymbol{q}')$. $\qquad\square$

### D.3 COMPUTING MMPs WITH NPN TRANSFORMATIONS

**Definition D.5** (PN-invariant criterion). *Let $p \in \mathbb{Z}^{2^k}$ be an MP, $\nu \in \mathbb{B}$ and $\pi : \{1, \ldots, k\} \to \{1, \ldots, k\}$ be a bijection. A criterion vector $c \in \mathbb{Z}^{2^k}$ is PN-invariant if*

$$\|c \odot p\|_1 = \|c \odot (\tau_\nu \circ \tau_\pi)(p)\|_1. \tag{182}$$

**Lemma D.13** (Uniform criterion is PN-invariant). *Let $k \in \mathbb{N}_0$. $c_k^{uni}$ is PN-invariant.*

*Proof.*

$$\|c_k^{uni} \odot p\|_1 = \left\| c_k^{uni} \odot \left( \hat{f}(\emptyset)\chi_\emptyset + \sum_{S \in \mathcal{P}(\{1,\ldots,k\})\setminus\{\emptyset\}} \hat{f}(S)\chi_S \right) \right\|_1 \tag{183}$$

$$= \left\| \sum_{S \in \mathcal{P}(\{1,\ldots,k\})\setminus\{\emptyset\}} \hat{f}(S)\chi_S \right\|_1 \tag{184}$$

$$= \sum_{S \in \mathcal{P}(\{1,\ldots,k\})\setminus\{\emptyset\}} |\hat{f}(S)| \tag{185}$$

Let $\nu = 1$ (a similar argument can be used to prove the case where $\nu = 0$, and hence $\tau_\nu = Id.$).

$$\|c_k^{uni} \odot (\tau_\nu \circ \tau_\pi)(p)\|_1$$

$$= \left\| c_k^{uni} \odot \left( (1 - \hat{f}(\emptyset))\chi_\emptyset + \sum_{S \in \mathcal{P}(\{1,\ldots,k\})\setminus\{\emptyset\}} (-\hat{f}(\pi^{-1}(S)))\chi_S \right) \right\|_1 \tag{186}$$

$$= \left\| \sum_{S \in \mathcal{P}(\{1,\ldots,k\})\setminus\{\emptyset\}} (-\hat{f}(\pi^{-1}(S)))\chi_S \right\|_1 \tag{187}$$

$$= \sum_{S \in \mathcal{P}(\{1,\ldots,k\})\setminus\{\emptyset\}} |-\hat{f}(\pi^{-1}(S))| \tag{188}$$

$$= \sum_{S \in \mathcal{P}(\{1,\ldots,k\})\setminus\{\emptyset\}} |\hat{f}(S)| \tag{189}$$

$\square$

**Lemma D.14** (Degree criterion is PN-invariant). *Let $k \in \mathbb{N}_0$. $c_k^{deg}$ is PN-invariant.*

*Proof.*

$$\|c_k^{deg} \odot p\|_1 = \left\| c_k^{deg} \odot \left( \hat{f}(\emptyset)\chi_\emptyset + \sum_{S \in \mathcal{P}(\{1,\ldots,k\})\setminus\{\emptyset\}} \hat{f}(S)\chi_S \right) \right\|_1 \tag{190}$$

$$= \left\| \sum_{S \in \mathcal{P}(\{1,\ldots,k\})\setminus\{\emptyset\}} (|S| \cdot \hat{f}(S))\chi_S \right\|_1 \tag{191}$$

$$= \sum_{S \in \mathcal{P}(\{1,\ldots,k\})\setminus\{\emptyset\}} ||S| \cdot \hat{f}(S)| \tag{192}$$

Let $\nu = 1$ (a similar argument can be used to prove the case where $\nu = 0$, and hence $\tau_\nu = Id.$).

$$\|c_k^{deg} \odot (\tau_\nu \circ \tau_\pi)(p)\|_1$$

$$= \left\| c_k^{deg} \odot \left( (1 - \hat{f}(\emptyset))\chi_\emptyset + \sum_{S \in \mathcal{P}(\{1,\ldots,k\})\setminus\{\emptyset\}} (-\hat{f}(\pi^{-1}(S)))\chi_S \right) \right\|_1 \tag{193}$$

$$= \left\| \sum_{S \in \mathcal{P}(\{1,\ldots,k\})\setminus\{\emptyset\}} (-|S| \cdot \hat{f}(\pi^{-1}(S)))\chi_S \right\|_1 \tag{194}$$

$$= \sum_{S \in \mathcal{P}(\{1,\ldots,k\})\setminus\{\emptyset\}} |-|S| \cdot \hat{f}(\pi^{-1}(S))| \tag{195}$$

$$= \sum_{S \in \mathcal{P}(\{1,\ldots,k\})\setminus\{\emptyset\}} ||S| \cdot \hat{f}(S)| \tag{196}$$

$\square$

**Theorem D.3** (Computing MMPs with NPN transformations under PN-invariant criterions). *Let $c \in \mathbb{Z}^{2^k}$ be a PN-invariant criterion, $p, q \in \mathbb{Z}^{2^k}$ be MPs of BFs such that $p = (\tau_\nu \circ \tau_\pi)(q)$, and $q' \in [q]_\sigma$ be an MMP with respect to criterion $c$. Then $p' = (\tau_\nu \circ \tau_\pi)(q') \in [p]_\sigma$ is an MMP with respect to criterion $c$.*

*Proof.* If $q'$ is an MMP with respect to criterion $c$, then

$$q' \in \arg\min_{r \in [q]_\sigma} C(r) = \arg\min_{r \in [q]_\sigma} \|c \odot r\|_1. \tag{197}$$

Moreover, since $c$ is PN-invariant, we have

$$\|c \odot q'\|_1 = \|c \odot (\tau_\nu \circ \tau_\pi)(q')\|_1 = \|c \odot p'\|_1 = \alpha, \tag{198}$$

and by Theorem D.2 $p' \in [p]_\sigma$. We claim $p' \in \arg\min_{r \in [p]_\sigma} \|c \odot r\|_1$.

Suppose, for sake of contradiction, $p' \notin \arg\min_{r \in [p]_\sigma} \|c \odot r\|_1$, then there exists $\hat{p} \in [p]_\sigma$ such that $\|c \odot \hat{p}\|_1 < \alpha$. Let $\hat{q} = (\tau_\pi^{-1} \circ \tau_\nu^{-1})(\hat{p}) = (\tau_\nu \circ \tau_{\pi^{-1}})(\hat{p})$, where the second equality holds due to Theorem D.1. Following Theorem D.2, $\hat{q} \in [q]_\sigma$, and since $c$ is PN-invariant, we have

$$\|c \odot \hat{p}\|_1 = \|c \odot (\tau_\nu \circ \tau_{\pi^{-1}})(\hat{p})\|_1 = \|c \odot \hat{q}\|_1 < \alpha, \tag{199}$$

which is a contradiction, as we assumed $q' \in \arg\min_{r \in [q]_\sigma} \|c \odot r\|_1$. $\qquad\square$

---

**Algorithm 1:** MP- and MMP-based technology mapping with NPN classes

---

**Input:** BN $G_{\mathcal{F}} = (V, E)$ and vertex labeling function $\psi_{\mathcal{F}} : V \to \mathcal{F}$, where $\mathcal{F} = \cup_{k=1}^{K} \mathbb{B}^{2^k}$

**Output:** Functions $\psi_{\mathcal{F}'}^{\boldsymbol{p}}, \psi_{\mathcal{F}'}^{\boldsymbol{p}'} : V \to \mathcal{F}'$, where $\mathcal{F}' = \cup_{k=1}^{K} \mathbb{Z}^{2^k}$

**Function :** techMapNPN($G_{\mathcal{F}}$, $\psi_{\mathcal{F}}$)

1   Initialize $\psi_{\mathcal{F}'}^{\boldsymbol{p}} \leftarrow \emptyset$ and $\psi_{\mathcal{F}'}^{\boldsymbol{p}'} \leftarrow \emptyset$

2   **for** $k \leftarrow 0$ **to** $K$ **do**

3     $C \leftarrow$ npnClassify($G_{\mathcal{F}}$, $\psi_{\mathcal{F}}$, $k$)            // Compute NPN classes

4     **for** *each NPN class* $C_{\kappa_{\boldsymbol{f}}} = \{(\boldsymbol{f}_i, \tau_i^{-1}) \mid \boldsymbol{f}_i = \tau_i^{-1}(\kappa_{\boldsymbol{f}})\} \in C$ **do**

5       $\kappa_{\boldsymbol{p}} \leftarrow$ ttToMP($\kappa_{\boldsymbol{f}}$)        // Compute MP of NPN canonical form

6       **for** *each unique permuted phase assignment subset*
       $S_{\pi(\phi)} = \{(\boldsymbol{f}, \tau^{-1}) \in C_{\kappa_{\boldsymbol{f}}} \mid \tau^{-1} = \tau_{\nu} \circ \tau_{\pi^{-1}} \circ \tau_{\pi(\phi)}\}$ **do**

7         **if** $|S_{\pi(\phi)}| = |\{(\boldsymbol{f}, \tau^{-1})\}| = 1$ **then**

8           $\boldsymbol{p} \leftarrow \tau^{-1}(\kappa_{\boldsymbol{p}})$

9           $\boldsymbol{p}' \leftarrow$ mpToMMP($\boldsymbol{p}$)                // Compute MMP of $\boldsymbol{p}$

10           $\psi_{\mathcal{F}'}^{\boldsymbol{p}} \leftarrow \psi_{\mathcal{F}'}^{\boldsymbol{p}} \cup \{(v, \boldsymbol{p}) \in V \times \mathbb{Z}^{2^k} \mid \psi_{\mathcal{F}}(v) = \boldsymbol{f}\}$

11           $\psi_{\mathcal{F}'}^{\boldsymbol{p}'} \leftarrow \psi_{\mathcal{F}'}^{\boldsymbol{p}'} \cup \{(v, \boldsymbol{p}') \in V \times \mathbb{Z}^{2^k} \mid \psi_{\mathcal{F}}(v) = \boldsymbol{f}\}$

12         **else**

13           $\kappa_{\boldsymbol{p}}^{\tau_{\pi(\phi)}} \leftarrow \tau_{\pi(\phi)}(\kappa_{\boldsymbol{p}})$

14           $\kappa_{\boldsymbol{p}'}^{\tau_{\pi(\phi)}} \leftarrow$ mpToMMP($\kappa_{\boldsymbol{p}}^{\tau_{\pi(\phi)}}$) // Compute MMP once for $\boldsymbol{p}$ sharing $\tau_{\pi(\phi)}$

15           **for** $(\boldsymbol{f}, \tau^{-1}) \in S_{\pi(\phi)}$ **do**

16             $\boldsymbol{p} \leftarrow (\tau_{\nu} \circ \tau_{\pi^{-1}})(\kappa_{\boldsymbol{p}}^{\tau_{\pi(\phi)}})$

17             $\boldsymbol{p}' \leftarrow (\tau_{\nu} \circ \tau_{\pi^{-1}})(\kappa_{\boldsymbol{p}'}^{\tau_{\pi(\phi)}})$

18             $\psi_{\mathcal{F}'}^{\boldsymbol{p}} \leftarrow \psi_{\mathcal{F}'}^{\boldsymbol{p}} \cup \{(v, \boldsymbol{p}) \in V \times \mathbb{Z}^{2^k} \mid \psi_{\mathcal{F}}(v) = \boldsymbol{f}\}$

19             $\psi_{\mathcal{F}'}^{\boldsymbol{p}'} \leftarrow \psi_{\mathcal{F}'}^{\boldsymbol{p}'} \cup \{(v, \boldsymbol{p}') \in V \times \mathbb{Z}^{2^k} \mid \psi_{\mathcal{F}}(v) = \boldsymbol{f}\}$

20           **end**

21         **end**

22       **end**

23     **end**

24   **end**

25   **return** $\psi_{\mathcal{F}'}^{\boldsymbol{p}}, \psi_{\mathcal{F}'}^{\boldsymbol{p}'}$

---

## D.4   OBJECTIVE-AWARE OPTIMIZATION ALGORITHM USING NPN CLASSES

We propose an optimization-objective-aware NPN classification algorithm for MP- and MMP-based technology mapping. The algorithm firstly classifies functions into their respective NPN equivalence classes by computing canonical forms and corresponding NPN transformations. The canonical forms are then mapped to their MP representations. Next, each equivalence class is partitioned into subsets that share permuted phase assignments $\tau_{\pi(\phi)}$. The "permuted phase assignment NPN canonical forms" $\kappa_{\boldsymbol{p}}^{\tau_{\pi(\phi)}}$ of each subset are then minimized with respect to a PN-invariant criterion. Finally, the MPs and MMPs of other members of the subset are computed following Theorem D.3. See Algorithm 1.

*Remark* D.6 (Correctness for Algorithm 1). The ordering of the cache hierarchy, and specifically the order in which the inverse NPN transformation is computed, is correct following Theorem D.1. Furthermore, the statement that all MMPs are indeed MMPs according to criterion $\boldsymbol{c}^{uni}$ or $\boldsymbol{c}^{deg}$ follows from Lemma D.13, Lemma D.14, and Theorem D.3.

*Remark* D.7 (Time complexity of NPN classification). Let $m$ denote the number of unique $k$-input BFs being classified. The NPN classification algorithm we used for our experimental results to compute NPN canonical forms and transformations requires time $O(m2^k k) + e$, where $O(m2^k k)$ is due to computing a signature vector for $m$ unique functions, and $e$ denotes exhaustive enumeration time (which requires checking at most $2^{k+1}k!$ NPN transformations per unique BF) (Zhou et al., 2020). Letting $e = O(m2^{k+1}k!)$, we have time $O(m2^k k + m2^{k+1}k!)$. We note that asymptotic analysis

does not provide significant insight here, as for practical reasons of synthesis $K \lesssim 10$. Heuristics, symmetries, low-level implementations, and multi-level caches (Petkovska et al., 2016; Zhou et al., 2020) make NPN classification relatively fast in comparison to the aggregate collection of ttToMP and mpToMMP subproblems we aim to solve.

*Remark* D.8 (Time complexity of Algorithm 1). Let $G_{\mathcal{F}_f} = (V, E)$ and $\psi_{\mathcal{F}_f} : V \to \mathcal{F}_f$, where $\mathcal{F}_f = \cup_{k=1}^{K} \mathbb{B}^{2^k}$, be the $K$-LUT Boolean network (BN) and vertex labeling function provided as input to Algorithm 1, respectively. Let $V_k = \{u \in V \mid indegree(u) = k\}$ be the set of vertices with $k$ inputs in $G_{\mathcal{F}_f}$, $\psi_{\mathcal{F}_f}[V_k] = \left\{ \boldsymbol{g} \in \mathbb{B}^{2^k} \mid \exists v \in V_k \left( \psi_{\mathcal{F}_f}(v) = \boldsymbol{g} \right) \right\}$ be the set of $k$-input BFs in $G_{\mathcal{F}_f}$, and $\kappa\left[\psi_{\mathcal{F}_f}[V_k]\right] = \left\{ \boldsymbol{g} \in \mathbb{B}^{2^k} \mid \exists \boldsymbol{h} \in \psi_{\mathcal{F}_f}[V_k] \left( \kappa(\boldsymbol{h}) = \boldsymbol{g} \right) \right\}$ be the set of $k$-input NPN canonical forms in $G_{\mathcal{F}_f}$.

We now define the quantities of interest for our analysis of time complexity. Let

- $v = |V_k| \in \mathbb{N}_0$ denote the number of $k$-input BFs in $G_{\mathcal{F}_f}$,
- $m = \left|\psi_{\mathcal{F}_f}[V_k]\right| \in \mathbb{N}_0$ denote the number of *unique* $k$-input BFs in $G_{\mathcal{F}_f}$,
- $c = \left|\kappa\left[\psi_{\mathcal{F}_f}[V_k]\right]\right| \in \mathbb{N}_0$ denote the number of NPN equivalence classes on the set of $k$-input BFs in $G_{\mathcal{F}_f}$, and
- $s \in \mathbb{N}_0$ denote the number of permuted phase assignment subsets on the set of $k$-input BFs in $G_{\mathcal{F}_f}$.

Next, we determine bounds on these quantities.

- $0 \le v$, as $v$ does not have an upper bound.
- $0 \le m \le 2^{2^k}$, as there are $2^{2^k}$ $k$-input BFs.
- $\frac{m}{2^{k+1}k!} \le c \le m$. The upper bound occurs when all $m$ unique BFs fall into different NPN equivalence classes. The lower bound requires some analysis. Recall from Remark D.2, there are $2^{k+1}k!$ NPN transformations on $k$-input BFs. Moreover, let $\boldsymbol{f} \in \psi_{\mathcal{F}_f}[V_k]$. Now, suppose that applying the $2^{k+1}k!$ NPN transformations on $\boldsymbol{f}$ returns $2^{k+1}k!$ unique BFs. Then the size of the NPN equivalence class of $\boldsymbol{f}$ is $|[\boldsymbol{f}]_{\text{NPN}}| = 2^{k+1}k!$. Hence, if $[\boldsymbol{f}]_{\text{NPN}} \subseteq \psi_{\mathcal{F}_f}[V_k]$, the NPN class accounts for $2^{k+1}k!$ of the $m$ unique BFs in $\psi_{\mathcal{F}_f}[V_k]$. Consequently, if this happens for all BFs in $\psi_{\mathcal{F}_f}[V_k]$, then the number of NPN equivalence classes on $\psi_{\mathcal{F}_f}[V_k]$ would be $\frac{m}{2^{k+1}k!}$.
- $\frac{m}{2k!} \le s \le m$ and $c \le s$. These bounds require some analysis. Let $C_{\kappa_f} = \{(\boldsymbol{f}_i, \tau_i^{-1}) \mid \boldsymbol{f}_i = \tau_i^{-1}(\kappa_{\boldsymbol{f}})\}$, where $\boldsymbol{f}_i \in \psi_{\mathcal{F}_f}[V_k]$, be an NPN equivalence class with NPN canonical form $\kappa_{\boldsymbol{f}}$. Then a permuted phase assignment subset $S_{\pi(\phi)} = \{(\boldsymbol{f}, \tau^{-1}) \in C_{\kappa_f} \mid \tau^{-1} = \tau_\nu \circ \tau_{\pi^{-1}} \circ \tau_{\pi(\phi)}\}$ is a subset of the NPN equivalence class that shares the same permuted phase assignment $\pi(\phi) \subseteq \{1, \ldots, k\}$. Notice, that there are $2^k$ possible permuted phase assignments. For the lower bound $\frac{m}{2k!} \le s$, suppose $|C_{\kappa_f}| = 2^{k+1}k!$. Then the NPN class will be partitioned into $2^k$ permuted phase assignment subsets, each of size $\frac{2^{k+1}k!}{2^k} = 2k!$. Consequently, if this happens for all BFs in $\psi_{\mathcal{F}_f}[V_k]$, then the number of permuted phase assignment subsets on $\psi_{\mathcal{F}_f}[V_k]$ would be $\frac{m}{2k!}$. For the upper bound $s \le m$, this occurs when all $m$ unique BFs fall into different permuted phase assignment subsets. Finally, since permuted phase assignment subsets are subsets of NPN equivalence classes, we have that $c \le s$.

Before continuing with the time complexity, we note some corollaries of the analysis performed. Note that all BFs in an NPN equivalence class share an NPN canonical form $\kappa_{\boldsymbol{p}}$, and that for all functions in the class Algorithm 1 computes ttToMP only once. Hence, since an NPN equivalence class can have up to $2^{k+1}k!$ members, Algorithm 1 will reduce the number of calls to ttToMP by up to $2^{k+1}k! - 1$ for the class.

Similarly, note that all BFs in a permuted phase assignment subset of an NPN class share a "permuted phase assignment NPN canonical form" $\kappa_{\boldsymbol{p}}^{\tau_{\pi(\phi)}} = \tau_{\pi(\phi)}(\kappa_{\boldsymbol{p}})$, and that for all functions in the subset Algorithm 1 computes mpToMMP only once. Hence, since permuted phase assignment subsets can have up to $2k!$ members, Algorithm 1 will reduce the number of calls to mpToMMP by up to $2k! - 1$ for the subset.

We now continue with the time complexity of Algorithm 1. Firstly, the algorithm computes NPN canonical forms and transformations for each of the $m$ functions in $\psi_{\mathcal{F}_f}[V_k]$, requiring time $O(m2^k k+$

$m2^{k+1}k!$) (see Remark D.7). A cache key (the hexadecimal string representation of a function's truth table) must then be computed per vertex, requiring time $v \cdot O(2^k)$. $\mathtt{ttToMP}$ is then computed for each NPN canonical form, requiring time $c \cdot O(2^k k)$ (using the algorithm from Gavier et al. (2023)). Next, at most $k$ input negations $\tau_{\phi_i}$ are performed to compute each of the $s$ permuted phase assignment NPN canonical forms, requiring time $s \cdot k \cdot O(2^k)$ (see Remark D.5). $\mathtt{mpToMMP}$ is then computed for each of the $s$ permuted phase assignment NPN canonical forms, requiring time $s \cdot 2^{O(k2^k)}$ (see Remark C.2). Finally, for each of the $m$ unique BFs in $\psi_{\mathcal{F}_f}[V_k]$, an input permutation $\tau_\pi$ and output negation $\tau_\nu$ are computed, requiring time $m \cdot O(2^k)$ (see Remark D.5). The time complexity of Algorithm 1 for a $K$-LUT BN is therefore:

$$T_{\textit{NPN classes}}(K) = O(m2^K K + m2^{K+1}K!) + v \cdot O(2^K) +$$
$$+ c \cdot O(2^K K) + sK \cdot O(2^K) + s2^{O(K2^K)} + m \cdot O(2^K) \tag{200}$$
$$= O(m2^{K+1}K! + v2^K + c2^K K + s2^K K + m2^K) + s2^{O(K2^K)} \tag{201}$$
$$= O(m2^{K+1}K! + v2^K + c2^K K) + s2^{O(K2^K)} \tag{202}$$
$$= O((2mK! + v + cK)2^K) + s2^{O(K2^K)} \tag{203}$$
$$= O((v + mK! + cK)2^K) + s2^{O(K2^K)}. \tag{204}$$

*Remark* D.9 (Time complexity of the *Naive* algorithm from § 4.2). We follow the notation introduced in Remark D.8. The *Naive* algorithm computes $\mathtt{ttToMP}$ and $\mathtt{mpToMMP}$ per vertex in $V_k$, requiring time $v \cdot O(2^k k)$ (using the algorithm from Gavier et al. (2023)) and $v \cdot 2^{O(k2^k)}$ (see Remark C.2), respectively. Hence, the time complexity of the *Naive* algorithm for a $K$-LUT BN is:

$$T_{\textit{Naive}}(K) = v \cdot O(2^K K) + v2^{O(K2^K)} \tag{205}$$
$$= O((vK)2^K) + v2^{O(K2^K)} \tag{206}$$

*Remark* D.10 (Time complexity of the *Cached* algorithm from § 4.2). We follow the notation introduced in Remark D.8. The *Cached* algorithm computes a cache key (the hexadecimal string representation of a function's truth table) per vertex in $V_k$, requiring time $v \cdot O(2^k)$. It then computes $\mathtt{ttToMP}$ and $\mathtt{mpToMMP}$ per unique function in $\psi_{\mathcal{F}_f}[V_k]$, requiring time $m \cdot O(2^k k)$ (using the algorithm from Gavier et al. (2023)) and $m \cdot 2^{O(k2^k)}$ (see Remark C.2), respectively. Hence, the time complexity of the *Cached* algorithm for a $K$-LUT BN is:

$$T_{\textit{Cached}}(K) = v \cdot O(2^K) + m \cdot O(2^K K) + m2^{O(K2^K)} \tag{207}$$
$$= O(v2^K + m2^K K) + m2^{O(K2^K)} \tag{208}$$
$$= O((v + mK)2^K) + m2^{O(K2^K)} \tag{209}$$

*Remark* D.11 (Additional space used by the *Naive* and *Cached* algorithms from § 4.2 and Algorithm 1). We follow the notation introduced in Remark D.8. Firstly, note that storing a truth table requires space $O(2^k)$, and storing an MP/MMP with coefficients in $\{-2^{k-1}, \ldots, 2^{k-1}\}$ requires space $O(k2^k)$. Furthermore, storing $\tau_\pi$ requires space $O(k \log k)$ from storing a sequence of $k$ unique integers in $\{0, 1, \ldots, k-1\}$, storing $\tau_\phi$ requires space $O(k)$ from storing which inputs are to be negated, and storing $\tau_\nu$ requires space $O(1)$ from storing whether the output is to be negated. The *Naive* algorithm requires no additional space. The *Cached* algorithm requires $m \cdot O(K2^K)$ additional space for storing the MP/MMP cache of unique functions. Lastly, Algorithm 1 requires $m \cdot O(K^2)$ additional space for NPN classification, then $m \cdot O(K \log K) + c \cdot O(K2^K) + s \cdot O(K2^K)$ additional space for the NPN transformations of unique functions, the top-level cache of NPN canonical forms, and the second-level cache of permuted phase assignment subset NPN canonical forms, respectively.

*Remark* D.12 (Comparison of the *Naive* and *Cached* algorithms from § 4.2 and Algorithm 1). We follow the notation introduced in Remark D.8. For the set of $k$-input BFs of a $K$-LUT BN, *Naive* requires $v$ calls to $\mathtt{ttToMP}$ and $v$ calls to $\mathtt{mpToMMP}$. *Cached* requires $m \leq v$ calls to $\mathtt{ttToMP}$ and $m \leq v$ calls to $\mathtt{mpToMMP}$. Algorithm 1 requires $\frac{m}{2^{k+1}k!} \leq c \leq m$ calls to $\mathtt{ttToMP}$ and $\frac{m}{2k!} \leq s \leq m$ calls to $\mathtt{mpToMMP}$. For practical reasons of synthesis, $K \lesssim 10$ (as many operations and structures grow exponentially in time and space). Consequently, for fixed $K \lesssim 10$, as the number of vertices $v$ in a BN increases, we expect NPN classes and permuted phase assignment subsets to become more densely populated, meaning $s \to \frac{m}{2k!}$ and $c \to \frac{m}{2^{k+1}k!}$, increasing the benefits of Algorithm 1. See Appendix G.3 for an empirical analysis of the quantities $v$, $m$, $c$ and $s$.

### D.5 Relation to Existing NPN Classification Techniques and NPN Invariance

Negation-Permutation-Negation (NPN) classification forms a well-established and extensively studied subfield within the broader domain of Boolean function (BF) classification (Sasao & Butler, 2022). This interest in NPN classification is largely driven by its utility in areas such as logic synthesis, technology mapping, and logic verification (Hinsberger & Kolla, 1998). In modern logic synthesis tools, NPN classification is commonly used in rewriting algorithms, which optimize technology-independent representations of BNs (Mishchenko et al., 2006). Rewriting for And-Inverter Graphs (AIGs) typically involves precomputing a set of optimized structures for each $k$-input NPN canonical form and storing them within a hash lookup table. Doing this for $k \leq 4$ is common today and computationally tractable (Li et al., 2024). However, for $k = 5$ and $k = 6$ there are $666,126$ and $2.0 \times 10^{14}$ NPN equivalence classes, respectively, meaning both the time to compute and the space to store the optimized structures quickly becomes intractable (Sasao & Butler, 2022).

In this paper, we discuss the optimization of NN representations of BNs. It was noted by Gavier et al. (2023) that the number of layers in the *layer-merged* NN representation of a BN, following the technology mapping sequence reviewed in § 2.2 with $K$-LUT mapping, approximately depends on $\left(\log_2(K)\right)^{-1}$. Consequently, applications that are latency critical would aim to use the largest value of $K$. From our preceding discussion, it is evident that an optimization algorithm which precomputes and stores the MMP for each $k$-input NPN or permuted-phase-NPN canonical form, $k \leq K$, would be intractable both in time and space. For this reason, we could not utilize NPN classification in the same way as existing rewriting algorithms based on precomputed hash lookup tables.

Instead of precomputing and storing MMPs, our solution involves computing them on-the-fly. Indeed, the straightforward approach would be to compute them per unique BF in the $K$-LUT BN. However, our goal was to utilize NPN equivalence to reduce the number of times we would need to solve the MMP optimization problem. Based on the analysis in Remark D.8, it would be ideal if the MMP optimization problem could be solved once per NPN equivalence class (as opposed to once per permuted phase assignment subset of an NPN equivalence class, which is what is done in Algorithm 1). Following the results presented in Appendix D.3 and specifically the proof of Theorem D.3, this would seem to require that the criterions used in the MMP optimization problem are NPN-invariant. In Lemma D.13 and Lemma D.14, we respectively proved that the uniform criterion $c^{uni}$ and the degree criterion $c^{deg}$, the criterions we care about for compressing the NN representation of the BF, are PN-invariant. That is, they are invariant under input permutation and output negation transformations. Therefore, the final step to being NPN-invariant would be N-invariance, that is, invariance under input negation transformations. However, as the following lemmas prove, the criterions $c^{uni}$ and $c^{deg}$ that we are interested in are not N-invariant.

**Definition D.6** (N-invariant criterion). Let $p \in \mathbb{Z}^{2^k}$ be an MP and $\phi \subseteq \{1, \ldots, k\}$. A criterion vector $c \in \mathbb{Z}^{2^k}$ is N-invariant if

$$\|c \odot p\|_1 = \|c \odot \tau_\phi(p)\|_1. \tag{210}$$

**Lemma D.15** (Uniform criterion is not N-invariant). *Let $k \in \mathbb{N}_0$, $k \geq 2$. $c_k^{uni}$ is not N-invariant.*

*Proof.* Let $p = [1, -1, -1, 1, \mathbf{0}_{2^k-2^2}]^\mathsf{T} \in \mathbb{Z}^{2^k}$ and $\phi = \{1, 2\}$, where $\mathbf{0}_{2^k-2^2} \in \mathbb{Z}^{2^k-2^2}$ is the all zeros vector. Then $\tau_\phi(p) = [0, 0, 0, 1, \mathbf{0}_{2^k-2^2}]^\mathsf{T}$, and

$$\|c_k^{uni} \odot p\|_1 = 3 \neq 1 = \|c_k^{uni} \odot \tau_\phi(p)\|_1. \tag{211}$$

$\square$

**Lemma D.16** (Degree criterion is not N-invariant). *Let $k \in \mathbb{N}_0$, $k \geq 2$. $c_k^{deg}$ is not N-invariant.*

*Proof.* Let $p = [1, -1, -1, 1, \mathbf{0}_{2^k-2^2}]^\mathsf{T} \in \mathbb{Z}^{2^k}$ and $\phi = \{1, 2\}$, where $\mathbf{0}_{2^k-2^2} \in \mathbb{Z}^{2^k-2^2}$ is the all zeros vector. Then $\tau_\phi(p) = [0, 0, 0, 1, \mathbf{0}_{2^k-2^2}]^\mathsf{T}$, and

$$\|c_k^{deg} \odot p\|_1 = 4 \neq 2 = \|c_k^{deg} \odot \tau_\phi(p)\|_1. \tag{212}$$

$\square$

Table 4: $c_3^{deg}$-weighted one-norms for MPs in $\mathbb{Z}^{2^3}$, where $\phi = \{1, 2, 3\}$ and $\tau_\phi(p_{0\texttt{x}16}) = p_{0\texttt{x}68}$.

| | $\boldsymbol{p}_{0\texttt{x}16}$ | $\boldsymbol{p}'_{0\texttt{x}16}$ | $\boldsymbol{p}_{0\texttt{x}68}$ | $\boldsymbol{p}'_{0\texttt{x}68}$ | $\tau_\phi(\boldsymbol{p}'_{0\texttt{x}16})$ |
|---|---|---|---|---|---|
| $\|\cdot\|_{1,\boldsymbol{c}_3^{deg}}$ | 24 | 15 | 15 | 9 | 21 |

This is why Algorithm 1 computes MMPs per permuted phase assignment NPN canonical form, since all members of the permuted phase assignment subset are related by input permutation and output negation transformations, which do not change the MMP solution cost under the PN-invariant criterions $c^{uni}$ and $c^{deg}$. To make these ideas concrete, we conclude this discussion with an example of what can go wrong when trying to share MMP solutions w.r.t. non-N-invariant criterions among functions that are related by input negation transformations.

Let $f_{0\texttt{x}16} : \mathbb{B}^3 \to \mathbb{B}$ be the BF with truth table $0\texttt{x}16$. The MP representation of $f_{0\texttt{x}16}$ is

$$p_{0\texttt{x}16}(x_1, x_2, x_3) = x_1 + x_2 + x_3 - 2x_1x_2 - 2x_1x_3 - 2x_2x_3 + 3x_1x_2x_3. \tag{213}$$

An MMP representation of $f_{0\texttt{x}16}$ w.r.t. $c_3^{deg}$ is

$$p'_{0\texttt{x}16}(x_1, x_2, x_3) = x_1 + x_2 + x_3 - 2x_1x_2 - 2x_1x_3 - 2x_2x_3. \tag{214}$$

Let $f_{0\texttt{x}68} : \mathbb{B}^3 \to \mathbb{B}$ be the BF with truth table $0\texttt{x}68$. The MP representation of $f_{0\texttt{x}68}$ is

$$p_{0\texttt{x}68}(x_1, x_2, x_3) = x_1x_2 + x_1x_3 + x_2x_3 - 3x_1x_2x_3. \tag{215}$$

An MMP representation of $f_{0\texttt{x}68}$ w.r.t. $c_3^{deg}$ is

$$p'_{0\texttt{x}68}(x_1, x_2, x_3) = -1 + x_1 + x_2 + x_3 - 2x_1x_2x_3. \tag{216}$$

Let $\phi = \{1, 2, 3\}$ and consider the input negation transformation $\tau_\phi$. Firstly, notice that $(p_{0\texttt{x}16} \circ \tau_\phi)(x_1, x_2, x_3) = p_{0\texttt{x}68}(x_1, x_2, x_3)$. Next, we consider applying $\tau_\phi$ to the MMP $p'_{0\texttt{x}16}(x_1, x_2, x_3)$,

$$(p'_{0\texttt{x}16} \circ \tau_\phi)(x_1, x_2, x_3) = -3 + 3x_1 + 3x_2 + 3x_3 - 2x_1x_2 - 2x_1x_3 - 2x_2x_3. \tag{217}$$

Following Theorem D.2, $\tau_\phi(p'_{0\texttt{x}16}) \in [p_{0\texttt{x}68}]_\sigma$. Therefore, $\tau_\phi(p'_{0\texttt{x}16})$ *could* be an MMP of $p_{0\texttt{x}68}$ w.r.t. $c_3^{deg}$. However, as Table 4 shows, $\tau_\phi(p'_{0\texttt{x}16})$ does not have minimum solution cost within $[p_{0\texttt{x}68}]_\sigma$. Indeed, its cost under $\|\cdot\|_{1,c_3^{deg}}$ is greater than both the found MMP $p'_{0\texttt{x}68}$ and the original MP $p_{0\texttt{x}68}$. Consequently, after mapping from the MMP solution $p'_{0\texttt{x}16}$ using $\tau_\phi$, mpToMMP would need to be called again to find an MMP of $p_{0\texttt{x}68}$. However, this defeats the purpose of using NPN equivalence since we do not save a call to the computationally expensive optimization procedure of mpToMMP.

# E ARCHITECTURE-AWARE LOSSLESS OPTIMIZATION

**Definition E.1** (Boolean network). A Boolean network (BN) $G_{\mathcal{F}} = (V, E)$ is a directed-acyclic graph (DAG), with vertex labeling function $\psi : V \to \mathcal{F}$, that represents an $n$-input, $m$-output Boolean function (BF) $\mathcal{B} : \mathbb{B}^n \to \mathbb{B}^m$. Vertices with indegree zero are *primary inputs* (PI), and vertices with outdegree zero are *primary outputs* (PO). The labeling function $\psi$ assigns a single-output BF to each non-PI vertex in the graph. A directed edge $(u, v) \in E$ exists if the output of $u$'s BF is an input to $v$'s BF. The depth of vertex $v \in V$, denoted $depth(v) \in \mathbb{N}_0$, is the length of the longest path from any PI to $v$. The depth of a BN, denoted $depth(G_{\mathcal{F}})$, is the maximum vertex depth in the BN.

*Remark* E.1 (Types of NN optimization algorithms). Let $G_{\mathcal{F}} = (V, E)$ be a BN and $V_{\text{PI}} \subseteq V$ denote the set of PIs. NN optimization algorithms select a subset of vertices $\mathcal{M} \subseteq V \setminus V_{\text{PI}}$ to be represented as MMPs in the synthesized NN. We consider three types of NN optimization algorithms.

1. `optNone`: no vertices in the BN are represented as MMPs ($\mathcal{M} = \emptyset$). This results in no compression and corresponds to the SOTA NN-based technology mapping solution of Gavier et al. (2023).

2. `optAll`: all vertices in the BN are represented as MMPs ($\mathcal{M} = V \setminus V_{\text{PI}}$).

3. `optMaintainDepth`: a subset of vertices are selected to be represented as MMPs such that the depth of the resulting *layer-merged* NN (see § 2.2) does not increase ($\mathcal{M} \subseteq V \setminus V_{\text{PI}}$).

We propose a greedy algorithm for `optMaintainDepth` that runs in time $O(V(V + E))$.

**Definition E.2** (Path). A path of length $n$, denoted $P_n$, in BN $G_{\mathcal{F}} = (V, E)$ is a finite sequence of vertices $P_n = (v_0, v_1, \ldots, v_n) \in V^{n+1}$, where $\forall i \in \{0, \ldots, n-1\}\big((v_i, v_{i+1}) \in E\big)$.

**Definition E.3** (Leeway). The leeway of vertex $v \in V$, denoted $leeway(v) \in \mathbb{N}_0$, in BN $G_{\mathcal{F}} = (V, E)$ is defined as

$$leeway(v) = depth(G_{\mathcal{F}}) - depth(v) - lpl(v), \tag{218}$$

where $lpl(v)$ denotes the length of the longest path from $v$ to any PO in $G_{\mathcal{F}}$.

**Lemma E.1** (Zero leeway indicates membership in longest path). *Let $G_{\mathcal{F}} = (V, E)$ be a BN and $v \in V$ be a vertex. Then $leeway(v) = 0 \iff v$ is on a longest path in $G_{\mathcal{F}}$.*

*Proof.* Recall that $depth(v)$ is the length of the longest path from any PI to $v$ and $depth(G_{\mathcal{F}})$ is the maximum vertex depth. Let $n = depth(v) + lpl(v)$ and $P_n = (v_0, \ldots, v, \ldots, v_n) \in V^{n+1}$ be a longest length path in $G_{\mathcal{F}}$ involving $v$. If $leeway(v) = 0$, then $depth(G_{\mathcal{F}}) = depth(v) + lpl(v) = n$, which implies $v$ is on a longest path in $G_{\mathcal{F}}$. If $v$ is on a longest path in $G_{\mathcal{F}}$, then $n = depth(v) + lpl(v) = depth(G_{\mathcal{F}})$, which implies $leeway(v) = 0$. $\qquad\square$

---

**Algorithm 2:** Maintain depth NN optimization algorithm (`optMaintainDepth`)

**Input:** BN $G_{\mathcal{F}} = (V, E)$, where $\mathcal{F} = \cup_{k=1}^{K} \mathbb{Z}^{2^k}$, and function $\delta : V \to \mathbb{N}_0$ representing the difference in criterion between MP and MMP representations of $v \in V$
**Output:** Vertices $\mathcal{M} \subseteq V$ selected to be represented as MMPs
**Function:** `optMaintainDepth`$(G_{\mathcal{F}}, \delta)$

1    $G'_{\mathcal{F}} \leftarrow$ augment$(G_{\mathcal{F}}, \{v \in V \mid v.\text{outdegree} = 0\})$
2    $\mathcal{M} \leftarrow \{v \in V \mid v.\text{outdegree} = 0 \land \delta(v) > 0\}$         `// Consider POs`
3    computeLeeways$(G'_{\mathcal{F}})$
4    **for** $v \in V$, *in descending order w.r.t. $\delta$,* **do**       `// Greedy candidate selection`
5       **if** $\big(v.\text{indegree} > 0\big) \land \big(v.\text{outdegree} > 0\big) \land \big(v.\text{leeway} > 0\big) \land \big(\delta(v) > 0\big)$ **then**
6          $\mathcal{M} \leftarrow \mathcal{M} \cup \{v\}$
7          $G'_{\mathcal{F}} \leftarrow$ augment$(G'_{\mathcal{F}}, \{v\})$
8          computeLeeways$(G'_{\mathcal{F}})$
9       **end**
10   **end**
11   **return** $\mathcal{M}$

---

*Remark* E.2 (Correctness for Algorithm 2 (`optMaintainDepth`)). We prove correctness for Algorithm 2 by showing that the subset of vertices $\mathcal{M} \subseteq V$ it selects to be represented as MMPs do not cause the depth of the resulting *layer-merged* NN to increase.

Let $G_{\mathcal{F}} = (V, E)$ be a BN. We denote the *unmerged* NN and *layer-merged* NN (see § 2.2) of $G_{\mathcal{F}}$ resulting from synthesis as $N$ and $N'$, respectively.

Let
$$P_d^{G_{\mathcal{F}}} = (v_0, v_1, v_2, \ldots, v_i, v_{i+1}, \ldots, v_{d-1}, v_d) \tag{219}$$
be a longest path in $G_{\mathcal{F}}$, where $d = depth(G_{\mathcal{F}})$, $v_0$ is a PI, and $v_d$ is a PO. Then
$$P_{2d}^N = (v_0, h_1, v_1, h_2, v_2, \ldots, h_i, v_i, h_{i+1}, v_{i+1}, \ldots, h_{d-1}, v_{d-1}, h_d, v_d) \tag{220}$$
is a longest path in $N$, where $h_i$ represent some *hidden* vertex of $v_i$, and
$$P_{d+1}^{N'} = (v_0, h_1, h_2, \ldots, h_i, h_{i+1}, \ldots, h_{d-1}, h_d, v_d) \tag{221}$$
is a longest path in $N'$. Equation 220 holds since there are no vertices in $G_{\mathcal{F}}$ representing constant BFs (constant 1 or constant 0), as these vertices are combined with their successors by replacing successor BFs with their positive or negative cofactor. Thus, each $v_i \in P_d^{G_{\mathcal{F}}}$, $i \in \{1, \ldots, d\}$, will have at least one hidden vertex $h_i$ as a predecessor in $N$.

We make two remarks about selecting vertices in $G_{\mathcal{F}}$ to be represented as MMPs. Firstly, POs can always be represented as MMPs without increasing the depth of $N'$. Taking $v_d$ as an example, we see that adding a non-linearity at $v_d$ does not prevent the merging of any vertices since $v_d \in P_{2d}^N$ does not have a $h_{d+1}$ to merge with in $N$.

Secondly, note that if some intermediate vertex $v_i \in P_d^{G_{\mathcal{F}}}$, $i \in \{1, \ldots, d-1\}$, is selected to be represented as an MMP in $G_{\mathcal{F}}$, then since a non-linearity must be applied at $v_i$ in $N$, $v_i$ cannot be merged with $h_{i+1}$. This would cause the length of $P_{d+1}^{N'}$, and in turn the depth of $N'$, to increase by one. Consequently, any intermediate vertex $v \in V$, that is, a vertex that is neither a PI nor PO, can only be selected to be an MMP if $v$ does not appear on a longest path in $G_{\mathcal{F}}$. After selecting such a $v$, the lengths of all paths in $N'$ involving $v$ will increase by 1, which may prevent the selection of other vertices that could have been represented as MMPs.

Since Algorithm 2 only selects intermediate vertices with positive leeway and POs to be MMPs, following Lemma E.1, the depth of $N'$ will not increase.

---

**Algorithm 3:** Computing leeways

**Input:** BN $G_{\mathcal{F}} = (V, E)$
**Function:** `computeLeeways(`$G_{\mathcal{F}}$`)`

```
1  for d = G_F.depth, G_F.depth −1, . . . , 0 do
2      for v ∈ G_F.verticesAtDepth(d) do
3          if v.outdegree = 0 then
4              v.lpl ← 0                          // Longest path length (lpl)
5          else
6              v.lpl ← max({u.lpl ∈ ℕ_0 | (v, u) ∈ E}) + 1
7          end
8          v.leeway ← G_F.depth − v.depth − v.lpl
9      end
10 end
```

---

*Remark* E.3. Algorithm 3 runs in time $O(|V| + |E|)$.

*Remark* E.4. Algorithm 4 runs in time $O(|E|)$.

*Remark* E.5 (Runtime analysis for Algorithm 2). Algorithm 2 runs in time $O(V(V+E))$. The for-loop (lines 4:10 of Algorithm 2) run $O(V)$ times. Each iteration runs in time $O(1) + O(E) + O(V + E) = O(V + E)$. There may be $O(V)$ additional vertices and edges added via augmentations. However, this does not affect the asymptotic runtime, as $O((V + V) + (E + V)) = O(V + E)$.

*Remark* E.6 (Space complexity for Algorithm 2). Algorithm 2 runs in space $O(V + E)$. Storing the input graph requires space $O(V + E)$, and $O(V)$ additional space is used to store (i) vertices and edges added via augmentations, (ii) vertex properties such as leeway, and (iii) the selected candidate set $\mathcal{M} \subseteq V$.

---

**Algorithm 4:** Augmenting a BN with a temporary vertex

**Input:** BN $G_{\mathcal{F}} = (V, E)$ and vertices $A \subseteq V$
**Output:** Augmented BN $G'_{\mathcal{F}} = (V', E')$
**Function:** augment($G_{\mathcal{F}}$, $A$)

1 **for** $v \in A$ **do**
2      $V' \leftarrow V \cup \{v_{temp}\}$
3      $E' \leftarrow (E \setminus \{(u, v) \in E \mid u \in V\}) \cup \{(u, v_{temp}) \in (V')^2 \mid (u, v) \in E\} \cup \{(v_{temp}, v)\}$
4      $v_{temp}.\text{depth} \leftarrow v.\text{depth}$
5      $v.\text{depth} \leftarrow v.\text{depth} + 1$
6 **end**
7 **return** $G'_{\mathcal{F}} = (V', E')$

---

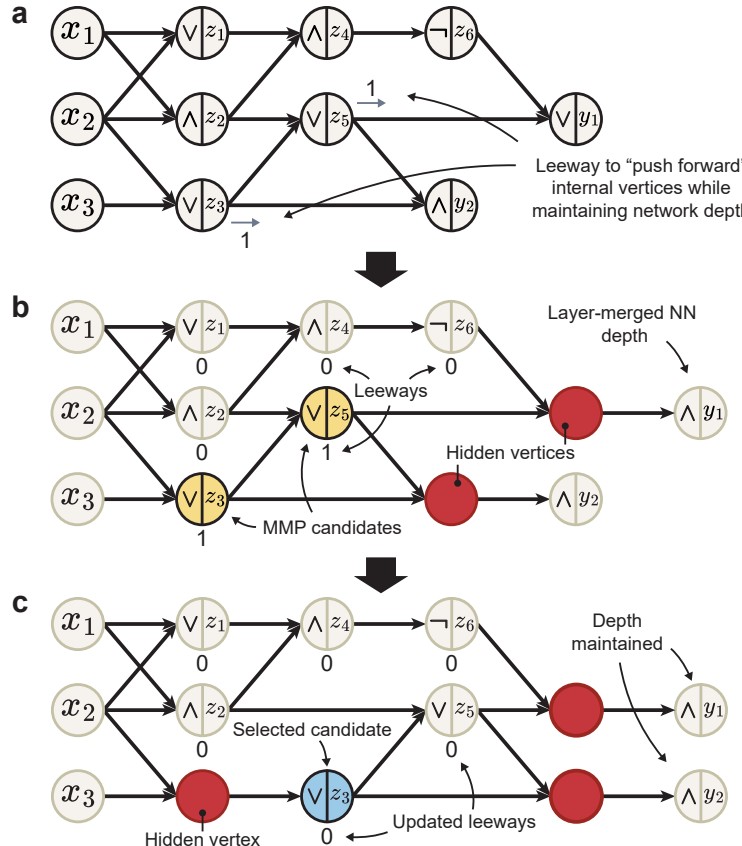

Figure 7: Architecture-aware neural network (NN) lossless compression algorithm that selects a subset of vertices $\mathcal{M}$ to be represented as minimal multilinear polynomials (MMPs) while maintaining the *layer-merged* NN depth. **a**, Boolean network (BN) with three inputs $(x_i)$, two outputs $(y_i)$, and internal vertices $(z_i)$, where $z_3, z_5$ have leeway (Definition E.3) greater than zero. **b**, Output vertices are always augmented with hidden vertices (red). An MMP candidate (yellow) is an internal vertex with positive leeway that is associated with a function whose multilinear polynomial (MP) can be minimized. Here we assume minimization is with respect to the number of terms in the MP, and hence $a \vee b = a + b - ab$ can be minimized to $a + b$, whereas $a \wedge b = ab$ and $\neg a = 1 - a$ cannot. **c**, Supposing $z_3$ is next selected to be represented as an MMP (blue), the newly introduced hidden vertex (red) causes the leeway of $z_5$ to drop to 0. Since no MMP candidates remain, the algorithm terminates with MMP selection $\mathcal{M} = \{z_3\}$. The NN synthesized from the BN in **c** is functionally equivalent to that of **b**, has the same depth, yet has 4 fewer neurons and 10 fewer connections (see Appendix H).

## F  DIGITAL CIRCUITS AND AUTOMATA USED IN EXPERIMENTS

The technology mapping sequence discussed in § 2.2, and the optimization techniques we propose, work for arbitrary Boolean networks (BNs). Hence, given a specification of a discrete graphical model, if we can encode the model as a BN, then it can be provided as input to the optimized technology mapping sequence to arrive at an optimized NN representation of the BN. In this section, we describe encoding techniques for, and instances of, the two discrete graphical models that were used in our experimental results; digital circuits and deterministic finite automata.

**Digital circuits.**  Digital circuits specified in hardware description languages can be converted into BNs via a process known as technology independent synthesis (see Gavier et al. (2023)). We provide additional information and references for the four open-source digital circuits used in the experimental results of the main text.

- `uart`[4] is a communications core that parses internal bus commands via a Universal Asynchronous Receiver-Transmitter (UART) interface.
- `sha3`[5] is a cryptographic core implementing Secure Hash Algorithm 3 (SHA-3) with an output length of 512 bits.
- `ecg`[6] is a cryptographic core implementing scalar multiplication of an element in the elliptic curve group.
- `aes`[7] is a cryptographic core implementing the encryption function of the Advanced Encryption Standard (AES) symmetric-key algorithm with a key length of 128 bits.

We also provide details on the digital circuits used in the extended results (see Appendix G.2).

- `add` is a 64-bit full adder.
- `alu` is a 32-bit arithmetic logic unit implementing $+, -, \times, \div$, etc.
- `rand8` and `rand12` are randomly generated $n$-input, $n$-output look-up table (LUT) digital circuits, where $n \in \{8, 12\}$.
- `riscv`[8] is a RISC-V 3-stage pipeline.

**Deterministic finite automata.**  A deterministic finite automaton (DFA) is a computational model with finite memory. Formally, it is a 5-tuple $D = (Q, \Sigma, \delta, q_0, F)$, where

1. $Q$ is a finite set called the states,
2. $\Sigma$ is a finite set called the alphabet,
3. $\delta : Q \times \Sigma \to Q$ is the transition function,
4. $q_0 \in Q$ is the start state, and
5. $F \subseteq Q$ is the set of accept states (Sipser, 1996).

We now present a BN construction for an arbitrary DFA $D = (Q, \Sigma, \delta, q_0, F)$ using a one-hot encoding for states and symbols. Let $V_{PI} = \{q_1, q_2, \ldots, q_{|Q|}\} \cup \{\sigma_1, \sigma_2, \ldots, \sigma_{|\Sigma|}\}$ be the set of primary input vertices, $V_{PO} = \{q'_1, q'_2, \ldots, q'_{|Q|}\}$ be the set of primary output vertices, and $V = V_{PI} \cup V_{PO}$ be the set of vertices. Let $\Delta_{q'} = \{(q, \sigma) \in Q \times \Sigma \mid \delta(q, \sigma) = q'\}$ be the preimage of $q' \in V_{PO}$. Let $E = \{(u, q') \in V_{PI} \times V_{PO} \mid \exists (q, \sigma) \in \Delta_{q'} (u = q \vee u = \sigma)\}$ be the set of edges, and $\psi : V \to \mathcal{F}$ be the labeling function with specification

$$\psi(q') = \bigvee_{(q, \sigma) \in \Delta_{q'}} (q \wedge \sigma), \tag{222}$$

for all $q' \in V_{PO}$ (recall the labeling function assigns a single output BF to each non-PI vertex). Then, $G_{\mathcal{F}} = (V, E)$ with labeling function $\psi$ is a BN representation of DFA $D$. We note that the resulting BN can be optimized using technology independent synthesis tools.

---

[4] https://opencores.org/projects/uart2bus
[5] https://opencores.org/projects/sha3
[6] https://opencores.org/projects/ecg
[7] https://opencores.org/projects/tiny_aes
[8] https://github.com/ucb-bar/riscv-mini

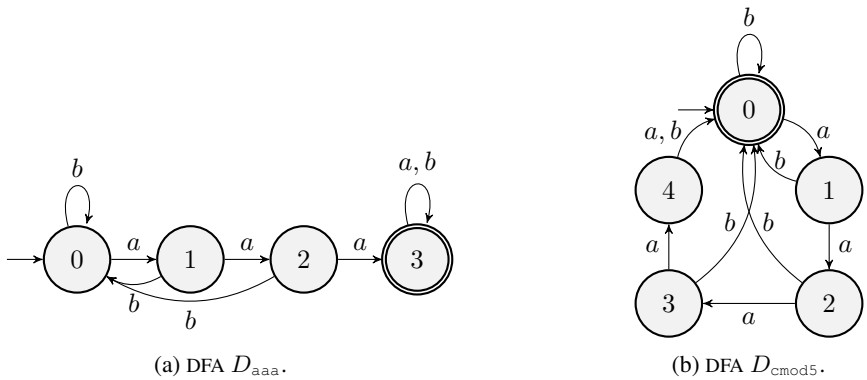

(a) DFA $D_{\mathtt{aaa}}$.    (b) DFA $D_{\mathtt{cmod5}}$.

Figure 8: State diagrams of the DFAs used in the extended results.

We now provide details on the DFAs used in the extended results (see Appendix G.2).

- `aaa` is the BN representation of a DFA $D_{\mathtt{aaa}}$ whose language is the set of all strings over the alphabet $\Sigma = \{a, b\}$ containing the substring "$aaa$". The regular expression for this language is $(a + b)^* aaa (a + b)^*$. Formally, let $D_{\mathtt{aaa}} = (Q, \Sigma, \delta, 0, F)$ be a DFA, where
    1. $Q = \{0, 1, 2, 3\}$ is the set of states,
    2. $\Sigma = \{a, b\}$ is the alphabet,
    3. $0 \in Q$ is the start state,
    4. $F = \{3\}$ is the set of accept states, and
    5. transition function $\delta : Q \times \Sigma \to Q$ is specified in Figure 8a.
- `cmod5` (counter $\bmod\, 5$) is the BN representation of a DFA $D_{\mathtt{cmod5}}$ whose language is the set of all strings over the alphabet $\Sigma = \{a, b\}$ containing either (i) $m$ $a$'s (increments), or (ii) a suffix consisting of a $b$ (reset) followed by $m$ $a$'s (increments), where $m \equiv 0 \pmod 5$. The regular expression for this language is $(a^5)^* + (a + b)^* b (a^5)^*$. Formally, let $D_{\mathtt{cmod5}} = (Q, \Sigma, \delta, 0, F)$ be a DFA, where
    1. $Q = \{0, 1, 2, 3, 4\}$ is the set of states,
    2. $\Sigma = \{a, b\}$ is the alphabet,
    3. $0 \in Q$ is the start state,
    4. $F = \{0\}$ is the set of accept states, and
    5. transition function $\delta : Q \times \Sigma \to Q$ is specified in Figure 8b.

# G EXTENDED RESULTS

## G.1 ACCELERATING MP-BASED TECHNOLOGY MAPPING WITH NPN CLASSIFICATION

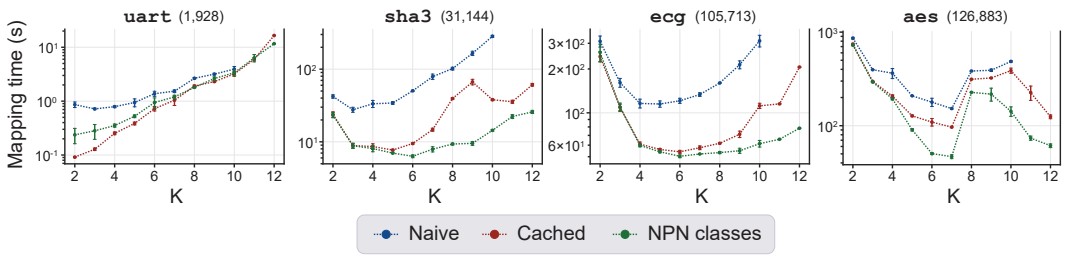

Figure 9: Accelerating MP- (multilinear polynomial) based technology mapping (and in turn NN-based technology mapping) with NPN classification. MP-based technology mapping takes a Boolean network (BN) with vertices associated with $k$-input truth tables, $1 \le k \le K$, and computes the functionally equivalent MP for each vertex. *Naive* (blue) computes the MPs per vertex in the BN, while *Cached* (red) computes them per unique function. *NPN classes* (green) implements the proposed NPN classification algorithm, computing MPs via NPN canonical forms and transformations. *NPN classes* achieves consistent speedups when the overhead of computing NPN canonical forms and transformations is insignificant to that of computing MPs. Lines and error bars denote the sample mean and standard deviation over 3 trials. Logic gate counts, following technology independent synthesis, are shown in parentheses.

## G.2 RESULTS ON ADDITIONAL BOOLEAN NETWORKS

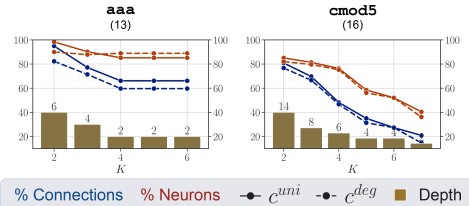
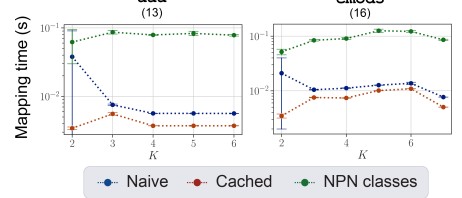

(a) Reduction in connections (blue) and neurons (red) relative to the *unmerged* NN by optimizing all sub-NNs with the proposed `optAll` algorithm.

(b) Mapping time for MP-based technology mapping.

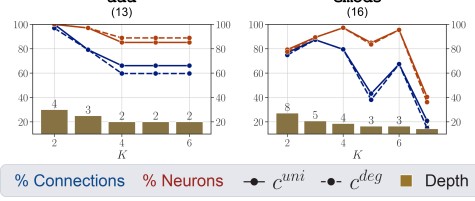
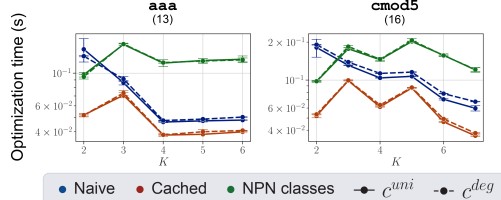

(c) Reduction in connections and neurons relative to the *layer-merged* NN by optimizing a subset of sub-NNs selected by the proposed `optMaintainDepth` algorithm. The size of the *layer-merged* NN is able to be reduced without increasing its depth.

(d) Mapping time for MP- and MMP-based technology mapping w.r.t. the uniform ($c^{uni}$) and degree ($c^{deg}$) criterions.

Figure 10: NN lossless optimization and mapping acceleration results for DFA BNs `aaa` and `cmod5`. NN lossless optimization with respect to the uniform ($c^{uni}$) and degree ($c^{deg}$) criterions for MMPs for *unmerged* (a) and *layer-merged* (c) NNs. NN depths are shown in gold. Mapping time for MP-based technology mapping (b) and MP- and MMP-based technology mapping (d). See § 4.2 for details on the different methods (*Naive, Cached, NPN classes*). Lines and error bars denote the sample mean and standard deviation over 3 trials. Logic gate counts, following the DFA BN construction and technology independent synthesis, are shown in parentheses.

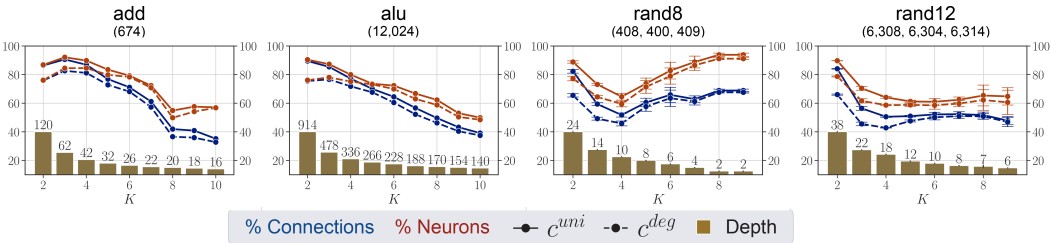

(a) Reduction in connections (blue) and neurons (red) relative to the *unmerged* NN by optimizing all sub-NNs with the proposed `optAll` algorithm.

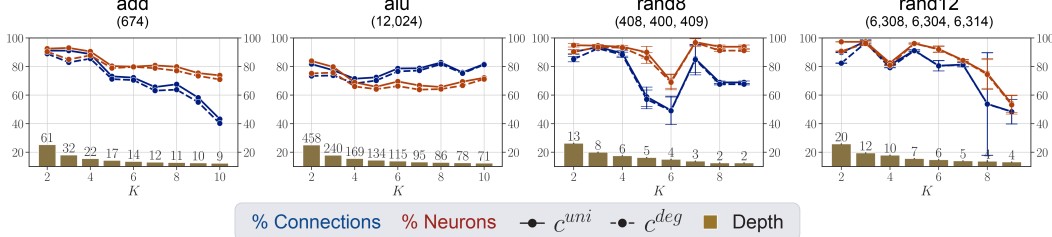

(b) Reduction in connections and neurons relative to the *layer-merged* NN by optimizing a subset of sub-NNs selected by the proposed `optMaintainDepth` algorithm. The size of the *layer-merged* NN is able to be reduced without increasing its depth.

Figure 11: NN lossless optimization with respect to the uniform ($c^{uni}$) and degree ($c^{deg}$) criterions for MMPs. NN depths are shown in gold. Logic gate counts, following technology independent synthesis, are shown in parentheses. For `rand8` and `rand12`, lines and error bars denote the sample mean and standard deviation across 3 different circuits.

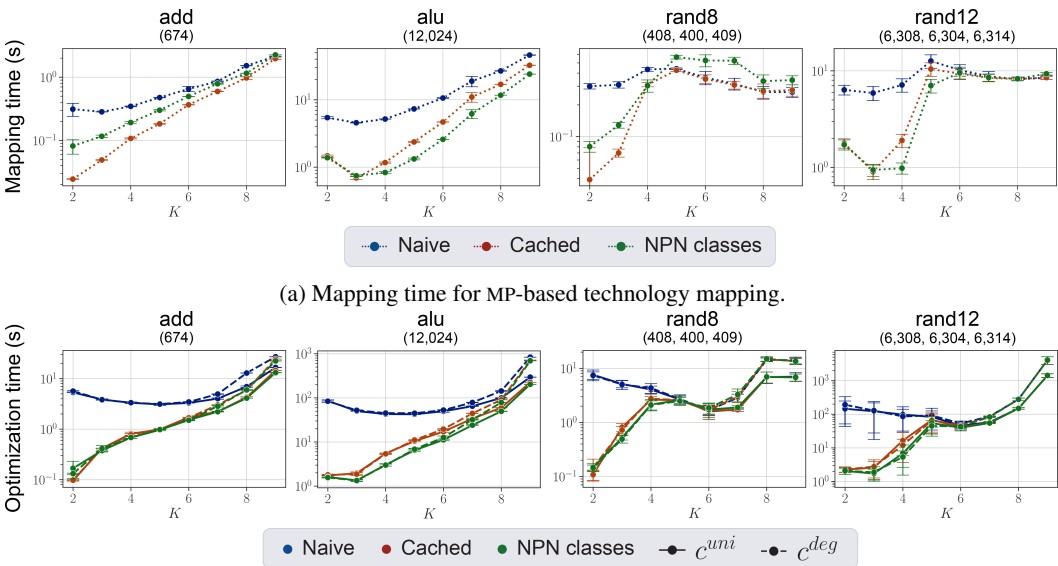

(a) Mapping time for MP-based technology mapping.

(b) Mapping time for MP- and MMP-based technology mapping w.r.t. the uniform ($c^{uni}$) and degree ($c^{deg}$) criterions.

Figure 12: Accelerating MP- and MMP-based technology mapping with NPN classification. *Naive* (blue) computes the MP (a) or MP and MMP (b) per vertex in the BN, while *Cached* (red) computes them per unique function. *NPN classes* (green) implements the proposed NPN classification algorithm. For `add` and `alu`, lines and error bars denote the sample mean and standard deviation over 3 trials. For `rand8` and `rand12`, they denote sample mean and standard deviation across 3 different BNs with 3 trials each. Logic gate counts, following technology independent synthesis, are shown in parentheses.

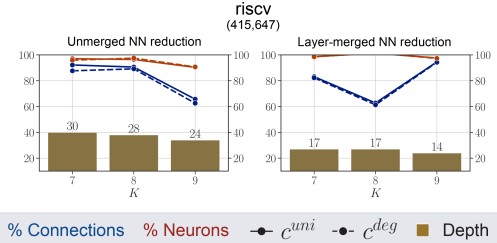 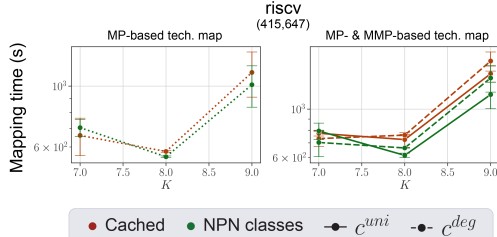

Figure 13: NN lossless optimization and mapping acceleration results for `riscv`. NN lossless optimization w.r.t. the uniform ($c^{uni}$) and degree ($c^{deg}$) criterions for MMPs for *unmerged* (far left) and *layer-merged* (mid left) NNs. NN depths are shown in gold. Mapping time for MP-based technology mapping (mid right) and MP- and MMP-based technology mapping w.r.t. the uniform ($c^{uni}$) and degree ($c^{deg}$) criterions (far right). Lines and error bars denote the sample mean and standard deviation over 3 trials. Logic gate count, following technology independent synthesis, is shown in parentheses.

We present additional results on three different classes of digital circuit BNs (see Figure 11, Figure 12, and Figure 13), as well as results on DFA BNs (see Figure 10), using the same experimental setting as in the main text (see § 4). We provide details on the digital circuits and automata in Appendix F, and summarize our findings below.

**Well-known/Arithmetic circuits (`add` & `alu`)**   In comparison to the main results, we achieve a new best under the depth-constrained NN compression setting with `add`, reducing the connections and neurons in the *layer-merged* NN by 60% and 30%, respectively. When compressing the *unmerged* NN of `add` and `alu`, we observe that as $K$ increases so does the relative reduction in size (also observed for `uart` and `ecg` in the main results). As for mapping time, `add` is too small to see benefits with *NPN classes*. While `alu` is also relatively small, we see a speedup in mapping time with *NPN classes*.

**Arbitrary/Random circuits (`rand8` & `rand12`)**   We achieve considerable compression in NN size for `rand8` and `rand12`. We also find that the *NPN classes* and *Cached* algorithms are not useful for these BNs for large $K$. This is because as $K$ increases, if BFs are equally likely to occur in a BN, the likelihood of having two BFs in the same NPN class, or in the same BN, tends to zero (see Appendix G.3). We note that the applications in the *Broader Impacts* statement (see Appendix I) mainly involve non-random BNs, for which our experiments have demonstrated the value of *NPN classes*.

**Large circuit / Core (`riscv`)**   `riscv` is 3.3 times larger than `aes` (the largest BN we reported in the main results). For `riscv`, our techniques were able to reduce connections and neurons by up to 40% and 10%, respectively, without a depth constraint, and by up to 40% and -1%, respectively, in the depth-constrained setting (the increase in neurons is justified by the $\ell_1$-norm relaxation, see Appendix C.1). We also observed a mean speedup using *NPN classes* over the *Cached* algorithm.

**Deterministic finite automata (`aaa` & `cmod5`)**   Following the BN construction for DFAs discussed in Appendix F, the BNs for $D_{\text{aaa}}$ and $D_{\text{cmod5}}$ have 6 and 7 primary input vertices, respectively. Consequently, in Figure 10, we limit $K$-LUT mapping to $K \leq 6$ and $K \leq 7$ for the BNs `aaa` and `cmod5`, respectively. Since the maximum number of inputs to any BF in `aaa` and `cmod5` is 4 and 7, respectively, $K$-LUT mapping applied to `aaa` with $K \in \{4, 5, 6\}$ results in the same $K$-LUT BN. This is why we observe the same reduction results for `aaa` with $K \in \{4, 5, 6\}$ in Figure 10a and Figure 10c. Moreover, we see that the resulting NN representations for `aaa` and `cmod5` with $K \in \{4, 5, 6\}$ and $K = 7$, respectively, have only two layers. In these cases, the *unmerged* and *layer-merged* NNs are identical since inter-depth composition cannot be applied (see § 2.2). Even though these BNs are significantly smaller than the digital circuit BNs we examine, we are still able to achieve substantial compression; observing a reduction in connections and neurons of 40% and 12%, respectively, for `aaa` with $K \in \{4, 5, 6\}$ under the $c^{deg}$ criterion (reducing connections from 62 to 37, and neurons from 27 to 24), and a reduction in connections and neurons of 85% and 64%, respectively, for `cmod5` with $K = 7$ under the $c^{deg}$ criterion (reducing connections from 407 to 60, and neurons from 94 to 34). As for mapping time, all $K$-LUT BNs of `aaa` and `cmod5` have $\leq 19$ BFs. Hence, these BNs are too small to see benefits from the proposed NPN classification algorithm.

## G.3 EMPIRICAL ANALYSIS OF NPN CLASSIFICATION

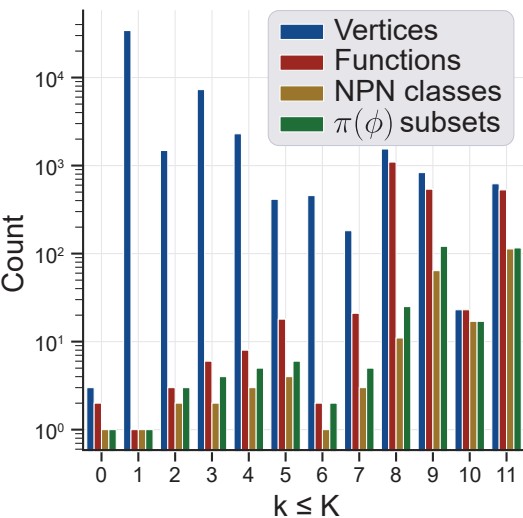

Figure 14: Empirical analysis of the quantities introduced in Remark D.8 for the $K$-LUT Boolean network (BN) of aes with $K = 11$. The number of vertices (blue), unique functions (red), NPN classes (gold) and permuted phase assignment subsets (green) in the BN are shown for $0 \leq k \leq K$.

We empirically examine Remark D.12 for the largest circuit (aes) and largest $K$ ($K = 11$) presented in the main text for MP- and MMP-based technology mapping. Since there are $2^{2^k}$ BFs and the lower bound on the number of NPN classes is $\frac{2^{2^k}}{2^{k+1}k!}$ (Sasao & Butler, 2022), as $k$ tends to infinity the number of NPN classes will approach the number of unique functions. Consequently, the *Cached* algorithm from § 4.2 and the *NPN classes* algorithm (i.e., Algorithm 1) would require the same number of calls to ttToMP and mpToMMP. However, $K$ is generally limited to be around 10 due to the exponential runtime and space requirements of synthesis operations and data structures, respectively. Hence, as mentioned in Remark D.12, we can make use of NPN equivalence classes to speed up mapping time; a statement which was validated by our empirical results. In Figure 14, we observe both ends of the bounds presented in Remark D.12 for the number of NPN classes and the number of permuted phase assignment subsets. While for $k = 5$ there are 18 unique functions, 4 NPN classes and 6 permuted phase assignment subsets, for $k = 8$, there are 1,096 unique functions yet only 11 NPN classes and 25 permuted phase assignment subsets. Thus, the *Cached* algorithm requires 1,096 calls to ttToMP and mpToMMP, whereas the *NPN classes* algorithm requires 11 calls to ttToMP and 25 calls to mpToMMP. This explains the observed speedup of $5.9\times$ in Figure 6.

# H SYNTHESIZED NEURAL NETWORKS FOR FIGURE 7

See Figure 15 and Figure 16.

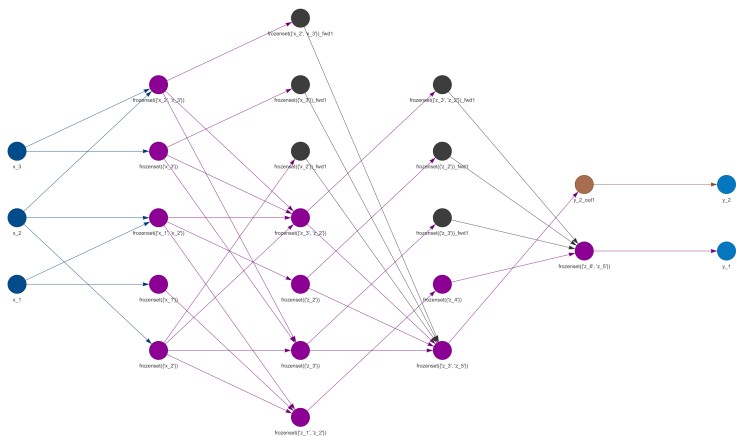

Figure 15: *Layer-merged* neural network (NN) for panel **b** of Figure 7. The NN has 24 neurons, 38 connections, and a depth of 5.

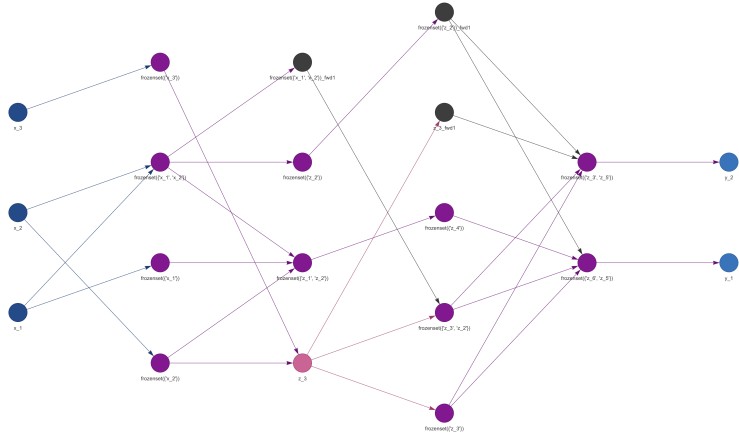

Figure 16: *Layer-merged* neural network (NN) for panel **c** of Figure 7. The optMaintainDepth NN compression algorithm was applied under the uniform criterion. The NN has 20 neurons, 28 connections, and a depth of 5.

## I   BROADER IMPACTS

In this work, we propose a novel lossless optimization technique for neural network (NN) representations of Boolean networks (BNs) and an algorithm for accelerating the conversion from BN to NN. Here, we briefly discuss the positive impacts our results suggest for two important domains: BN (in particular, digital circuit) simulation and neurosymbolic artificial intelligence (AI).

Register-transfer level (RTL) simulation is used to achieve functional verification of digital circuits (Tan & Rosdi, 2014). Although the use of GPUs for RTL simulation was proposed as early as 2011 (Qian & Deng, 2011), due to the continued developments in GPU hardware, modern machine learning frameworks, and the proliferation of GPUs in data centers, there has been a recent call for accelerating RTL simulation on parallel processing hardware platforms (Zhang et al., 2020). This has resulted in GPU RTL simulation tools (Lin et al., 2022) and their integration into more advanced functional verification techniques, such as hardware fuzzing (Lin et al., 2023). NN-based technology mapping was recently applied in a GPU RTL simulation methodology, leveraging batched stimulus to parallelize the evaluation of test cases (Gavier et al., 2023). Our proposed NN optimization technique reduces the memory footprint of the resulting NN, meaning more test cases can be batched in parallel and throughput can be further increased. Furthermore, our NPN classification technique for accelerating mapping time is crucial for the adoption of NN-based simulation methodologies in practice, as time delays are expensive in the interative design flow of digital circuits. Our proposed techniques are also applicable to other Boolean network simulation frameworks, such as gene regulatory networks (Bornholdt, 2008; Biswas et al., 2021).

Neurosymbolic AI is a promising paradigm for integrating learning-based techniques with symbolic algorithms that provide robustness and explainability. However, neurosymbolic methods generally involve heterogeneous computing architectures, which diverge from the current hardware roadmap of improving matrix-vector multiplication (Wan et al., 2024). One way of utilizing recent advances in matrix-vector multiplication architectures, such as compute-in-memory crossbars (Rao et al., 2023; Ambrogio et al., 2023), is to use NN-based technology mapping to convert the symbolic circuits into matrix-vector multiplications. Hence, all components of the neurosymbolic system can be run on the same hardware architecture. Moreover, the high NN sparsity that results from $K$-LUT-based technology mapping (Gavier et al., 2023) implies that the mapped NN is likely to satisfy the thermal constraints that accompany three-dimensional crossbars (Boahen, 2022). Our proposed NN optimization techniques reduce matrix sizes and energy requirements for neurosymbolic approaches that would use such a matrix-vector multiplication architecture, which is of particular relevance for memory- and energy-limited edge computing applications.

For example, autonomous systems often integrate data from multiple sensors to make informed decisions about their actions. These systems typically employ NN encoders to transform high-dimensional, continuous data into a discrete set of symbolic classifications, which encapsulate critical information for the system. These classifications are then passed to a logical decision-making algorithm that determines the system's actions. In a modern computing architecture, the NN encoders could leverage compute-in-memory crossbars for efficient processing. However, the decision-making algorithms are generally executed on digital processors, resulting in a heterogeneous system design. Using NN-based technology mapping, it is possible to translate logical circuits into a series of matrix-vector multiplications. This approach enables a homogeneous system architecture consisting exclusively of crossbars and their associated peripherals. Our lossless NN optimization techniques, which reduce matrix sizes and the number of required operations in the resulting NN, would reduce the area requirements on the crossbar and lower energy consumption for such an architecture while ensuring the decision-making algorithm is computed without error – an aspect that is vital for system safety and reliability.

