# OpenReview forum: "Optimizing Neural Network Representations of Boolean Networks"
_ICLR.cc/2025/Conference — ICLR 2025 Poster_

### Official Review · Reviewer_vaK3 · 2024-11-03

**Soundness:** 3
**Presentation:** 3
**Contribution:** 3
**Rating:** 8
**Confidence:** 3

**Summary:**

This paper addresses the challenge of optimizing neural network (NN) representations of Boolean networks (BNs). The authors point out that while NNs can represent Boolean functions, the current state-of-the-art method for deriving these representations often results in suboptimal NNs with a large number of neurons and connections, leading to inefficient simulation. Existing NN compression techniques are either lossy or inapplicable due to the specific nature of the NN-based technology mapping problem.

The paper makes three key contributions. First, it proposes a novel lossless technique for optimizing two-layer NN representations of Boolean functions, focusing on reducing the number of neurons and connections while preserving functional equivalence. This is achieved by formulating the problem as a constrained minimization task and employing a convex relaxation technique to solve it.  Second, the authors introduce an objective-aware optimization algorithm that leverages Negation-Permutation-Negation (NPN) classification. This algorithm exploits shared representations among the two-layer sub-networks to significantly speed up the optimization process, demonstrating a substantial speedup over naive and caching-based solutions. Third, an architecture-aware lossless optimization algorithm is proposed, targeting both unmerged and layer-merged NN architectures. This algorithm determines which sub-NNs should be minimized to achieve overall NN size reduction while optionally maintaining the depth of the layer-merged network, which is critical for latency-sensitive applications. Experimental results on benchmark BNs derived from digital circuits show significant reductions in the size of the resulting NNs, confirming the effectiveness of the proposed optimization techniques.

**Strengths:**

**Originality:** The paper tackles the problem of optimizing NN representations of Boolean networks from a fresh perspective. While NN compression is a well-studied area, the authors identify a specific gap in existing techniques, namely the lack of *lossless* methods applicable to the BN-to-NN mapping problem. They creatively combine ideas from Boolean function classification (NPN) and convex optimization to develop a new approach to address this gap. The concept of leveraging NPN classes to accelerate the optimization by exploiting shared representations among subproblems is particularly original and demonstrates a deep understanding of the underlying structure of the problem. Moreover, the introduction of the *leeway* concept and the architecture-aware algorithm for maintaining depth in layer-merged NNs showcases innovative thinking in adapting the optimization process to different NN architectures.

**Quality:** The paper is technically sound, with rigorous mathematical formulations and proofs supporting the proposed methods. The authors carefully define the necessary concepts and notation, ensuring clarity in their technical exposition. The experimental methodology is well-designed, with appropriate benchmarks and metrics used for evaluation. The comparison against relevant baselines (naive and caching solutions) provides a strong validation of the proposed techniques. The inclusion of additional results and analysis in the appendix further reinforces the quality of the work.

**Clarity:** The paper is generally well-written and organized. The introduction effectively motivates the problem and summarizes the key contributions. The background section provides the necessary context and definitions, though perhaps could benefit from a slightly higher-level motivating example early on for a broader audience.  The use of figures, especially Figure 1, significantly aids in understanding the optimization problem and the proposed approach. The steps of the algorithms are clearly presented, and the results are reported in a concise and informative manner.

**Significance:** The paper addresses an important problem with practical implications for various domains. Efficient BN simulation is crucial in areas like circuit verification and design automation. The proposed optimization techniques can lead to substantial reductions in NN size and faster optimization times, making NN-based BN simulation more practical and scalable. Moreover, the connection to neurosymbolic AI highlights the potential of the work for advancing this emerging field. The ability to represent symbolic systems efficiently using NNs could pave the way for new hardware architectures and algorithms that combine the strengths of both symbolic and connectionist AI approaches.  The paper's focus on lossless compression is also significant from a safety and reliability perspective, as it ensures that the optimized NN representation remains functionally equivalent to the original BN.

**Weaknesses:**

1. **Impact of L1 Relaxation:** The authors acknowledge that relaxing the ℓ0-norm objective to ℓ1 might lead to suboptimal solutions. However, the paper lacks a detailed analysis of the extent to which this relaxation affects the quality of the results.  Quantifying the gap between the ℓ0 and ℓ1 solutions for the benchmark BNs, or investigating alternative approximation methods for the ℓ0-norm minimization, would provide a more complete understanding of the trade-offs involved.  Perhaps experiments comparing the optimized NN size obtained with ℓ1 relaxation to theoretical lower bounds achievable with ℓ0 could highlight the potential room for improvement.

2. **Scalability to Larger BNs:** The experimental results suggest that the optimization time can become substantial for large BNs and higher values of K (maximum input size of LUTs).  While the NPN classification algorithm offers speedups compared to caching, the paper does not thoroughly investigate the scalability limitations of the overall method.  Analyzing the runtime complexity as a function of BN size and K, and potentially exploring strategies for further improving the efficiency of the optimization process (e.g., by leveraging parallelism or more sophisticated data structures), would be beneficial.  Consider profiling the algorithms to pinpoint bottlenecks and focus optimization efforts.

3. **Clarity and Accessibility for a Broader Audience:** Although the technical content is generally well-explained, the paper could benefit from a more intuitive and accessible introduction to the problem and its significance. Providing a high-level illustrative example that highlights the practical implications of optimizing NN representations of BNs would engage a broader readership within the ICLR community. While the paper currently focuses on a specialized audience with expertise in Boolean functions, making it more approachable for readers with a general machine learning background would enhance its impact.

**Questions:**

1. **Generalization to other Boolean network domains:** The experimental results focus primarily on digital circuits. Could the authors elaborate on the applicability of their methods to other types of Boolean networks, such as gene regulatory networks or biological networks? Are there any specific adaptations or considerations needed for these domains?  Presenting results on even a small set of non-circuit BNs would greatly bolster the claim of general applicability.

2. **Scalability analysis and potential optimizations:** The optimization time appears to grow considerably with BN size and K. Could the authors provide a more detailed analysis of the computational complexity of their methods?  Are there any potential optimizations or algorithmic improvements that could be explored to enhance scalability, such as parallelization or more efficient data structures? A breakdown of execution time for different stages of the algorithm would help identify bottlenecks

3. **Clarifying the impact of NPN transformations on the number of MMP computations:**  The paper mentions that using NPN classification can reduce the number of MMP computations compared to function caching. However, the precise reduction factor ((2^k)!/2^(k+1)k!) is not immediately intuitive.  Could the authors provide a more detailed explanation of how this reduction is achieved and its significance in practice, perhaps with a concrete example for a small value of k?  It would be especially insightful to directly visualize how many MMP computations are saved for each benchmark circuit.

4.  **Connection to Neurosymbolic AI Implementations:**  The paper mentions the potential of the work for neurosymbolic AI, but the link is somewhat abstract.  Could the authors expand on how specifically the proposed methods could be integrated into neurosymbolic systems? For example, are there specific neurosymbolic architectures or frameworks where these optimized NN representations would be particularly beneficial?  Perhaps a concrete example application scenario, even if hypothetical, could illustrate the potential.

---

> ### Author Response · Authors · 2024-11-24
> **Rebuttal by Authors (1/2)**
>
> We sincerely appreciate your positive feedback and insightful comments.
>
> We have uploaded a new PDF submission with changes listed in the *Global Response to the Reviewers* comment. Please refer to the new PDF when reading our responses below.
>
> ---
>
> **Weakness 1.**
>
> > *1. **Impact of L1 Relaxation.***
>
> Thank you for your insightful comment. We strongly agree that a theoretical analysis on the consequences of the $\ell_1$-relaxation for the MMP optimization problem, and an investigation of alternative approximations for the $\ell_0$-norm minimization, would provide great insight into the nature of the problem and trade-offs involved. However, we believe that such a formal investigation lies beyond the scope of this work.
>
> As a preliminary investigation on the topic, we have added a new Subsection C.1 to Appendix C that addresses the following question regarding solution suboptimality: by optimizing the MP representation of a BF w.r.t. the weighted one-norm, do we ever increase the value of the weighted zero-norm? We find that this does not occur for 2-input and 3-input BFs. However, it does occur for a small fraction of 4-input BFs.
>
> As a more general note, obtaining $\ell_0$-norm minima for every possible BF using brute-force search is computationally intractable, as there are $2^{2^k}$ BFs with $k$ inputs, and finding the minimum of each BF requires time $2^{O(k2^k)}$ (Remark C.2). Hence, exhaustive analyses on $\ell_0$-norm minima for $k\geq 4$ are currently infeasible.
>
> ---
>
> **Weakness 2.**
>
> > *2. **Scalability to Larger BNs.***
>
> We address this comment in our response to **Question 2**.
>
> ---
>
> **Weakness 3.**
>
> > *3. **Clarity and Accessibility for a Broader Audience.***
>
> Thank you for your thoughtful feedback on improving the clarity and accessibility of our paper. We appreciate your recognition of the technical content's quality and agree that the introduction could be refined to better connect with a broader audience within the ICLR community.
>
> For the final version, we will explore incorporating a high-level example into the introduction that highlights the practical implications of optimizing NN representations of BNs — perhaps with reference to the more concrete example that we have added to the Broader Impacts section as per your suggestion in **Question 4**. We value your comment as a guiding principle for these revisions.

---

> > ### Author Response · Authors · 2024-11-24
> > **Rebuttal by Authors (2/2)**
> >
> > **Question 1.**
> >
> > > *1. **Generalization to other Boolean network domains.***
> >
> > Thank you for your question and suggestion. The technology mapping sequence reviewed in Section 2.2 of the text, and the optimization techniques we propose, are applicable to arbitrary Boolean networks. The only adaptation required for new domains is the specification of a BN construction for the object of interest.
> >
> > In light of your suggestion, we have updated Appendix F to include a BN construction for deterministic finite automata, as well as the specifications for two automata we present extended experimental results on. In Section G.2 of Appendix G, we present new results on compressing NN representations of BNs that encode these automata.
> >
> > ---
> >
> > **Question 2.**
> >
> > > *2. **Scalability analysis and potential optimizations.***
> >
> > Regarding scalability, we have significantly expanded Subsection E.4 of Appendix E, providing time and space complexity analysis for our NPN classification algorithm, as well as for the baseline algorithms *Naive* and *Cached*. In Remark E.12, we compare the three algorithms and also consider the scalability of our NPN classification algorithm to larger BNs.
> >
> > The matrix multiplication version of `ttToMP` (Lemma B.1), NPN transformations (Remark E.4), and integer linear program constraint evaluation for `mpToMMP` are all operations that are suitable for GPU acceleration. Moreover, as mentioned by the reviewer, the calls to these subroutines could be parallelized. We plan to pursue such implementation optimizations in future work.
> >
> > ---
> >
> > **Question 3.**
> >
> > > *3. **Clarifying the impact of NPN transformations on the number of MMP computations.***
> >
> > In light of your question, we have significantly expanded upon the time complexity analysis for our NPN classification algorithm in Remark E.8 of Appendix E. We hope this exposition makes the reduction factor intuitive.
> >
> > As an example to explain the lower bound on the number of NPN classes and permuted phase assignment subsets, suppose we have a 3-input BF $\boldsymbol{f}\in\mathbb{Z}^{2^3}$ such that applying the $2^{3+1}3! = 96$ possible NPN transformations to $\boldsymbol{f}$ results in $m=96$  unique BFs. Now suppose we have a 3-LUT BN consisting of all these 96 unique BFs. Then such a BN will have a single NPN equivalence class ($c = \frac{m}{2^{3+1}3!} = 1$), and there will be a single call to `ttToMP` for its NPN canonical form. Since there are $2^{3} = 8$ possible input negations, the NPN class of 96 BFs will split into $s=8$ permuted phase assignment subsets of size $\frac{2^{3+1}3!}{2^{3}} = 2 \cdot 3! = 12$. Hence, there are $s = \frac{m}{2 \cdot 3!} = 8$ permuted phase assignment NPN canonical forms, and we compute `mpToMMP` for each of them. Overall, for this particular BN, we are able to decrease the number of calls to `ttToMP` from 96 to 1, and the number of calls to `mpToMMP` from 96 to 8. We hope this example makes the lower bounds on $c$ and $s$ more concrete.
> >
> > In Appendix G.3, we provide an empirical analysis of these quantities for the `aes` BN at $K=11$, and discuss how many `ttToMP` and `mpToMMP` computations are being saved by using the proposed NPN classification algorithm.
> >
> > ---
> >
> > **Question 4.**
> >
> > > *4. **Connection to Neurosymbolic AI Implementations.***
> >
> > In light of your suggestion, we have extended the Broader Impacts section of the Appendix to include an example application scenario. In short, we describe a hypothetical computing architecture for autonomous systems that involves decision making using sensor fusion. Our methodology would generate a size- and energy-optimized neural network that is functionally equivalent to the decision-making algorithm and applicable for use in a homogeneous matrix-vector multiplication computing architecture.

---

> > > ### Author Response · Authors · 2024-11-30
> > >
> > > Dear Reviewer vaK3,
> > >
> > > We sincerely appreciate the effort you have invested in reviewing our paper, and have taken care to address each of your comments and suggestions in our rebuttal.
> > >
> > > As the rebuttal period is nearing its conclusion, we wanted to kindly follow up to ensure that our responses have reached you. If there are any further clarifications or points of discussion, we would be happy to address them.
> > >
> > > We deeply value your insights and thank you for your time and effort in reviewing our work.
> > >
> > > Best regards,
> > >
> > > Authors

---

> > > ### Comment · Reviewer_vaK3 · 2024-12-02
> > >
> > > Thanks the authors for the detailed answers to my questions and concerns. i think the paper is worthy of an explicit accept

---

> > > > ### Author Response · Authors · 2024-12-02
> > > >
> > > > Thank you for taking the time to review our rebuttal and provide a reply, we are encouraged by your words. In a time when responses during the rebuttal period are not always guaranteed, we are especially grateful for your engagement and thoughtful consideration.

---

### Official Review · Reviewer_eC2r · 2024-11-03

**Soundness:** 4
**Presentation:** 4
**Contribution:** 3
**Rating:** 8
**Confidence:** 2

**Summary:**

In this work, the authors present an optimization framework for deriving representations of neural networks of Boolean networks. The overall goal of the proposed framework is to overcome known limitations of current state-of-the-art methodologies that result in suboptimal neural networks, in terms of number of neurons and connections, which hinders their application in real-case scenarios. More specifically, the proposed method introduces a lossless technique for optimizing neurons and connections needed in a two-layer NN representation of a Boolean function. This is achieved by establishing an optimization problem that transforms the pruning of the network architecture into monomial reduction tasks for a given polynomial. The lossless functionality between the Minimized Multilinear Polynomial and the represented Boolean function is achieved by incorporating the heavyside threshold of the NN, with the relaxation of the optimization objective to the $l_1$-norm providing the required convexity. Due to the NP-hard nature of the proposed optimization, the authors introduce an objective-aware optimization, which is based on Negation-Permutation-Negation classification, that constructs subclasses of multilinear polynomial representations for the minimization of the Boolean functions, exploiting the shared representations among them and accelerating the NN optimization process using unique function caching. Finally, the paper provides two alternatives for optimizing the Neural Networks, one that involves all the vertices of the binary networks to the minimization of the multilinear polynomial and another that selects the subset of vertices in such a way that the depth of the resulting layer-merged neural network does not increase. The proposed method achieves significant improvements in contrast to the state-of-the-art approach in terms of optimization speed and the required connections and neurons.

**Strengths:**

The paper is very well organized and written, providing the necessary background as well as the required proofs to support the claims of the authors. The experimental results clearly reflect the contribution of this work.

The proposed method outperforms the current state-of-the-art in terms of decreasing the size of neural networks, which means the number of connections and neurons, while simultaneously preserving the equivalent functionality. The proposed lossless compression, along with the objective-aware optimization, resulting in a faster and more efficient solution than the state-of-the-art.

The paper establishes a novel framework for lossless optimization techniques for neural network representation of boolean networks that can provide further advantages to neurosymbolic AI.

**Weaknesses:**

The proposed method is established in two-layer NN representation without discussing the potential generalization on the (2 + n) layer NN representation and the theoretical limits of the proposed method regarding this potential. Taking into account the non-stochastic nature of the proposed NPN transformation and the required time $O(m2^k) + e$, the proposed algorithm seems quite limited to the 2-layer NN representation. However, a further discussion of this can provide fruitful insights for future work.

Even though the relaxation of the optimization objective provides the required convexity, the problem still remains NP-hard. Indeed, the proposed deterministic solution ensures the lossless functionality of the binary network with the caching solution providing significant acceleration, hindering, however, the scalability of the proposed method in target networks. To this end, I recommend further discussion of the existing stochastic methodologies in the bibliography for lossy solution, studying the accuracy-efficiency tradeoff between deterministic and non-deterministic methodologies. In my opinion, the deterministic linear programing nature of the proposed optimization method should be noted in the abstract of the paper.

**Questions:**

See Weaknesses

---

> ### Author Response · Authors · 2024-11-24
> **Rebuttal by Authors**
>
> We sincerely appreciate your positive feedback and insightful comments.
>
> We have uploaded a new PDF submission with changes listed in the *Global Response to the Reviewers* comment. Please refer to the new PDF when reading our responses below.
>
> ---
>
> **Weakness 1.**
>
> > *The proposed method is established in two-layer NN representation without discussing the potential generalization on the $(2 + n)$ layer NN representation and the theoretical limits of the proposed method regarding this potential. Taking into account the non-stochastic nature of the proposed NPN transformation and the required time $O(m2^k) + e$, the proposed algorithm seems quite limited to the 2-layer NN representation. However, a further discussion of this can provide fruitful insights for future work.*
>
> Thank you for your perceptive comment. As the reviewer points out, the proposed algorithm optimizes two-layer sub-NN representations of BFs by finding MMPs. Consequently, our approach is centralized around the $k$-input BFs that form the $K$-LUT BN, and the construction that maps MPs/MMPs to two-layer NNs. Extending the existing methodology to consider $(2 + n)$ layer NN representations may therefore require grouping multiple BFs and solving for an optimized NN representation of the group. We agree with the reviewer that either extending the proposed optimization technique or developing new methods to consider optimization across multiple layers presents a promising direction for future research. However, such an investigation lies beyond the scope of this work.
>
> ---
>
> **Weakness 2.**
>
> > *Even though the relaxation of the optimization objective provides the required convexity, the problem still remains NP-hard. Indeed, the proposed deterministic solution ensures the lossless functionality of the binary network with the caching solution providing significant acceleration, hindering, however, the scalability of the proposed method in target networks. To this end, I recommend further discussion of the existing stochastic methodologies in the bibliography for lossy solution, studying the accuracy-efficiency tradeoff between deterministic and non-deterministic methodologies. In my opinion, the deterministic linear programing nature of the proposed optimization method should be noted in the abstract of the paper.*
>
> Thank you for your insightful comment. We strongly agree that a detailed analysis of the accuracy-efficiency tradeoff between the proposed lossless optimization and deterministic/non-deterministic lossy optimization techniques would provide significant insight, and we plan to explore this direction in future work. However, since the focus of this paper was on presenting a new lossless optimization technique, we believe a comprehensive comparison to lossy techniques falls beyond its scope.
>
> In light of your suggestion, we have updated the abstract to state that the optimization algorithm we propose is deterministic.

---

> > ### Comment · Reviewer_eC2r · 2024-11-26
> >
> > I would like to thank the authors for answering my comment. I will maintain my score.

---

> > > ### Author Response · Authors · 2024-12-02
> > >
> > > Thank you for taking the time to review our rebuttal and provide a reply. In a time when responses during the rebuttal period are not always guaranteed, we are especially grateful for your engagement and thoughtful consideration.

---

### Official Review · Reviewer_Wkfo · 2024-11-03

**Soundness:** 2
**Presentation:** 2
**Contribution:** 3
**Rating:** 6
**Confidence:** 1

**Summary:**

This paper presents a new approach to optimizing neural network representations of Boolean networks. The authors propose a technique compressing NNs via minimizing  the MP representation of each Boolean function in the network.

**Strengths:**

* The paper provides a solid theoretical foundation for the proposed approach, such as the equation introduction of new concepts like NPN and detailed analysis of the underlying optimization problem.
* Evaluation looks comprehensive, including various bnchmarks, to demonstrate the effectiveness especialy in reducing network size and improving optimization time.
* The proposed method looks interesting and novel as it combines multiple techniques including MP-based mapping, NPN equivalence classes and objec-aware optmization, to optimize representations of BNs.

**Weaknesses:**

Please see questions.

**Questions:**

* how well does it scale, as solving integer LP can be more demanding than solving LPs.
* Computation complexity wiase, how does it compare against SOTAs?
* What's the impat of  NPN equivalence lasses on the optimization process?

---

> ### Author Response · Authors · 2024-11-24
> **Rebuttal by Authors**
>
> Many thanks for your positive feedback and insightful comments.
>
> We have uploaded a new PDF submission with changes listed in the *Global Response to the Reviewers* comment. Please refer to the new PDF when reading our responses below.
>
> ---
>
> **Question 1.**
>
> > *How well does it scale, as solving integer LP can be more demanding than solving LPs.*
>
> The integer linear program (LP) we solve to obtain an MMP representation of a $k$-input BF is an NP-hard problem, and a brute-force search requires time $2^{O(k2^k)}$. As the reviewer points out, this is indeed a more demanding problem than linear programming, for which polynomial-time algorithms are known. Relaxing the MMP integer LP to a LP over the reals would yield a time complexity polynomial in $2^k$. However, such a relaxation can lead to infeasible solutions that would break functional equivalence — the key property our methods are designed to preserve.
>
> To scale to large BNs, our paper discusses the use of other techniques such as $\ell_1$-relaxation of the MMP problem, our proposed NPN classification algorithm, and limiting the size of $k \leq K$ with $K$-LUT mapping.
>
> In light of your question, we have significantly expanded Remark C.2 in Appendix C. Please refer to this remark for a more detailed discussion on this topic.
>
> ---
>
> **Question 2.**
>
> > *Computation complexity wise, how does it compare against SOTAs?*
>
> The SOTA NN-based technology mapping solution of Gavier et al. (2023) computes the MP representation of each $k$-input BF in the BN. However, Gavier et al. (2023) do not specify whether this is done per vertex, or per unique BF in the BN. Consequently, we developed baseline algorithms, *Naive* and *Cached,* to reflect these two kinds of approaches. We have added a comprehensive analysis on the time and space complexity of these methods in comparison to the proposed NPN classification algorithm in Appendix E.4.
>
> ---
>
> **Question 3.**
>
> > *What's the impact of NPN equivalence classes on the optimization process?*
>
> The architecture-aware lossless optimization algorithms we propose require computing the MP and MMP of each BF in the input BN. Since `ttToMP` and `mpToMMP` are computationally demanding, our goal was to minimize the number of times these subroutines are invoked. If $c$ functions are within the same NPN equivalence class, then we only need to compute `ttToMP` for one of them, and can use NPN transformations to find the MPs for the remaining $c-1$. Similarly, if $s$ functions are within the same permuted phase assignment subset of an NPN equivalence class, then we only need to compute `mpToMMP` for one of them, and can use NPN transformations to find the MMPs for the remaining $s-1$ (assuming the criterion we use for MMPs is PN-invariant). In the latter case, the fact that the MMPs we compute using NPN transformations are indeed minima w.r.t. the PN-invariant criterion means the quality (number of reduced connections and neurons) of the NN optimization process is unaffected by the use of NPN equivalence classes, yet by using them we save calls to `ttToMP` and `mpToMMP`. We provide a more detailed analysis in Appendix E.4.
>
> ---
>
> In closing, we are more than willing to address any additional questions or provide further clarification if needed.

---

> > ### Comment · Reviewer_Wkfo · 2024-11-26
> > **Response to Rebuttal**
> >
> > Thank you the authors for providing detailed response to my questions and updating the manuscript.

---

> > > ### Author Response · Authors · 2024-12-02
> > >
> > > Thank you for taking the time to review our rebuttal and provide a reply. In a time when responses during the rebuttal period are not always guaranteed, we are especially grateful for your engagement and thoughtful consideration.
> > >
> > > If our explanations have addressed any uncertainties in your assessment of our submission, we kindly ask you to consider updating your confidence score to reflect this.
> > >
> > > Thank you once again for your time and effort in reviewing our work. Please let us know if there is anything else we can clarify or address.

---

### Official Review · Reviewer_VSF5 · 2024-11-04

**Soundness:** 2
**Presentation:** 3
**Contribution:** 2
**Rating:** 5
**Confidence:** 4

**Summary:**

This paper proposes to optimize neural network representations of Boolean networks by improving the efficiency with NPN classification of sub-networks and considering objective in sub-networks during optimization. It achieves up to 5.9x speedup than the caching solution and reduces neurons and connections of final representations by 60% and 70% respectively.

**Strengths:**

This paper proposes to speedup the optimization of neural network representation of Boolean functions and consider architecture constraints. Instead of solving each subproblems independently, the paper finds solutions of each NPN class and exploit shared optimized representations of two-layer sub-networks. The optimization of sub networks is modeled as finding a polynomial with fewer monomials or monomial degrees. In architecture aware lossless optimization part, the level constraints are considered when performing sub-network replacement.

**Weaknesses:**

1. The scientific novelty is limited. NPN classification and level constraints based DAG optimization are common techniques used in logic synthesis tools and neural network compilers.
2. The k-input LUT technology mapping lacks fair comparison with other traditional DAG optimization methods such as ABC (Boolean DAG optimization tools including technology indepedent and technology dependent optimization) and AI compilers like Google XLA.
3. Only two-layer sub-NN optimization is considered which is relatively too local for better neurons and level optimization.

**Questions:**

1. Please provide comparison with traditional DAG optimization methods for a fair comparison.

---

> ### Author Response · Authors · 2024-11-24
> **Rebuttal by Authors (1/2)**
>
> Many thanks for your insightful comments.
>
> We have uploaded a new PDF submission with changes listed in the *Global Response to the Reviewers* comment. Please refer to the new PDF when reading our responses below.
>
> ---
>
> **Weakness 1.**
>
> >*1. The scientific novelty is limited. NPN classification and level constraints based DAG optimization are common techniques used in logic synthesis tools and neural network compilers.*
>
> For our problem setting, NPN classification can be used to find the MPs within the same NPN class. However, applying NPN classification directly for MMPs will lead to nonoptimal solutions since the optimization cost functions we consider are not invariant within the same NPN class. In this paper, we bring scientific novelty by developing the theoretical results that allow us to utilize NPN classification for MPs and MMPs.
>
> We have added a new section to the Appendix, Section E.5, that discusses the relation of our NPN classification algorithm to existing NPN techniques from logic synthesis, and highlights the subtleties involved with the optimization cost functions we consider.
>
> Regarding DAG optimization techniques, we provide a detailed response under our response to **Weakness 2**.
>
> ---
>
> **Weakness 2.**
>
> >*2. The k-input LUT technology mapping lacks fair comparison with other traditional DAG optimization methods such as ABC (Boolean DAG optimization tools including technology independent and technology dependent optimization) and AI compilers like Google XLA.*
>
> Thank you for your comment, it has helped us to realize where our methods fit into the bigger picture of BN and NN optimization.
>
> We would like to begin by saying that our optimization methodology is not incompatible with (i) DAG optimization or (ii) AI compiler techniques. On the contrary, the former can be applied before K-LUT mapping; for convenience, we assume that the BN is already optimized in the DAG sense. The latter are performed over a NN representation, and can be applied after the NN has been optimized with our method. An illustration of these steps is shown below.
>
> ``
> BN → (i) → BN → (Our Methods) → NN → (ii)
> ``
>
> We believe that there may be a misunderstanding regarding the methodology of our paper and how it relates to (i) logic synthesis DAG optimization methods and (ii) AI/DL/ML compilers. To ensure we are on the same page, we will define (i) and (ii), and then relate them to our methods.
>
> **(i) Logic synthesis DAG optimization methods:**
> - Input: BN
> - Output: size/depth optimized BN
>
> Logic synthesis tools, such as ABC [1] and mockturtle [2], support various optimizations for technology-independent and technology-dependent representations of Boolean networks (BNs). Common BN representations include And-Inverter Graphs (AIGs) and K-LUT networks. Optimization methods for AIGs [3, 4] and K-LUT networks [5], such as rewriting, refactoring, and resubstitution [3], seek to minimize size (measured by the number of nodes in the graphs), while other methods, such as algebraic rebalancing [6], seek to minimize depth.
>
> **Relating Our Methods to (i)**
>
> In Section 2.2 of the paper, we review the SOTA for NN-based technology mapping (Gavier et al., 2023), which takes a BN as input and converts it into a functionally equivalent NN. Note that the logic synthesis DAG optimization methods can be applied to the BN prior to this representation conversion, which is what Gavier et al. (2023) propose with their Yosys + ABC workflow. Hence, these optimizations are orthogonal to our techniques, and can be applied in combination with them.
>
> **(ii) AI compilers:**
> - Input: NN
> - Output: optimized high-level IR, CPU/GPU instructions
>
> AI compilers take a NN model as input, and convert it into a high-level intermediate representation (IR) which we will refer to as the computation graph. Various optimizations are then applied to this representation, such as node-level optimizations (no-op elimination, zero-dim-tensor elimination), block-level optimizations (algebraic simplification, operator fusion, operator sinking) and dataflow-level optimizations (common sub-expression elimination, dead code elimination). The AI compiler will then perform further optimizations that are technology dependent, based on a target architecture (CPU/GPU etc.) [7].
>
> **Relating Our Methods to (ii):**
>
> Our technique optimizes the NN representation of the BN, eliminating neurons and connections from linear layers of the Heaviside NN. The computation graph optimizations briefly reviewed in (ii) of AI compilers do not perform such operations. Indeed, the NN we obtain after optimization can be passed to an AI compiler for further optimization. Hence, these optimizations are also orthogonal to our techniques, and can be applied in combination with them.

---

> ### Author Response · Authors · 2024-11-24
> **Rebuttal by Authors (2/2)**
>
> **Weakness 3.**
>
> > *3. Only two-layer sub-NN optimization is considered which is relatively too local for better neurons and level optimization.*
>
> We would like to highlight that the parameter $K$ governs the degree of locality in the optimization process. Specifically, as $K$ increases, the optimization becomes progressively more global. For instance, in the extreme case of $K = n$, where $n$ is the number of inputs to the BN, the resulting optimized NN reduces to two-layers, and its connections and hidden neurons are optimized by accounting for the entire functionality of the original BN.
>
> Exploring alternative approaches that optimize across multiple layers when $K < n$ is an interesting direction for future research; however, it lies beyond the scope of this work.
>
> ---
>
> **Question 1.**
>
> > *1. Please provide comparison with traditional DAG optimization methods for a fair comparison.*
>
> As discussed in our response to **Weakness 2**, optimization methods for DAG representations of BNs, including AIG/$K$-LUT representations, are orthogonal to our work. Moreover, optimization methods for DAG representations of NN computation graphs that are utilized by AI compilers are also orthogonal to our work. Consequently, we do not include comparisons with these methods.
>
> ---
>
> In closing, we appreciate the reviewer’s insights and recognize that there may be related works we have inadvertently missed. If the reviewer is aware of any specific literature that is closely aligned with our methods, we would be grateful for their recommendations.
>
> ---
>
> **References**
>
> [1] https://github.com/berkeley-abc/abc
>
> [2] https://github.com/lsils/mockturtle
>
> [3] Mishchenko, Alan, Satrajit Chatterjee, and Robert Brayton. "DAG-aware AIG rewriting a fresh look at combinational logic synthesis." *Proceedings of the 43rd annual Design Automation Conference*. 2006.
>
> [4] Li, Yingjie, et al. "DAG-aware Synthesis Orchestration." *IEEE Transactions on Computer-Aided Design of Integrated Circuits and Systems* (2024).
>
> [5] Riener, Heinz, et al. "On-the-fly and DAG-aware: Rewriting Boolean networks with exact synthesis." *2019 Design, Automation & Test in Europe Conference & Exhibition (DATE)*. IEEE, 2019.
>
> [6] Cortadella, Jordi. "Timing-driven logic bi-decomposition." *IEEE Transactions on Computer-Aided Design of Integrated Circuits and Systems* 22.6 (2003): 675-685.
>
> [7] Li, Mingzhen, et al. "The deep learning compiler: A comprehensive survey." *IEEE Transactions on Parallel and Distributed Systems* 32.3 (2020): 708-727.

---

> > ### Author Response · Authors · 2024-11-30
> >
> > Dear Reviewer VSF5,
> >
> > We are grateful for the time you have invested in reviewing our paper. We have carefully studied and addressed each of your comments in our rebuttal.
> >
> > As the rebuttal period is nearing its conclusion, we wanted to kindly follow up to ensure that our responses have reached you. If there are any further clarifications or points of discussion, we would be happy to address them.
> >
> > We deeply value your insights and thank you for your time and effort in reviewing our work.
> >
> > Best regards,
> >
> > Authors

---

> > > ### Author Response · Authors · 2024-12-02
> > >
> > > Dear Reviewer VSF5,
> > >
> > > We hope this message finds you well. With less than 24 hours remaining for reviewers to post messages to the authors, we wanted to kindly follow up regarding our responses to your comments.
> > >
> > > We respectfully ask that you review our submission in light of the additional information we provided in our rebuttal. We would sincerely appreciate at least an acknowledgment of our responses. If there are any remaining points of clarification or concerns, we would be glad to address them promptly.
> > >
> > > Thank you once again for your time and effort in reviewing our work. Your thoughtful consideration means a great deal to us.
> > >
> > > Best regards,
> > >
> > > Authors

---

### Author Response · Authors · 2024-11-24
**Global Response to the Reviewers**

We sincerely thank the reviewers for their comprehensive evaluation of our submission, their insightful suggestions, and their contributions to enhancing the overall quality of our work. Moreover, we are encouraged by the positive feedback from the reviewers in the initial review, particularly regarding several key aspects:

**1) Theoretical contributions.** *The paper provides a solid theoretical foundation […] and detailed analysis of the underlying optimization problem (**Reviewer Wkfo**). The paper is technically sound, with rigorous mathematical formulations and proofs supporting the proposed methods (**Reviewer vaK3**).*

**2) Novelty.** *The paper tackles the problem of optimizing NN representations of Boolean networks from a fresh perspective (**Reviewer vaK3**). The authors identify a specific gap in existing techniques […]. They creatively combine ideas from Boolean function classification (NPN) and convex optimization to develop a new approach to address this gap (**Reviewer vaK3**). The concept of leveraging NPN classes to accelerate the optimization by exploiting shared representations among subproblems is particularly original (**Reviewer vaK3**). The paper establishes a novel framework for lossless optimization techniques for neural network representation of Boolean networks (**Reviewer eC2r**).*

**3) Broader impacts.** *The paper addresses an important problem with practical implications for various domains (**Reviewer vaK3**). The experimental results clearly reflect the contribution of this work (**Reviewer eC2r**). The connection to neurosymbolic AI highlights the potential of the work for advancing this emerging field (**Reviewer vaK3**).*

**4) Presentation.** *The paper is very well organized and written, providing the necessary background as well as the required proofs to support the claims of the authors (**Reviewer eC2r**). The paper is generally well-written and organized (**Reviewer vaK3**). The steps of the algorithms are clearly presented, and the results are reported in a concise and informative manner (**Reviewer vaK3**).*

We also appreciate the insightful and perceptive questions posed by the Reviewers. We have carefully considered each comment and hope that our responses below adequately address the Reviewers’ concerns.

---

The changelog below summarizes the revisions that have been made to the PDF submission.

- **Appendix C (MMP Representation)**
    1. We have expanded Remark C.2 to discuss in detail the time complexity for obtaining an MMP representation of a BF, the significance of the $\ell_1$-relaxation on running time, and why we do not consider a relaxation to a linear program over the reals.
    2. We have included a new Subsection C.1, which addresses the topic of solution suboptimality due to the $\ell_1$-relaxation of MMP problems.
- **Appendix D (Architecture-Aware Lossless Optimization)**
    1. We have added a new Remark D.6 on the space complexity of ``optMaintainDepth``.
- **Appendix E (NPN Classification Algorithm)**
    1. We have significantly expanded Subsection E.4 on the NPN classification algorithm, providing a thorough analysis of time and space complexity in comparison to the baselines.
    2. We have included a new Subsection E.5, which relates our NPN classification algorithm to existing NPN classification techniques in logic synthesis, and discusses important considerations behind our proposed algorithm.
- **Appendix F (Digital Circuits and Automata used in Experiments)**
    1. We have expanded this section to highlight the applicability of our optimization techniques to arbitrary Boolean network domains.
    2. In particular, we added a Boolean network construction for deterministic finite automata, as well as the specifications for two automata we present extended experimental results on.
- **Appendix G (Extended Results)**
    1. In Section G.2, we present new results on optimizing NN representations of BNs that encode deterministic finite automata. We summarize our findings in the text.
- **Appendix I (Broader Impacts)**
    1. We have extended this section to include an example application scenario for a neurosymbolic AI system that could utilize a homogeneous computing architecture. We explain the significance of the proposed optimization methods for such a system.

---

### Meta-Review · Area_Chair_2mSh · 2024-12-21

**Metareview:**

This paper proposes a new and interesting method for optimizing neural network representations of Boolean networks.

Strengths:

1. The paper identifies a unique gap in existing network compression methods,  and proposes a lossless technique for optimizing a two-layer NN representation of a Boolean function. The introduction of NPN to exploit shared representations is particularly interesting.

2. The paper is theoretically solid. Moreover, it provided extensive experimental evaluations.

3. The paper is well-written. It is clear, well-structured.

Weaknesses:

One weakness is about the generalization. The proposed techniques focus on two-layer NNs. Extending the discussion to multi-layer NNs or other types of Boolean networks, such as biological or regulatory networks, would strengthen the claims of general applicability.

Overall, this submission is technically sound, addresses a significant problem, and demonstrates very good experimental results. I would like to recommend acceptance.

**Additional Comments On Reviewer Discussion:**

The authors provided a detailed rebuttal, addressing most of the reviewers’ concerns.

---

### Decision · Program_Chairs · 2025-01-22

Accept (Poster)